# QUASI-OPTIMAL REINFORCEMENT LEARNING WITH CONTINUOUS ACTIONS

**Yuhan Li**[*a]    **Wenzhuo Zhou**[*b]    **Ruoqing Zhu**[a]
[a]University of Illinois Urbana Champaign
[b]University of California Irvine
[*]Equal contribution

## ABSTRACT

Many real-world applications of reinforcement learning (RL) require making decisions in continuous action environments. In particular, determining the optimal dose level plays a vital role in developing medical treatment regimes. One challenge in adapting existing RL algorithms to medical applications, however, is that the popular infinite support stochastic policies, e.g., Gaussian policy, may assign riskily high dosages and harm patients seriously. Hence, it is important to induce a policy class whose support only contains near-optimal actions, and shrink the action-searching area for effectiveness and reliability. To achieve this, we develop a novel *quasi-optimal learning algorithm*, which can be easily optimized in off-policy settings with guaranteed convergence under general function approximations. Theoretically, we analyze the consistency, sample complexity, adaptability, and convergence of the proposed algorithm. We evaluate our algorithm with comprehensive simulated experiments and a dose suggestion real application to Ohio Type 1 diabetes dataset.

## 1 INTRODUCTION

Learning good strategies in a continuous action space is important for many real-world problems (Lillicrap et al., 2015), including precision medicine, autonomous driving, etc. In particular, when developing a new dynamic regime to guide the use of medical treatments, it is often necessary to decide the optimal dose level (Murphy, 2003; Laber et al., 2014; Chen et al., 2016; Zhou et al., 2021). In infinite horizon sequential decision-making settings (Luckett et al., 2019; Shi et al., 2021), learning such a dynamic treatment regime falls into a reinforcement learning (RL) framework. Many RL algorithms (Mnih et al., 2013; Silver et al., 2017; Nachum et al., 2017; Chow et al., 2018b; Hessel et al., 2018) have achieved considerable success when the action space is finite. A straightforward approach to adapting these methods to continuous domains is to discretize the continuous action space. However, this strategy either causes a large bias in coarse discretization (Lee et al., 2018a; Cai et al., 2021a;b) or suffers from the the curse of dimensionality (Chou et al., 2017) for fine-grid.

There has been recent progress on model-free reinforcement learning in continuous action spaces without utilizing discretization. In policy-based methods (Williams, 1992; Sutton et al., 1999; Silver et al., 2014; Duan et al., 2016), a Gaussian distribution is used frequently for policy distribution representation, while its mean and variance are parameterized using function approximation and updated via policy gradient descent. In addition, many actor-critic based approaches, e.g., soft actor-critic (Haarnoja et al., 2018b), ensemble critic (Fujimoto et al., 2018) and Smoothie (Nachum et al., 2018a), have been developed to improve the performance in continuous action spaces. These works target to model a Gaussian policy for action allocations as well.

However, there are two less-investigated issues in the aforementioned RL approaches, especially for their applications in the healthcare (Fatemi et al., 2021; Yu et al., 2021). First, existing methods that use an infinite support Gaussian policy as the treatment policy may assign arbitrarily high dose levels, which may potentially harm the patient (Yanase et al., 2020). Hence, these approaches are not reliable in practice due to safety and ethical concerns. It would be more desirable to develop a policy class to identify the near-optimal (Tang et al., 2020), or at least safe, action regions, and reduce

---

¶ Correspondence to: Wenzhuo Zhou <wenzhuz3@uci.edu> and Ruoqing Zhu <rqzhu@illinois.edu>

the optimal action search area for reliability and effectiveness. Those actions out of the identified region are discriminated as non-optimal, and would be screened out with zero densities in the policy distribution. Second, for many real-world applications, the action spaces are bounded due to practical constraints. Examples include autonomous driving with a limited steering angle and dose assignment with a budget or safety constraint. In these scenarios, modeling an optimal policy by an infinite support probability distribution, e.g., Gaussian policy, would inevitably introduce a non-negligible off-support bias as shown in Figure 2. In consequence, the off-support bias damages the performance of policy learning and results in a biased decision-making procedure. Instead, constructing a policy class with finite but adjustable support might be one of the demanding solutions.

In this work, we take a substantial step towards solving the aforementioned issues by developing a novel *quasi-optimal learning* algorithm. Our development hinges upon a novel quasi-optimal Bellman operator and stationarity equation, which is solved via minimizing an unbiased kernel embedding loss. Quasi-optimal learning estimates an implicit stochastic policy distribution whose support region only contains near-optimal actions. In addition, our algorithm overcomes the difficulties of the non-smoothness learning issue and the double sampling issue (Baird, 1995), and can be easily optimized using sampled transitions in off-policy scenarios without training instability and divergence. The main contribution of this paper can be summarized as follows:

- We construct a novel Bellman operator and develop a reliable stochastic policy class, which is able to identify quasi-optimal action regions in scenarios with a bounded or unbounded action space. This address the shortcomings of existing approaches relying on modeling an optimal policy with infinite support distributions.
- We formalize an unbiased learning framework for estimating the designed quasi-optimal policy. Our framework avoids the double sampling issue and can be optimized using sampled transitions, which is beneficial in offline policy optimization tasks.
- We thoroughly investigate the theoretical properties of the quasi-optimal learning algorithm, including the adaptability of the quasi-optimal policy class, the loss consistency, the finite-sample bound for performance error, and the convergence analysis of the algorithm.
- Empirical analyses are conducted with comprehensive numerical experiments and a real-world case study, to evaluate the model performance in practice.

## 2 PRELIMINARIES

**Notations** We first give an introduction to our notations. For two strictly positive sequences $\{\Psi(m)\}_{m\geq 1}$ and $\{\Upsilon(m)\}_{m\geq 1}$, the notation $\{\Psi(m)\}_{m\geq 1} \lesssim \{\Upsilon(m)\}_{m\geq 1}$ means that there exists a sufficiently small constant $c \geq 0$ such that $\Psi(n) \leq c\Upsilon(n)$. $\|\cdot\|_{L^p}$ and $\|\cdot\|_\infty$ denote the $L^p$ norm and supremum-norm, respectively. We define the set indicator function $\mathbb{1}_{\text{set}}(x) = 1$ if $x \in$ set or $0$ otherwise. The notation $\mathbb{P}_n$ denotes the empirical measure i.e., $\mathbb{P}_n = \frac{1}{n}\sum_{i=1}^n$. For two sets $\aleph_0$ and $\aleph_1$, the notation $\aleph_0 \setminus \aleph_1$ indicates that the set $\aleph_0$ excluding the elements in the set $\aleph_1$. We write $|\aleph_0|$ as the cardinality of the set $\aleph_0$. For any Borel set $\aleph_2$, we denote $\sigma(\aleph_2)$ as the Borel measure of $\aleph_2$. We denote a probability simplex over a space $\mathcal{F}$ by $\Delta(\mathcal{F})$, and in particular, $\Delta_{\text{convex}}(\mathcal{F})$ indicates the convex probability simplex over $\mathcal{F}$. We denote $\lfloor\cdot\rfloor$ as the floor function, and use $\mathcal{O}$ as the convention.

**Background** A Markov decision process (MDP) is defined as a tuple $< \mathcal{S}, \mathcal{A}, \mathbf{P}, R, \gamma >$, where $\mathcal{S}$ is the state space, $\mathcal{A}$ is the action space, $\mathbf{P} : \mathcal{S} \times \mathcal{A} \to \Delta(\mathcal{S})$ is the unknown transitional kernel, $R : \mathcal{S} \times \mathcal{S} \times \mathcal{A} \to \mathbb{R}$ is a bounded reward function, and $\gamma \in [0, 1)$ is the discounted factor. In this paper, we focus on the scenario of continuous action space. We assume the offline data consists of $n$ i.i.d. trajectories, i.e., $\mathcal{D}_{1:n} = \{S_i^1, A_i^1, R_i^1, S_i^2, \ldots, S_i^{T_i}, A_i^T, R_i^T, S_i^{T+1}\}_{i=1}^n$, where the length of trajectory $T$ is assumed to be non-random for simplicity. A policy $\pi$ is a map from the state space to the action space $\pi : \mathcal{S} \to \mathcal{A}$. The learning goal is to search an optimal policy $\pi^*$ which maximizes the expected discounted sum of rewards. $V_t^\pi(s) = \mathbb{E}_\pi\left[\sum_{k=1}^\infty \gamma^{k-1} R^{t+k}|S^t = s\right]$ is the value function under a policy $\pi$, where $\mathbb{E}_\pi$ is taken by assuming that the system follows a policy $\pi$, and the Q-function is defined as $Q_t^\pi(s, a) = \mathbb{E}_\pi\left[\sum_{k=1}^\infty \gamma^{k-1} R^{t+k}|S^t = s, A^t = a\right]$. In a time-homogenous Markov process (Puterman, 2014), $V_t^\pi(s)$ and $Q_t^\pi(s, a)$ do not depend on $t$. The optimal value function $V^*$ is the unique fixed point of the Bellman operator $\mathcal{B}$, $\mathcal{B}V(s) := \max_a \mathbb{E}_{S^{t+1}\sim\mathbf{P}(s,a)}\left[R^t + \gamma V(S^{t+1})|S^t = s, A^t = a\right]$. Then $\mathcal{B}V^*(s) = V^*(s)$ for any $s \in \mathcal{S}$. An optimal policy $\pi^*$ can be obtained by taking the *greedy* action of $Q^*(s, a)$, that is $\pi^*(s) = \arg\max_a Q^*(s, a)$. For the rest of the paper, we use the short notation $\mathbb{E}_{s'|s,a}$ for the conditional expectation $\mathbb{E}_{s'\sim\mathbf{P}(s,a)}$; and $\mathbb{E}_{S^t, A^t, S^{t+1}}$ is short for $\mathbb{E}_{S^t\sim\upsilon, A^t\sim\pi_b(\cdot|S^t), S^{t+1}\sim\mathbf{P}(S^t, A^t)}$, where $\upsilon$ is a some fixed distribution and $\pi_b$ is some behavior policy.

## 3 METHODOLOGY

To start with, we first revisit the Bellman optimality equation via a policy explicit view,

$$\mathcal{B}V^*(s) := \max_\pi \mathbb{E}_{a\sim\pi(\cdot|s),\ S^{t+1}|s,a}\left[R(S^{t+1},s,a) + \gamma V^*(S^{t+1})\right] = V^*(s). \tag{1}$$

To obtain the optimal policy $\pi^*$ and value function $V^*$, an optimization idea is to minimize the discrepancy between the two sides of the equation under a $L^2$ loss. Unfortunately, there are several major challenges when it comes to optimization: (1) **Non-smoothness**: the Bellman operator involves a non-smoothed hard-$\max$ operator, which leads to training instability; (2) **Policy class**: As discussed in Section 1, it is necessary to induce an optimal policy class whose support consists of quasi-optimal sub-regions for reliability, and avoids off-support bias in Figure 2; (3) **Double sampling**: the unknown conditional expectation $\mathbb{E}_{S^{t+1}|s,a}$ is required to be double sampled for obtaining an unbiased sample approximation for $\mathbb{E}_{S^{t+1}|s,a}$. However, this is usually infeasible in real-world environments; (4) **Off-policy data**: directly minimizing the Bellman error is not easy to incorporate off-policy data. To address these issues, we propose a quasi-optimal counterpart of the Bellman equation (1).

### 3.1 QUASI-OPTIMAL BELLMAN OPERATOR

In this subsection, we aim to tackle the first two challenges. We propose a quasi-optimal counterpart for the Bellman operator $\mathcal{B}$ that simultaneously circumvents the non-smoothness obstacles, and induce a novel policy class which can identify quasi-optimal sub-regions in continuous action spaces.

We leverage the Legendre-Fenchel transform (Hiriart-Urruty & Lemaréchal, 2012) on the Bellman operator $\mathcal{B}$. For a convex probability simplex $\Delta_{\text{convex}}(\mathcal{A})$ and a strongly convex and continuous proximity function $\text{prox}(\pi) : \Delta_{\text{convex}}(\mathcal{A}) \to \mathbb{R}$, the Fenchel transform counterpart of $\mathcal{B}$ is defined as

$$\mathcal{B}_\mu V_\mu^*(s) = \max_{\pi \in \Delta_{\text{convex}}(\mathcal{A})} \int_{a\in\mathcal{A}} \left[Q_\mu^*(s,a)\pi(a|s) + \mu\text{prox}(\pi(a|s))\right] da, \tag{2}$$

where $Q_\mu^*(s,a) = \mathbb{E}_{S^{t+1}|s,a}[R(S^{t+1},s,a) + \gamma V_\mu^*(S^{t+1})]$, and $V_\mu^*(s)$ is the unique fixed point of the quasi-optimal Bellman operator $\mathcal{B}_\mu$. Note that, besides the smoothing purpose, we are also interested in constructing a stochastic optimal policy class that can screen out the non-optimal and sub-optimal actions. Therefore, we further define a special prox function class motivated by the rationale of $q$-logarithm as $\text{prox}(x) = \log_q(x) := \frac{x(1-x^{q-1})}{q-1}$, where $\int_{a\in\mathcal{A}} \text{prox}(\pi(a|s)) da = \frac{1}{q-1}(1 - \int_{a\in\mathcal{A}} \pi^q(a|s) da)$ essentially generalize the Shannon's entropy (Martins et al., 2020). In this paper, we focus on the setting that $q = 2$.

**Assumption 3.1.** *For any policy distribution $\pi \in \Delta_{convex}(\mathcal{A})$, its density is bounded above by a constant, i.e., $\pi(\cdot|s) \leq \mathbf{C}$ for all $s \in \mathcal{S}$.*

This assumption avoids some extreme cases where a stochastic policy distribution degenerates to be deterministic. In the following, we show several nice properties of the proposed Bellman operator.

**Proximal Approximation** The operator $\mathcal{B}_\mu$ is a proximal approximation to $\mathcal{B}$. This delivers two messages: firstly, the approximation bias is upper bounded; secondly, the operator $\mathcal{B}_\mu$ is a smoothed substitute for $\mathcal{B}$. In particular, Theorem 3.1 demonstrates that the approximation bias can vanish to zero for small enough $\mu$. In addition, the operator $\mathcal{B}_\mu$ has a differentiable and analytical form (3), which justifies that $\mathcal{B}_\mu$ is a smoothed counterpart of $\mathcal{B}$, see Corollary S.1 in Appendix for details.

**Theorem 3.1** (Proximal bias). *Under Assumption 3.1, for any $s \in \mathcal{S}$ and value function $V$, $\mathcal{B}_\mu V(s) - \mathcal{B}V(s) \in [\mu(1-\mathbf{C}), \mu]$.*

$$\mathcal{B}_\mu V_\mu^*(s) = \mu - \frac{1}{4\mu}\left(\frac{(\int_{a'\in\mathcal{W}_s} Q_\mu^*(s,a')da' - 2\mu)^2}{\sigma(\mathcal{W}_s)} - \int_{a\in\mathcal{W}_s} Q_\mu^{*\,2}(s,a)da\right), \tag{3}$$

where $\mathcal{W}_s$ denotes the the support of $\pi_\mu^*$ in (4) for a given state $s$.

**Quasi-optimal Support Region** In addition to the proximal approximation property, another unique and important property of $\mathcal{B}_\mu$ is inducing a policy $\pi_\mu^*$ whose support region contains all the actions with action-value higher than a certain threshold. The induced policy $\pi_\mu^*$ is bridged from the oracle Q-function:

$$\pi_\mu^*(a|s) = \left(\frac{Q_\mu^*(s,a)}{2\mu} - \frac{\int_{a\in\mathcal{W}_s} Q_\mu^*(s,a)da}{2\mu\sigma(\mathcal{W}_s)} + \frac{1}{\sigma(\mathcal{W}_s)}\right)^+, \tag{4}$$

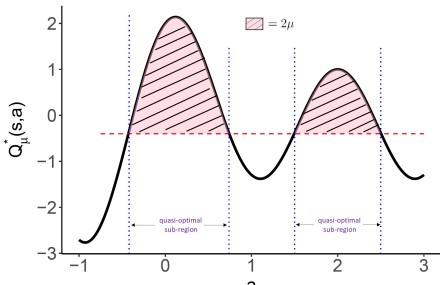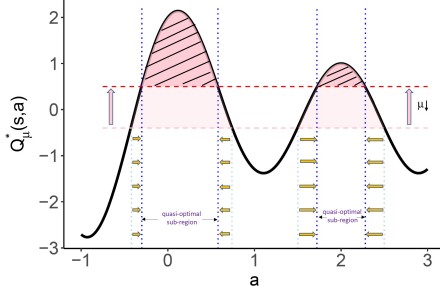

Figure 1: An illustrating example of the quasi-optimal sub-regions. In the left panel, the lowest admissible action-value corresponds to the horizontal red dashed line, and the integral difference is the shadowed pink area, which equals $2\mu$. As shown in the right panel, when $\mu$ decreases, the pink area shrinks, and the quasi-optimal sub-regions become narrower.

where the support of $\pi_\mu^*$, i.e., $\mathcal{W}_s := \bigcup_{a \in \mathcal{A}} a \mathbb{1}_{\texttt{screening set}}(a)$ with

$$\texttt{screening set} := \left\{ a \in \mathcal{A} : \int_{a' \in \mathcal{M}_s(a)} Q_\mu^*(s, a') da' - \sigma(\mathcal{M}_s(a)) Q_\mu^*(s, a) > 2\mu \right\}, \quad (5)$$

$$\mathcal{M}_s(a) := \bigcup_{a' \in \mathcal{A}} a' \mathbb{1}_{\{Q_\mu^*(s,a') > Q_\mu^*(s,a)\}}(a'). \quad (6)$$

This mechanism allows us to identify multiple sub-regions in the entire action space which only contains near-optimal actions, and weed out the sub-optimal and non-optimal support regions. Note that, the identified sub-region might not be joint in general, which is beneficial to the situation that the true Q-function has multiple modes. The screening set in (5) indicates that the threshold parameter $\mu$ not only controls the degree of smoothness, but also determines how the quasi-optimal region behaves and controls the screening intensity, as shown in Figure 1.

## 3.2 $q$-GAUSSIAN POLICY DISTRIBUTION

In this section, we bridge the induced policy distribution $\pi_\mu^*$ to an explainable $q$-Gaussian distribution. The $q$-Gaussian distribution is less favored for heavy tails, which makes it widely used in practice to model the effect of external stochasticity (d'Onofrio, 2013). In continuous actions problems, e.g., medical dose suggestion, the $q$-Gaussian distribution is a more suitable choice than the Gaussian distribution for policy modeling, since it can filter out non-optimal and risky dose levels, i.e., too high or too low dosage.

Motivated by the fact that the induced policy $\pi_\mu^*$ is feasible to identify quasi-optimal support sub-regions, and $q$-Gaussian policy distribution can realize bounded support, we conjectured that the $q$-Gaussian policy distribution might be recovered from the induced policy $\pi_\mu^*$. Fortunately, the $q$-Gaussian policy distribution is indeed a special case of the induced policy if $Q_\mu^*(s, a)$ is a concavely quadratic function with respect to the action $a$. We illustrate this phenomenon in Theorem 3.2.

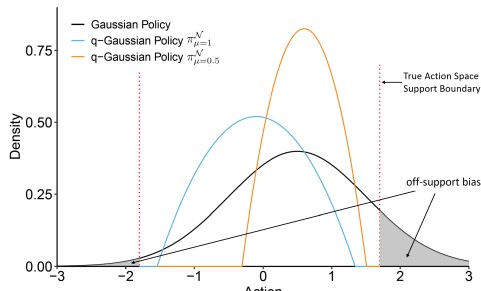

Figure 2: An illustrating example of bounded action space and q-Gaussian policy distribution. The Gaussian policy assigns non-zero probabilities density to all actions, even for those actions outside of the true action space support boundary. This causes the off-support bias. In contrast, the q-Gaussian policy relieves such off-support bias blessed by the boundedness of the quasi-optimal region.

**Theorem 3.2.** *Suppose $Q_\mu^*(s, a)$ is a concavely quadratic function over $a \in \mathcal{A}$, i.e., $Q_\mu^*(s, a) = -\alpha_1(s)a^2 + \alpha_2(s)a + \alpha_3(s) := Q_\mu^{\mathcal{N}}(s, a)$ where $\alpha_1(s), \alpha_2(s), \alpha_3(s)$ are functions over $s \in \mathcal{S}$ and $\alpha_1(s) > 0$ for all $s$, then the induced policy distribution $\pi_\mu^*(\cdot|s)$ would follow a q-Gaussian distribution with a density function*

$$\pi_\mu^*(a|s) = \left( \frac{\alpha_1(s)}{2\mu} \left( a + \frac{\alpha_2(s)}{2\alpha_1(s)} \right)^2 - \frac{3}{2} \left( \frac{\alpha_1(s)}{12\mu} \right)^{\frac{1}{3}} \right)^+ := \pi_\mu^{\mathcal{N}}(a|s), \quad (7)$$

*and a closed-form quasi-optimal support region*

$$\mathcal{W}_s = \left[ \frac{\alpha_2(s) - (12\alpha_1^2(s)\mu)^{\frac{1}{3}}}{2\alpha_1(s)}, \frac{\alpha_2(s) + (12\alpha_1^2(s)\mu)^{\frac{1}{3}}}{2\alpha_1(s)} \right] := \mathcal{W}_s^{\mathcal{N}}. \tag{8}$$

The policy distribution $\pi_\mu^{\mathcal{N}}(\cdot|s)$ behaves as a affine transformation of the standard $q$-Gaussian distribution with mean $-\frac{\alpha_2(s)}{2\alpha_1(s)}$, where the maximum action-value attains, i.e., $Q_\mu^{\mathcal{N}}(s, -\frac{\alpha_2(s)}{2\alpha_1(s)}) = \arg\max_{a \in \mathcal{A}} Q_\mu^{\mathcal{N}}(s, a)$. Note that the width of the quasi-optimal region is $\frac{(12\alpha_1^2(s)\mu)^{\frac{1}{3}}}{\alpha_1(s)}$ determined by the threshold parameter $\mu$. The actions within the region $\mathbb{R} \setminus \mathcal{W}_s^{\mathcal{N}}$ are discriminated as the non-optimal and would be assigned with zero probability densities. For a small $\mu$, i.e., strong screening intensity, a narrow region would be identified as the quasi-optimal, which yields a relatively conservative action recommendation. In contrast, with a large $\mu$, more actions are included in the support. In an extreme case, $\mathcal{W}_s^{\mathcal{N}}$ degenerates to $\mathbb{R}$ as $\mu \to \infty$. In Theorem 4.1 of Section 4, we investigate how the intensity of $\mu$ affects the induced policy distribution formally.

So far, we have obtained the closed-form representations for the general policy $\pi_\mu^*(\cdot|s)$ and $q$-Gaussian policy $\pi_\mu^{\mathcal{N}}$. However, how to make a policy estimation remains unknown. Indicated by the challenges in Section 3, we need to address the double sampling issue and utilize off-policy data in optimization. Both challenges cannot be easily solved by minimizing the Bellman error. Fortunately, the kernel embedding helps us to bypass the difficulties.

### 3.3 KERNEL EMBEDDING ON QUASI-OPTIMAL ERROR

In this subsection, we introduce the quasi-optimal learning framework for solving the induced policy $\pi_\mu^*$. First, we establish a stationary equation in Theorem 3.3. This helps to incorporate off-policy data. Then we leverage the idea of the kernel embedding (Gretton et al., 2012) to obtain an unbiased empirical loss without the double sampling issue.

**Theorem 3.3** (Stationarity equation). *Let $V_\mu^*$ be a fixed point of the quasi-optimal Bellman operator $\mathcal{B}_\mu$, and $\pi_\mu^*$ is the induced policy in (4). For any $s \in \mathcal{S}, a \in \mathcal{A}$, and $\mu \in (0, \infty)$, the pair $(V_\mu^*, \pi_\mu^*)$ satisfies the following equation:*

$$\mathbb{E}_{S^{t+1}|s,a} \left[ R(S^{t+1}, s, a) + \gamma V_\mu(S^{t+1}) \right] - \mu \text{prox}^\circ(\pi_\mu(a|s)) - \eta(s) + \varpi(s, a) = V_\mu(s). \tag{9}$$

*Here* $\text{prox}^\circ(x) = 2x - 1$, $\eta(s) : \mathcal{S} \to [-\mu\mathbf{C}, 0]$ *and* $\varpi(s, a) : \mathcal{S} \times \mathcal{A} \to \mathbb{R}^+$ *are Lagrange multipliers that* $\varpi(s, a) \cdot \pi_\mu(a|s) = 0$. *The discrepancy between the two sides of (9) is "quasi-optimal error".*

The equation (9) connects quasi-optimal value function $V_\mu^*$ and policy function $\pi_\mu^*$ along with any arbitrary state-action pair. This provides an easy way to incorporate off-policy data, i.e., the state-action pairs which are sampled from state-action visitation under the behavior policy, without adjusting the distribution mismatch.

**Min-max Optimization** One way to solve the equation (9) is minimizing the quasi-optimal error under a $L^2$ loss function. Unfortunately, the double sampling issue would still appear if replacing the unknown $\mathbb{E}_{S^{t+1}|s,a}[R(S^{t+1}, s, a) + \gamma V_\mu(S^{t+1})]$ in the quasi-optimal error by its one-sample bootstrapping counterpart $R^t + \gamma V_\mu(S^{t+1})$. Alternatively, inspired by the average Bellman error (Jiang et al., 2017), we propose to minimize a weighted average quasi-optimal error, and the unwanted conditional variance of the bootstrapping counterpart under $L^2$ loss could vanish. We define the loss $\mathcal{L}(V_\mu, \pi_\mu, \eta, \varpi, u)$ as

$$\mathbb{E}_{S^t, A^t, S^{t+1}} \left[ u\left(S^t, A^t\right) \cdot \left(\mathcal{G}_{V_\mu, \pi_\mu}\left(S^t, A^t, S^{t+1}\right) - \eta(S^t) + \varpi(S^t, A^t) - V_\mu(S^t)\right) \right],$$

where $\mathcal{G}_{V_\mu, \pi_\mu}(s, a, s') := R(s', s, a) + \gamma V_\mu(s') - \mu \text{prox}^\circ(\pi_\mu(a|s))$ and $u(\cdot) : \mathcal{S} \times \mathcal{A} \to \mathcal{R}$ is a bounded function in $L^2$ space $L^2(C_0) := \{u \in L^2 : \|u\|_{L^2} \leq C_0\}$. Essentially, the weight function $u$ is to fit the discrepancy of (9) and promotes the sample points with large quasi-optimal errors.

As $\mathcal{L}(V_\mu^*, \pi_\mu^*, \eta, \varpi, u) = 0$ holds for any $u$ function, this leads to a minimax optimization:

$$\min_{V_\mu, \pi_\mu, \eta, \varpi} \max_{u \in L^2(C_0)} \mathcal{L}^2(V_\mu, \pi_\mu, \eta, \varpi, u). \tag{10}$$

---

**Algorithm 1** Quasi-optimal Learning in Continuous Action Spaces

---

1: **Input** observed transition pairs data $\{(S_i^t, A_i^t, R_i^t, S_i^{t+1}) : t = 1, ..., T\}_{i=1}^n$.
2: **Initialize** the parameters of interests $(\theta, \xi) = (\theta^0, \xi^0)$, the mini-batch size $n_0$, the learning rate $\alpha_0$, the prox parameter $\mu$, the kernel bandwidth $bw_0$, and the stopping criterion $\varepsilon$.
3: **For iterations** $j = 1$ to $k$
4:     Randomly sample a mini-batch $\{(S_i^t, A_i^t, R_i^t, S_i^{t+1}) : t = 1, ..., T\}_{i=1}^{n_0}$.
5:     Decay the learning rate $\alpha_j = \mathcal{O}(j^{-1/2})$.
6:     Compute stochastic gradients with respect to $\theta$ and $\xi$: $\bar{\nabla}_\theta = \mathbb{P}_{n_0} \widehat{\nabla}_\theta \widehat{\mathcal{L}_U}$ and $\bar{\nabla}_\xi = \mathbb{P}_{n_0} \widehat{\nabla}_\xi \widehat{\mathcal{L}_U}$.
7:     Update the parameters of interest as $\theta^j \leftarrow \theta^{j-1} - \alpha_j \bar{\nabla}_\theta \widehat{\mathcal{L}_U}$, $\xi^j \leftarrow \xi^{j-1} - \alpha_j \bar{\nabla}_\xi \widehat{\mathcal{L}_U}$.
8:     Stop if $\|(\theta^j, \xi^j) - (\theta^{j-1}, \xi^{j-1})\| \leq \varepsilon$.
9: **Return** $\widehat{\theta} \leftarrow \theta^j, \widehat{\xi} \leftarrow \xi^j$.

---

**Kernel Representation** Solving the minimax optimization problem (10) is unstable, and it is also intractable due to the difficulty for the representation of $u$ in $L^2$ space. Fortunately, we identify continuity invariance between the reward function and the optimal weight function $u^*(\cdot)$ (see Theorem S.2 in Appendix). The optimal $u^*(\cdot)$ is continuous as long as the reward function is continuous, which is widely satisfied in real-world applications. As for a positive definite kernel $K$, a bounded reproducing kernel Hilbert space (RKHS) $H_{\text{RKHS}}(C_0) := \{u \in H_{\text{RKHS}} : \|u\|_K \leq C_0\}$ has a diminishing approximation error to any continuous function class as $C_0 \to \infty$ (Bach, 2017). This together with continuity invariance provides us a basis for representing the weight function in a bounded RKHS. This kernel representation further leads to a closed-form of the inner optimization maximizer (Gretton et al., 2012). The detailed derivation is provided in Theorem S.3 in Appendix. Upon this, the minimax optimization is reduced to only minimizing the loss

$$\mathcal{L}_U = \mathbb{E}_{S^t, \tilde{S}^t, A^t, \tilde{A}^t, S^{t+1}, \tilde{S}^{t+1}}[\Lambda_{V_\mu, \pi_\mu}(S^t, A^t, S^{t+1})K(S^t, A^t; \tilde{S}^t, \tilde{A}^t)\Lambda_{V_\mu, \pi_\mu}(\tilde{S}^t, \tilde{A}^t, \tilde{S}^{t+1})], \quad (11)$$

where $\Lambda_{V_\mu, \pi_\mu}(s, a, s') := \mathcal{G}_{V_\mu, \pi_\mu}(s, a, s') - \eta(s) + \varpi(s, a) - V_\mu(s)$ and $(\tilde{S}^t, \tilde{A}^t, \tilde{S}^{t+1})$ is an independent copy of transition pair $(S^t, A^t, S^{t+1})$.

It observes that the loss $\mathcal{L}_U$ is symmetric and kernel represented. This motivates us to use an unbiased U-statistic estimator to obtain the sample loss. Given the observed data, $\mathcal{D}_{1:n}$, with $n$ trajectories of length $T$, we can use a trajectory-based U-statistic estimator to capture the within-trajectory loss, thus the total loss $\mathcal{L}_U$ can be aggregated as the empirical mean of $n$ i.i.d. within trajectory loss:

$$\min_{V_\mu, \pi_\mu, \eta, \varpi} \widehat{\mathcal{L}_U} = \mathbb{P}_n \binom{T}{2} \sum_{1 \leq j \neq k \leq T} [\Lambda_{V_\mu, \pi_\mu}(S_i^j, A_i^j, S_i^{j+1})K(S^j, A^j; S^k, A^k)\Lambda_{V_\mu, \pi_\mu}(S_i^k, A_i^k, S_i^{k+1})]$$

$$\text{s.t.} \quad \varpi(a|s) \geq 0, \pi_\mu(a|s) \cdot \varpi(a|s) = 0 \ \text{and} \ \eta(s) \in [-\mu\mathbf{C}, 0] \ \text{for all} \ s \in \mathcal{S}, a \in \mathcal{A}. \quad (12)$$

The sample loss $\widehat{\mathcal{L}_U}$ is unbiased and consistent with the population loss $\mathcal{L}_U$. The consistency is justified in Theorem 4.2 via examining the tail behavior of $\widehat{\mathcal{L}_U}$. In essence, solving the equation (12) is a computationally intensive non-linear programming problem. Alternatively, we convert the constrained problem to an unconstrained problem by restricting the Lagrange multipliers. Thus, it can be solved by an unconstrained true gradient algorithm, i.e., Algorithm 1 under function approximation $(V_\mu, \pi_\mu, \eta, \varpi) = (V_\mu^\theta, \pi_\mu^\theta, \eta^\xi, \varpi^\theta)$; see Appendix for details.

## 4 THEORY

In this section, we study the theoretical properties of the proposed method. First, we study some general properties of the proposed quasi-optimal Bellman operator, given in Proposition S.1 and S.2 of Appendix. In Theorem 4.1, we disclose the effect of the intensity of prox parameter $\mu$ on the induced optimal policy distribution. Moreover, a non-asymptotic concentration bound is established in Theorem 4.2, showing the consistency and measuring the rate of convergence of $\widehat{\mathcal{L}_U}$ to $\mathcal{L}_U$. Further, the overall performance error of the algorithm is given in Theorem 4.3, where the performance error is decomposed as the four sources. Finally, we show that the proposed quasi-optimal learning is a convergent algorithm. Before we present the theoretical results, we introduce some assumptions on the boundedness condition of the MDP and the sample trajectory properties, respectively.

**Assumption 4.1.** *The reward function $R(s', s, a)$ is uniformly bounded, i.e, $\|R(\cdot)\|_\infty \leq R_{\max}$.*

**Assumption 4.2.** *Suppose $\{S^t, A^t\}_{t \geq 1}$ is a strictly stationary and exponentially $\beta$-mixing sequence with a mixing coefficient $\beta(m) \lesssim \exp(-\delta_1 m)$ for $m \geq 1$. We further assume that the behavior policy $\pi_b$, which is used to collect the offline data $\mathcal{D}_{1:n}$, satisfies that $\min_{a \in \mathcal{A}, s \in \mathcal{S}} \pi_b(a|s) > 0$.*

**Theorem 4.1** (Policy Adaptability). *Under Assumption 4.1, for all $s \in \mathcal{S}$, the quasi-optimal policy distribution $\pi_\mu^*(\cdot|s)$ degenerates to a uniform distribution over $\Delta(\mathcal{A})$ as $\mu \to \infty$, and $\pi_\mu^*(\cdot|s)$ concentrates in a point mass as $\mu \to 0$ and $\mathbf{C} \to \infty$.*

Theorem 4.1 formally investigates the effect of $\mu$ on $\pi_\mu^*(\cdot|s)$. In an extreme case that $\mu \to 0, \mathbf{C} \to \infty$, only the action maximizing $Q_\mu^*(s, a)$ would be included in the quasi-optimal region. In the following, we establish a non-asymptotic concentration inequality for the empirical loss in the non-i.i.d. case.

**Theorem 4.2.** *For any $\mu \in (0, \infty)$ and $\epsilon > 0$, under Assumptions 4.1-4.2, we have $\epsilon$-divergence of $|\widehat{\mathcal{L}}_U - \mathcal{L}_U|$ bounded in probability, i.e.,*

$$\mathbb{P}(|\widehat{\mathcal{L}}_U - \mathcal{L}_U| > \epsilon) \leq C_1 \exp\left(-\frac{\epsilon^2 T - C_2 \epsilon M_{\max}^2 \sqrt{T}}{M_{\max}^2 + (\frac{\epsilon}{2} - \frac{C_2 M_{\max}^2}{\sqrt{T}}) \log T \log \log(T)}\right) + C_3 \exp\left(\frac{-n\epsilon^2}{M_{\max}^4}\right),$$

*where $C_1, C_2$ and $C_3$ are some constants depending on $\delta_1$ respectively, and $M_{\max} = \frac{4}{1-\gamma} R_{\max} + \mu\mathbf{C}$.* Theorem 4.2 implies that $\widehat{\mathcal{L}}_U$ is a consistent estimator to $\mathcal{L}_U$, and thus avoiding the double sampling issue. Note that the concentration bound is sharper than the bound established in Chakrabortty & Kuchibhotla (2018) since we utilize a novel temporal correlatedness structure to decompose the U-statistic. We now analyze the performance error between the finite sample learner and true solution, which can be decomposed into four source errors.

**Theorem 4.3.** *Under Assumption 4.1-4.2, let $V_\mu^{\theta_1, k}$ be the optimizer from Algorithm 1 and $V^*$ is the optimal value function and $\kappa_{\min}$ be the smallest eigenvalue corresponding to an orthonormal basis of $L^2(\mathcal{S} \times \mathcal{A})$ space. With probability $1 - \delta$, the performance error is upper bounded by*

$$\|\widehat{V}_\mu^{\theta_1, k} - V^*\|_{L^2}^2 \leq \underbrace{\frac{C_4}{\kappa_{\min}(1-\gamma)^2}\left(\sqrt{\frac{C_5 D_{P\text{-}dim} \log\left(\frac{8C_4}{\delta}\right)}{n}} + \sqrt{\frac{2(\frac{\bar{\Delta}}{\delta_1} \vee 1)\bar{\Delta}}{C_6 \lfloor T/2 \rfloor}}\right)}_{\text{generalization error}} +$$

$$\underbrace{C_7 \frac{\mu^2(\mathbf{C} + |1 - \mathbf{C}| \vee 1)^2}{(1-\gamma)^2}}_{\text{proximal bias}} + \underbrace{C_8\left\|\widehat{V}_\mu^{\theta_1} - \widehat{V}_\mu^{\theta_1, k}\right\|_{L^2}^2}_{\text{optimization error}} + \epsilon_{\text{approximation error}},$$

*where $\bar{\Delta} = \frac{D_{P\text{-}dim} \log \lfloor T/2 \rfloor}{2} + \log(\frac{e}{\delta}) + \log^+\left(\frac{C_5 C_6^{D_{P\text{-}dim}}}{2}\right)$, $D_{P\text{-}dim} = P\text{-}dim(\Theta_1) + P\text{-}dim(\Theta_2) + P\text{-}dim(\Xi_1) + P\text{-}dim(\Xi_2)$, and $C_4, ..., C_8$ are some constants. Here $P\text{-}dim(\cdot)$ denotes the pseudo-dimension operator (Györfi, 2010), and $\Theta_1, \Theta_2, \Xi_1$ and $\Xi_2$ are function spaces for $V_\mu, \pi_\mu, \varpi$ and $\eta$, respectively. The $\epsilon_{\text{approximation error}}$ is from parametrization $(V_\mu^{\theta_1}, \pi_\mu^{\theta_2}, \varpi^{\xi_1}, \eta^{\xi_2})$ on $(V_\mu, \pi_\mu, \varpi, \eta)$.*

The above sample complexity bound gives an insight into the performance error of the proposed algorithm. The generalization error $\varepsilon_{\text{gerr}} = \mathcal{O}(1/\sqrt{T})$ if $n$ is as the same order of $T$, the proximal bias $\varepsilon_{\text{prox}} = \mathcal{O}(\mu^2)$ and the optimization error $\varepsilon_{\text{optim}} = \mathcal{O}(1/k)$ for $k$ iterations. Although the prox function introduces a proximal bias in the quasi-optimal Bellman operator $\mathcal{B}_\mu$, it leads to a smoothed approximation for $\mathcal{B}$. There exists a trade-off between the proximal bias and approximation error. As the increase of $\mu$, it enlarges the proximal bias but decreases the approximation error since true function space becomes more smoothed and easy for function approximation. On the other hand, a small $\mu$ leads to a small proximal bias but a relatively large approximation error.

**Theorem 4.4.** *Suppose $\widehat{\mathcal{L}}_U$ in Algorithm 1 is differentiable, but not necessarily convex, and its gradient $\nabla\widehat{\mathcal{L}}_U(\theta, \xi)$ is $M_{\mathcal{L}}$-Lipschitz and $\mathrm{Var}(\bar{\nabla}_\theta + \bar{\nabla}_\xi) \leq \sigma_0^2$. And suppose that the learning rate $\{\alpha_j\}$ are set to $\alpha_j = \min\left\{\frac{2}{M_{\mathcal{L}}}, \frac{\Lambda}{\sigma_0\sqrt{j}}\right\}$ for some $\Lambda \geq 0$ and $\varepsilon$ is sufficient small. Let $k = \widetilde{k}$ with $\mathbb{P}(\widetilde{k} = j) = \frac{\alpha_j(2 - M_{\mathcal{L}}\alpha_j)}{\sum_{j=1}^{k_\diamond}(\alpha_j(2 - M_{\mathcal{L}}\alpha_j))}$ for $j = 1, \ldots, k_\diamond$. Then, if $(\widehat{\theta}, \widehat{\xi})$ is the optimization solution and $(\theta^1, \xi^1)$ is the first step solution, we have*

$$\left\|\nabla\widehat{\mathcal{L}}_U(\widehat{\theta}, \widehat{\xi})\right\|_{L^2}^2 \leq 2M_{\mathcal{L}}\left(\widehat{\mathcal{L}}_U(\theta^1, \xi^1) - \min_{\theta, \xi}\widehat{\mathcal{L}}_U(\theta, \xi)\right)\left(\frac{M_{\mathcal{L}}}{k_\diamond} + \frac{\sigma_0}{M_{\mathcal{L}}\Lambda\sqrt{k_\diamond}}\right) + \frac{\Lambda\sigma_0 M_{\mathcal{L}}}{\sqrt{k_\diamond}},$$

Theorem 4.4 implies that the quasi-optimal learning algorithm is converges to a stationary point with a sub-linear rate $\mathcal{O}(1/\sqrt{k_\diamond})$ even if the empirical loss is non-convex. The property serves as a basis for applying non-linear function approximation with convergent guarantees. Theorem 4.4 is adapted from Corollary 2.2 in Ghadimi & Lan (2013) under a decay learning rate and a Euclidean stopping criterion. The convergence of Algorithm 1 is blessed by our unbiased stochastic gradient estimator.

## 5 RELATED WORKS

In this work, we propose a provably convergent and sample efficient off-policy optimization algorithm. Our learning algorithm is trained in a fully offline fashion, without any future online interaction with the environment. This connects our work to offline RL algorithms (Lange et al., 2012; Levine et al., 2020). Due to space limitations, we defer the discussion on offline RL in Appendix.

Algorithmically, our work is related to the entropy-regularized reinforcement learning algorithms (Rawlik et al., 2012; Haarnoja et al., 2017), but these works are fundamentally different from ours. Our formulation is motivated by constructing a proximal counterpart of the Bellman operator, which serves as a basis for the latter quasi-oracle learning algorithm. Besides, the major drawback of the existing algorithms (Lee et al., 2018b; Chow et al., 2018b; Vieillard et al., 2020) is the lack of theoretical guarantees when accompanied by function approximation. It is not clear whether the algorithm is convergent, generalizable, and consistent. In contrast, our algorithm is thoroughly examined on both theoretical and empirical fronts. Nachum et al. (2017); Chow et al. (2018b) exploit an analogous stationarity condition as in Theorem 3.3 and minimize the upper bound of the error, which is biased and encounters double sampling issue. In contrast, our work leverages the kernel embedding to bypass the double sampling issue, and is provably consistent. Unlike our algorithm, the algorithms in continuous control problems, e.g., (Haarnoja et al., 2018b; Nachum et al., 2018b; Lee et al., 2019) do not check the policy optimality, but separately model a pre-specified policy class. This may introduce an additional bias if the pre-specified policy class is misspecified.

Our approach exemplifies more recent efforts that aim to learn optimal policy with continuous actions (Lillicrap et al., 2015). One of our key innovations is to develop a policy class that can identify quasi-optimal sub-regions and the induced policy has a closed-form regarding value function. This distinguishes us from the approaches, e.g., (Silver et al., 2014; Mnih et al., 2016; Kumar et al., 2019; 2020). These methods typically require prior knowledge to determine pre-specified policy class and commonly use Gaussian family distribution, but unfortunately facing the risk from off-support bias.

Our work is also relevant to safe/risk-sensitive RL. When the risk measure is defined based on the reward, e.g., the quantile of return, it draws connections to our algorithm. Given potential application scenarios, quasi-optimal learning is also related to RL in healthcare domain and the trade-offs between safety and optimality. Tang et al. (2020) constructs set-valued policies of near-optimal actions allowing the interaction between the clinician and the decision support system. However, their method is not applicable in a fully offline setting. Fatemi et al. (2021) assesses regions of risk and identifies treatments to avoid in a safety-critical environment. Nevertheless, near-optimal regret guarantee is vacuous in their framework. Due to page limits, we provide a detailed discussion on safe and healthcare RL in Appendix.

## 6 EXPERIMENTS

In this section, we evaluate our proposed method on synthetic and real environments. We compare our method to the state-of-the-art baselines including DDPG (Lillicrap et al., 2015), SAC (Haarnoja et al., 2018a), BEAR (Kumar et al., 2019), Greedy-GQ (Ertefaie & Strawderman, 2018), V-Learning (Luckett et al., 2019). We also compete with two safe RL algorithms CQL (Kumar et al., 2020) and IQN (Dabney et al., 2018a) for a comprehensive comparison from the safety RL point of view.

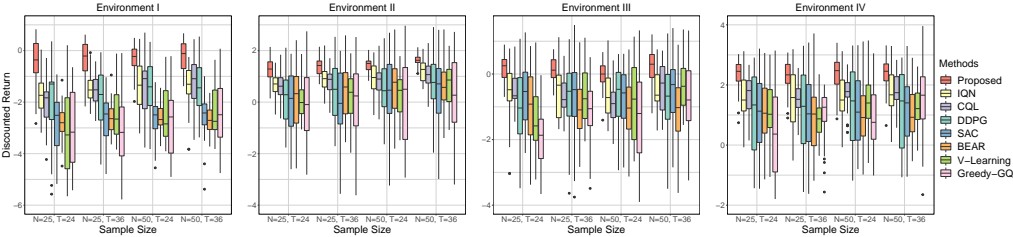

Figure 3: The boxplot of the discounted return over 50 repeated experiments.

**Synthetic Data** The four environments are simulated to mimic the real environments for continuous treatment applications. In Environment I and II, we consider a bounded action space to evaluate the potential of quasi-optimal learning for addressing off-support bias. The design of Environment III is to mimic safety-critical environment by incorporating the notion of safety into the reward function (Jia et al., 2020), i.e., the optimal dosage is unique, and a high dosage leads to excessive toxicity while a lower dosage is ineffective (Zang et al., 2014). This is helpful for examining safety performance. In Environment IV, all the methods are implemented and compared in a more complex environment. The detailed discussion on the experiment designs and settings is deferred to Section D in Appendix.

Figure 3 shows that our proposed method outperforms competing methods with a relatively small variance. This mainly benefits from identifying the quasi-optimal region, which guarantees the suggested action is near-optimal, hence improving the performance. In comparison, SAC and BEAR use a Gaussian policy and assign non-negligible positive densities to all actions, even for the non-optimal ones, which damages the model performance. Meanwhile, even though safe RL methods (i.e., CQL and IQN) show better performance and smaller variance compared with non-safe methods, their performance is still negatively affected by assigning non-zero densities to non-optimal actions. In addition, in Environment I and II with bounded action support, the competing methods are affected by an off-support bias which lowers their discounted return. In Environment III and IV, the performance gains of the proposed method are mainly from the well-recover of the quasi-optimal regions. Also, note that our algorithm achieves stable performance in small sample size settings, which is blessed by the smoothness and optimization-friendly of our algorithm. This is promising as limited data is common in medical applications. Additional experiment results including safety criterion, i.e., distribution of the discounted sum of rewards, sensitivity analysis of $\mu$ are provided in Appendix.

**Real Data: A Ohio Type 1 Diabetes Case Study** Ohio type 1 diabetes (OhioT1DM) dataset (Marling & Bunescu, 2020), which contains 2 cohorts of patients with Type-1 diabetes, each patient with 8 weeks of life-event data including health status measurements and insulin injection dosage. Clinicians are interested in adjusting insulin injection dose levels (Marling & Bunescu, 2020; Bao et al., 2011) based on patient's health status to maintain the glucose level in a certain range for safe dose suggestions. As each individual has dramatically distinctive glucose dynamics, We follow Zhu et al. (2020) to regard each patient data as an independent dataset, and the data from each day as a trajectory. The state variables are health status measurements, and the action space is a bounded insulin dose range. The glycemic index is regarded as a reward function to measure the goodness of dose suggestion. See more details of the experiment setup in Appendix. As shown in Table 1, the proposed method achieves the best performance among almost all patients. The proposed method mitigates the off-support bias in this bounded dosage space and outperforms the competing methods. This finding is consistent with the results in the synthetic data and demonstrates the potential of our method in continuous action spaces. Besides model performance, we illustrate the safety guarantee of the quasi-optimal learning with additional experiments results and analyses in Appendix.

Table 1: The discounted return for the policy improvement based on 50 repeated experiments.

| Patient ID | Proposed | DDPG | SAC | BEAR | Greedy-GQ | VL | CQL | IQN |
|---|---|---|---|---|---|---|---|---|
| 540 | **18.6 ± 0.6** | 14.1 ± 2.3 | 14.2 ± 1.2 | 13.7 ± 0.9 | 15.5 ± 2.4 | 14.1 ± 2.4 | 17.0 ± 0.9 | 18.2 ± 0.9 |
| 544 | **11.0 ± 0.7** | 7.5 ± 1.5 | 7.5 ± 2.5 | 5.9 ± 0.8 | 6.3 ± 2.9 | 8.1 ± 2.9 | 9.3 ± 1.0 | 9.8 ± 1.0 |
| 552 | 6.3 ± 0.4 | 4.8 ± 0.5 | 5.7 ± 1.0 | 3.6 ± 0.6 | 4.1 ± 1.8 | 5.2 ± 1.3 | **6.7 ± 0.7** | 6.1 ± 0.8 |
| 567 | 29.9 ± 1.5 | 30.0 ± 2.0 | 27.3 ± 2.2 | 29.6 ± 1.2 | 24.8 ± 3.8 | 20.2 ± 2.8 | **31.5 ± 1.1** | 29.8 ± **0.6** |
| 584 | **32.1 ± 0.8** | 27.0 ± 2.0 | 23.3 ± 3.2 | 26.9 ± 1.3 | 17.8 ± 3.2 | 18.7 ± 2.6 | 26.6 ± 1.3 | 27.7 ± 1.2 |
| 596 | **5.5 ± 1.1** | 4.1 ± 0.8 | 4.5 ± 0.9 | 2.7 ± 1.0 | 2.7 ± 1.8 | 3.7 ± 3.0 | 4.6 ± 0.6 | 4.7 ± **0.6** |
| 559 | **24.1 ± 1.4** | 20.1 ± 1.2 | 19.6 ± 1.2 | 19.6 ± **0.7** | 17.3 ± 1.6 | 20.6 ± 2.7 | 22.1 ± 1.3 | 22.6 ± 1.2 |
| 563 | **11.6 ± 0.6** | 8.4 ± 0.9 | 9.3 ± 0.7 | 8.4 ± 0.7 | 9.2 ± 1.5 | 8.8 ± 1.9 | 9.4 ± 0.7 | 9.9 ± 0.8 |
| 570 | 25.0 ± **0.8** | 24.5 ± 1.4 | **26.1 ± 0.8** | 25.8 ± 0.8 | 22.8 ± 1.6 | 22.6 ± 1.5 | 25.8 ± 0.9 | 25.9 ± 0.8 |
| 575 | **15.5 ± 1.0** | 10.4 ± 1.3 | 8.8 ± 1.4 | 10.2 ± 1.0 | 5.7 ± 2.8 | 8.5 ± 2.3 | 12.6 ± **0.9** | 12.7 ± 1.2 |
| 588 | **18.6 ± 0.7** | 14.2 ± 1.3 | 13.5 ± 1.5 | 12.0 ± 0.9 | 10.0 ± 3.1 | 8.6 ± 2.3 | 15.7 ± 0.8 | 15.9 ± 1.3 |
| 591 | **15.4 ± 1.0** | 12.3 ± 0.6 | 11.9 ± **0.6** | 12.8 ± 0.7 | 10.7 ± 1.7 | 10.5 ± 2.6 | 14.9 ± 0.6 | 15.2 ± 0.7 |

## 7    CONCLUSIONS

We introduce a novel quasi-oracle learning algorithm for continuous action allocations, which is particularly useful in determining the dose level when developing medical treatment regimes. The quasi-optimal learning algorithm is provably convergent in off-policy cases, and a PAC bound is provided to analyze its sample complexity. The promising results arise some interesting directions for future works, including extending the framework to online settings interacting with environments.

## 8 ACKNOWLEDGMENTS

The authors are grateful to the anonymous reviewers and area chair for valuable comments and suggestions. Ruoqing Zhu's research is partially supported by a grant from the National Science Foundation DMS-2210657.

## 9 REPRODUCIBILITY STATEMENT

We include the reproducible code for all the experiments and the guideline for access to the Ohio Type I Diabetes dataset in GitHub link `https://github.com/liyuhan529/Quasi-optimal-Learning`. The experiment details are provided in Appendix for reproducible purpose. All proofs of main theorems and addition theorems are included in Appendix.

## 10 ETHICS STATEMENT

The proposed method aims at finding optimal policy in continuous action space, with a special focus on medical applications. We believe the proposed work has potential applications in sequential treatment dose suggestion, e.g., managing diabetes through insulin injection. We admit that the proposed method needs additional validation experiments in controlled settings for practical use in medical applications to avoid abundant risks. The proposed algorithm should not be used as a stand-alone tool nor as a replacement of human experts.

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

# "Supplementary Materials for Quasi-optimal Reinforcement Learning with Continuous Actions"

# Appendices

## A  ADDITIONAL RELATED WORKS

We discuss additional related works in this section.

**Safe RL**   Safe Reinforcement Learning (safe-RL) aims at finding an optimal policy while ensuring safety (Garcıa & Fernández, 2015). In the safe-RL framework, the definition of safety and its guarantee varies based on the specific purpose of learning tasks. In our view, there are three mainstream works for safe RL.

- Safe Exploration: ensuring safe action allocations in the exploration process by incorporating prior knowledge, which often exists in online RL settings (Pham et al., 2018).
- Safety Constraints: finding an optimal policy that satisfies external user-specified safe constraints (Chow et al., 2018a; Gu et al., 2022).
- Risk-sensitivity and Conservatism: finding a policy maximizing the infinite-horizon cumulative discounted reward while incorporating the notion of risk (Morimura et al., 2010; Mavrin et al., 2019), e.g., value at risk (quantile), percentile performance, chance, the variance of return.

In medical applications, specifying explicit constraints is typically hard to realize in practice (Vincent, 2014). Alternatively, the notion of safety is usually incorporated in the design of reward functions, where high-risk actions lead to significantly low reward (Raghu et al., 2017; Jia et al., 2020).

Based on these, our quasi-optimal learning is closely related to the risk-sensitive RL framework, which aims to control value at risk to ensure safety. For example, maintaining the discounted return above a certain threshold (Tamar et al., 2015), reducing the variability of performance by avoiding extremely low performance (Ma et al., 2020), or target to maximize the robust performance criterion, e.g., quantile of the discounted return (Dabney et al., 2018b). Commonly used algorithms in risk-sensitive RL include conservative Q-learning (CQL; Kumar et al. (2020)) and implicit quantile network (IQN; (Dabney et al., 2018a)). CQL learns a conservative Q-function such that the expected value of a policy under this Q-function lower-bounds its true value and thus avoids selecting high-risk actions with over-estimation action value. IQN models the full quantile function for the state-action return distribution and yields risk-sensitive policies. For a more comprehensive empirical study, we compare the proposed algorithm with the aforementioned two safe RL baselines, conduct additional numerical experiments and analyze the results from the safety point of view.

**RL in healthcare**   Reinforcement learning has a wide variety of applications in healthcare (Yu et al., 2021). Some of the recent works aim to solve safety issues when applying RL to healthcare domains. Tang et al. (2020) considers identifying set-valued policies with near-optimal actions, which allows incorporating expert knowledge from clinicians to assist in decision making. As the same rationale in our proposed quasi-optimal region, Tang et al. (2020) also utilizes the value function to threshold a near-optimal action set. However, this method is only developed on discrete action space, and it is still not directly applicable in fully offline settings. Fatemi et al. (2021) considers identifying high-risk states in data-constrained offline settings by training two separate Q functions that model the probability of negative outcomes and positive outcomes respectively. They target to identify treatments proportional to their chance of leading to dead-ends, and attain safety by excluding these treatments from consideration. However, as they aim to identify possible "dead-ends" of a state space and treatments, there exists a trade-off between safety and optimality. In particular, it still has a gap for optimal treatment allocations.

Other interesting works in RL for healthcare including Henry et al. (2015); Komorowski et al. (2018) adopt RL algorithms for sepsis treatment recommendations, Jia et al. (2020) redefine the state variables and reward function to reflect practical safety concerns in sepsis treatments. We refer readers to (Yu et al., 2021) for a more comprehensive review.

**Offline RL**   The domain approaches of offline RL include fitted Q-iteration (FQI; Ernst et al. (2005); Riedmiller (2005); Munos & Szepesvári (2008); Szepesvári (2010) ), fitted policy iteration (Antos et al., 2007; Lagoudakis & Parr, 2003; Scherrer et al., 2012), Bellman Residual Minimization (BRM; Antos et al. (2008); Hoffman et al. (2011); Farahmand et al. (2016); Dai et al. (2018); Chen & Jiang (2019); Xie & Jiang (2020), gradient Q-learning (Maei et al., 2010; Ertefaie & Strawderman, 2018), and Advantage learning (Murphy, 2003; Shi et al., 2018; 2022). We refer the reader to Levine et al. (2020) for more comprehensive discussions on the topics of the offline RL.

In the aforementioned mainstreams of works, ours is closely related to the Bellman Residual Minimization. They learn the value function by solving a nested optimization problem, where the function space used for the inner and outer optimization must be the same. From the perspective of the couple optimization, their inner optimization plays a similar role as the inner maximization of the min-max framework. In addition to the fundamental difference in derivation, our min-max optimization can be reduced to a single minimization problem aided by the kernel representation, while they have to solve an unstable minimax optimization problem. Most importantly, our quasi-optimal learning framework provides a practical way to learn a reliable policy in continuous action space via quasi-optimal region identifications. to the best of our knowledge, no existing RL algorithms can achieve this.

## B TECHNICAL PROOFS

### B.1 PROOFS ON CONSTRUCTING QUASI-OPTIMAL BELLMAN OPERATOR

#### B.1.1 PROOF OF THEOREM S.1

**Theorem S.1.** *Assume the induced policy has density function $\pi_\mu^*(a|s) \leq \mathbf{C}$ for all $a, s$, where $\mathbf{C}$ is a given constant. Then the proximal Bellman operator $\mathcal{B}_\mu$ in equation (2) has a closed form equivalent:*

$$
\mathcal{B}_\mu V_\mu^*(s) = \mu \left\{ 1 - \int_{a \in \mathcal{W}_{s,1}} \left[ \left( \frac{\int_{a \in \mathcal{W}_{s,1}} Q_\mu^*(s,a)da}{2\mu\sigma(\mathcal{W}_{s,1})} - \frac{1}{\sigma(\mathcal{W}_{s,1})} \right)^2 - \left( \frac{Q_\mu^*(s,a)}{2\mu} \right)^2 \right] da \right\}
$$
$$
+ \frac{\mathbf{C}\sigma(\mathcal{W}_{s,1}) \int_{a \in \mathcal{W}_{s,2}} Q_\mu^*(s,a)da - \mathbf{C}\sigma(\mathcal{W}_{s,2}) \int_{a \in \mathcal{W}_{s,1}} Q_\mu^*(s,a)da}{2\sigma(\mathcal{W}_{s,1})} - \frac{\mu \mathbf{C}^2 \sigma(\mathcal{W}_{s,2})(\sigma(\mathcal{W}_{s,2}) + \sigma(\mathcal{W}_{s,1}))}{\sigma(\mathcal{W}_{s,1})},
$$

$$(13)$$

*where $\mathcal{W}_{s,1}$ refers to the set $\{a \in \mathcal{A} : \mathbf{C} > \pi_\mu^*(a|s) > 0\}$, $\mathcal{W}_{s,2}$ refers to the set $\{a \in \mathcal{A} : \pi_\mu^*(a|s) = \mathbf{C}\}$.*

**Proof:** The proof is mainly to check the KKT conditions of the maximization. The Lagrangian function of the RHS of (2) can be expressed as follows:

$$
L(\pi, \tilde{\eta}, \varpi_1, \varpi_2) = \mathbb{E}_{a \sim \pi(\cdot|s)} \left[ Q_\mu(s,a) + \mu \text{prox}(\pi(a|s)) \right] - \tilde{\eta}(s) \left( \int_{a \in \mathcal{A}} \pi(a|s)da - 1 \right)
$$
$$
+ \varpi_1(s,a)\pi(a|s) - \varpi_2(s,a)(\pi(a|s) - \mathbf{C}).
$$

The following KKT conditions are necessary for the maximizer $\pi_\mu^*$ in the equation:

- Primal: $\int_{a \in \mathcal{A}} \pi_\mu^*(a|s)da - 1 = 0$, $-\pi_\mu^*(a|s) \leq 0$, $\pi_\mu^*(a|s) \leq \mathbf{C}$.
- Duality: $\varpi_1(s,a) \geq 0$, $\varpi_2(s,a) \geq 0$.
- Complementary slackness: $\varpi_1(s,a)\pi_\mu^*(a|s) = 0$, $\varpi_2(s,a)(\pi_\mu^*(a|s) - \mathbf{C}) = 0$.
- Stationarity: $Q_\mu^*(s,a) + \mu(1 - 2\pi_\mu^*(a|s)) - \tilde{\eta}(s) + \varpi_1(s,a) - \varpi_2(s,a) = 0$.

We can obtain the equation for $\pi_\mu(a|s)$ from the stationary condition such that

$$
\pi_\mu^*(a|s) = \frac{1}{2} - \frac{1}{2\mu}[\tilde{\eta}(s) - Q_\mu^*(s,a) - \varpi_1(s,a) + \varpi_2(s,a)].
$$

Combined with complementary slackness condition,

- If $\pi_\mu^*(a|s) = 0$, then $\varpi_1(s,a) \geq 0$, $\varpi_2(s,a) = 0$, thus $Q_\mu^*(s,a) \leq \tilde{\eta}(s) - \mu$.
- If $\mathbf{C} > \pi_\mu^*(a|s) > 0$, then $\varpi_1(s,a) = \varpi_2(s,a) = 0$, thus $\tilde{\eta}(s) - \mu + 2\mu\mathbf{C} > Q_\mu^*(s,a) > \tilde{\eta}(s) - \mu$.
- If $\pi_\mu^*(a|s) = \mathbf{C}$, then $\varpi_1(s,a) = 0$, $\varpi_2(s,a) \leq 0$, thus $Q_\mu^*(s,a) \geq \tilde{\eta}(s) - \mu + 2\mu\mathbf{C}$.

Therefore, $\pi_\mu^*(s,a)$ can be expressed as:

$$
\pi_\mu^*(a|s) = \begin{cases} 0 & \text{if } Q_\mu^*(s,a) \leq \tilde{\eta}(s) - \mu \\ \frac{1}{2} - \frac{1}{2\mu}\left(\tilde{\eta}(s) - Q_\mu^*(s,a)\right) & \text{if } \tilde{\eta}(s) - \mu + 2\mu\mathbf{C} > Q_\mu^*(s,a) > \tilde{\eta}(s) - \mu \\ \mathbf{C} & \text{if } Q_\mu^*(s,a) \geq \tilde{\eta}(s) - \mu + 2\mu\mathbf{C} \end{cases} \quad (14)
$$

Meanwhile, notice that $\int_{a\in\mathcal{A}} \pi_\mu^*(s,a) = 1$, we can show that $\tilde{\eta}(s)$ has a closed form:

$$\tilde{\eta}(s) = \mu + \frac{\int_{a\in\mathcal{W}_{s,1}} Q_\mu^*(s,a)da - 2\mu + 2\mu\mathbf{C}\sigma(\mathcal{W}_{s,2})}{\sigma(\mathcal{W}_{s,1})},$$

where $\mathcal{W}_{s,1}$ refers to the set $\{a\in\mathcal{A} : C > \pi_\mu^*(a|s) > 0\}$, $\mathcal{W}_{s,2}$ refers to the set $\{a\in\mathcal{A} : \pi_\mu^*(a|s) = C\}$, and $\sigma(\mathcal{W}_{s,1})$, $\sigma(\mathcal{W}_{s,1})$ refers to the interval length of the corresponding set. We take $\tilde{\eta}(s)$ back to (14), we then have

$$\pi_\mu^*(a|s) = \begin{cases} 0 & \text{if } Q_\mu^*(s,a) \le \tilde{\eta}(s) - \mu \\ \frac{Q_\mu^*(s,a)}{2\mu} - \frac{\int_{a\in\mathcal{W}_{s,1}} Q_\mu^*(s,a)da}{2\mu\sigma(\mathcal{W}_{s,1})} + \frac{1-\mathbf{C}\sigma(\mathcal{W}_{s,2})}{\sigma(\mathcal{W}_{s,1})} & \text{if } \tilde{\eta}(s) - \mu + 2\mu\mathbf{C} > Q_\mu^*(s,a) > \tilde{\eta}(s) - \mu \\ \mathbf{C} & \text{if } Q_\mu^*(s,a) \ge \tilde{\eta}(s) - \mu + 2\mu\mathbf{C} \end{cases} \tag{15}$$

We finally plug in the closed form of $\pi_\mu^*(a|s)$ to (2), by some algebra, we have

$$\mathcal{B}_\mu V_\mu^*(s) = \mu\left\{1 - \int_{a\in\mathcal{W}_{s,1}}\left[\left(\frac{\int_{a\in\mathcal{W}_{s,1}} Q_\mu^*(s,a)da}{2\mu\sigma(\mathcal{W}_{s,1})} - \frac{1}{\sigma(\mathcal{W}_{s,1})}\right)^2 - \left(\frac{Q_\mu^*(s,a)}{2\mu}\right)^2\right]da\right\}$$

$$+ \frac{\mathbf{C}\sigma(\mathcal{W}_{s,1})\int_{a\in\mathcal{W}_{s,2}} Q_\mu^*(s,a)da - \mathbf{C}\sigma(\mathcal{W}_{s,2})\int_{a\in\mathcal{W}_{s,1}} Q_\mu^*(s,a)da}{2\sigma(\mathcal{W}_{s,1})} - \frac{\mu\mathbf{C}^2\sigma(\mathcal{W}_{s,2})(\sigma(\mathcal{W}_{s,2}) + \sigma(\mathcal{W}_{s,1}))}{\sigma(\mathcal{W}_{s,1})}.$$

### B.1.2 PROOF OF COROLLARY S.1

**Corollary S.1.** *When $\sigma(\mathcal{W}_{s,2}) = 0$, we denote $\mathcal{W}_1$ as $\mathcal{W}$, the closed form in* (13) *can be simplified as*

$$\mathcal{B}_\mu V_\mu^*(s) = \mu - \frac{1}{4\mu}\left(\frac{(\int_{a'\in\mathcal{W}_s} Q_\mu^*(s,a')da' - 2\mu)^2}{\sigma(\mathcal{W}_s)} - \int_{a\in\mathcal{W}_s} Q_\mu^{*2}(s,a)da\right).$$

**Proof:** We plug in $\sigma(\mathcal{W}_{s,2}) = 0$ to (13), then could obtain the result.

### B.1.3 PROOF OF THEOREM 3.1

**Proof of Theorem 3.1:** For any generic value function $V(s)$ and the corresponding generic Q-function $Q(s,a)$, we first build the lower bound:

$$\mathcal{B}_\mu V(s) = \max_{\pi\in\Delta_{\text{convex}}(\mathcal{A})} \mathbb{E}_{a\sim\pi(\cdot|s)}[Q(s,a) + \mu(1 - \pi(a|s))]$$

$$\ge \max_{\pi\in\Delta_{\text{convex}}(\mathcal{A})} \mathbb{E}_{a\sim\pi(\cdot|s)}[Q(s,a) + \mu - \mu\mathbf{C}]$$

$$= \mathcal{B}V(s) + \mu(1 - \mathbf{C}).$$

For the upper bound:

$$\mathcal{B}_\mu V(s) = \max_{\pi\in\Delta_{\text{convex}}(\mathcal{A})(\mathcal{A})} \mathbb{E}_{a\sim\pi(\cdot|s)}[Q(s,a) + \mu(1 - \pi(a|s))]$$

$$\le \max_{\pi\in\Delta_{\text{convex}}(\mathcal{A})(A)} \mathbb{E}_{a\sim\pi(\cdot|s)}[Q(s,a) + \mu]$$

$$= \mathcal{B}V(s) + \mu.$$

Therefore, we have $\mathcal{B}_\mu V(s) - \mathcal{B}V(s) \in [\mu(1 - \mathbf{C}), \mu]$.

### B.1.4 PROOF OF THEOREM 3.2

**Proof of Theorem 3.2:** Suppose $Q_\mu^*(s,a) = -\alpha_1(s)a^2 + \alpha_2(s)a + \alpha_3(s)$ with $\alpha_1(s) > 0$. We assume the density won't reach its boundary value $\mathbf{C}$ for this theorem, and we proceed by simplifying $\alpha_i(s)$ as $\alpha_i$ for $i = 1, 2, 3$. By Equation (4), we have

$$\pi_\mu^*(a|s) = \left\{\frac{Q_\mu^*(s,a)}{2\mu} - \frac{\int_{a\in\mathcal{W}_s} Q_\mu^*(s,a)da}{2\mu\sigma(\mathcal{W}_s)} + \frac{1}{\sigma(\mathcal{W}_s)}\right\}^+.$$

We first try to find the support set of $\pi_\mu^*(a|s)$. Since $Q_\mu^*$ takes the maximum value at $y = \frac{\alpha_2}{2\alpha_1}$, by the symmetric property of quadratic function, the support set should be of the form $\mathcal{W}_s = [y-l, y+l](l > 0)$. Additionally, the boundary point of the support set should be the solution of

$$\frac{Q_\mu^*(s,a)}{2\mu} - \frac{\int_{a\in\mathcal{W}_s} Q_\mu^*(s,a)da}{2\mu\sigma(\mathcal{W}_s)} + \frac{1}{\sigma(\mathcal{W}_s)} = 0,$$

with respect to $a$. Thus, we can find the boundary point of the support set by solving the equation with respect to $l$:

$$-\alpha_1(y \pm l)^2 + \alpha_2(y \pm l) + \alpha_3 = \frac{1}{2l}\int_{y-l}^{y+l}(-\alpha_1 a^2 + \alpha_2 a + \alpha_3)da - \frac{\mu}{l}.$$

It turns out that $l = \frac{(12\alpha_1^2\mu)^{\frac{1}{3}}}{2\alpha_1}$. Thus, the support set has the closed-form

$$\mathcal{W}_s = \left\{ a : a \in \left[ \frac{\alpha_2 - (12\alpha_1^2\mu)^{\frac{1}{3}}}{2\alpha_1}, \frac{\alpha_2 + (12\alpha_1^2\mu)^{\frac{1}{3}}}{2\alpha_1} \right] \right\}.$$

Therefore $\sigma(\mathcal{W}_s) = \frac{(12\alpha_1^2\mu)^{\frac{1}{3}}}{\alpha_1}$, and

$$\frac{\int_{a\in\mathcal{W}_s} Q_\mu^*(s,a)da}{2\mu\sigma(\mathcal{W}_s)} = -\frac{(12\alpha_1^2\mu)^{\frac{2}{3}} - 3\alpha_2^2}{24\mu\alpha_1} + \frac{\alpha_3}{2\mu}.$$

We plug in the result to the closed form of $\pi_\mu^*(a|s)$, and obtain the probability density function

$$\pi_\mu^*(a|s) = \left\{ \frac{\alpha_1}{2\mu}(a + \frac{\alpha_2}{2\alpha_1})^2 - \frac{3}{2}(\frac{\alpha_1}{12\mu})^{\frac{1}{3}} \right\}^+.$$

It is clear that the resulting distribution of $\pi_\mu^*(a|s)$ is of the exact form of $q$-Gaussian distribution with $q = 0, \beta = \frac{\alpha_1}{2\mu}$ and centered at $\frac{\alpha_2}{2\alpha_1}$.

### B.2 PROOFS ON QUASI-OPTIMAL STAIONARITY EQUATION

#### B.2.1 PROOF OF THEOREM 3.3

**Proof of Theorem 3.3:** By the stationary condition from Theorem S.1 we have

$$Q_\mu^*(s,a) + \mu(1 - 2\pi_\mu^*(a|s)) - \tilde{\eta}(s) + \varpi_1(s,a) - \varpi_2(s,a) = 0,$$

therefore, by the definition of $Q_\mu^*(s,a)$, we have

$$\mathbb{E}_{S^{t+1}|s,a}[R(S^{t+1},s,a)] + \gamma\mathbb{E}_{S^{t+1}|s,a}[V_\mu^*(S^{t+1})] + \mu(1 - 2\pi_\mu^*(a|s)) - \tilde{\eta}(s) + \varpi_1(s,a) - \varpi_2(s,a) = 0. \tag{16}$$

Notice that $\mathbb{E}_{S^{t+1}|s,a}[R(S^{t+1},s,a)] = r(s,a)$, and we take expectation with respect to $a$ following the policy distribution $\pi_\mu^*(a|s)$ from both sides of (16),

$$0 = \mathbb{E}_{a\sim\pi_\mu^*(a|s)}\left[ r(s,a) + \gamma\mathbb{E}_{S^{t+1}|s,a}[V_\mu^*(S^{t+1})] + \mu(1 - 2\pi_\mu^*(a|s)) - \tilde{\eta}(s) + \varpi_1(s,a) - \varpi_2(s,a) \right],$$

$$0 = \int_{a\in\mathcal{A}} \pi_\mu^*(a|s)\left[ r(s,a) + \gamma\mathbb{E}_{S^{t+1}|s,a}[V_\mu^*(S^{t+1})] + \mu(1 - 2\pi_\mu^*(a|s)) - \tilde{\eta}(s) + \varpi_1(s,a) - \varpi_2(s,a) \right]da.$$

According to the proximal Bellman optimality equation $\mathcal{B}_\mu V_\mu^*(s) = V_\mu^*(s)$, where $V_\mu^*(s)$ is the fixed point of $\mathcal{B}_\mu$. With the explicit definition of $V_\mu^*$, we observe that

$$0 = \int_{a\in\mathcal{A}} \pi_\mu^*(a|s)\left[ r(s,a) + \gamma\mathbb{E}_{S^{t+1}|s,a}[V_\mu^*(S^{t+1})] + \mu(1 - \pi_\mu^*(a|s)) \right]da - \int_{a\in\mathcal{A}} \mu\pi_\mu^{*2}(a|s)da$$

$$- \int_{a\in\mathcal{A}} \pi_\mu^*(a|s)\tilde{\eta}(s)da + \int_{a\in\mathcal{A}} \pi_\mu^*(a|s)\varpi_1(s,a)da - \int_{a\in\mathcal{A}} \pi_\mu^*(a|s)\varpi_2(s,a)da$$

$$= V_\mu^*(s) - \int_{a\in\mathcal{A}} \mu\pi_\mu^{*2}(a|s)da - \int_{a\in\mathcal{A}} \pi_\mu^*(a|s)\tilde{\eta}(s)da + \int_{a\in\mathcal{A}} \pi_\mu^*(a|s)\varpi_1(s,a)da$$

$$- \int_{a\in\mathcal{A}} \pi_\mu^*(a|s)\varpi_2(s,a)da$$

Meanwhile $\int_{a\in\mathcal{A}} \pi_\mu^*(a|s)\tilde{\eta}(s)da = \tilde{\eta}(s)\int_{a\in\mathcal{A}} \pi_\mu^*(a|s)da = \tilde{\eta}(s)$ by the property of density, $\int_{a\in\mathcal{A}} \pi_\mu^*(a|s)\varpi_1(s,a)da = 0$, and $\int_{a\in\mathcal{A}} \pi_\mu^*(a|s)\varpi_2(s,a)da = \mathbf{C}\int_{a\in\mathcal{A}} \varpi_2(s,a)da$ by complete slackness, we further have

$$V_\mu^*(s) - \mu\int_{a\in\mathcal{A}} \pi_\mu^{*^2}(a|s)da - \tilde{\eta}(s) - \mathbf{C}\int_{a\in\mathcal{A}} \varpi_2(s,a)da = 0.$$

Since $0 \leq \pi_\mu^*(a|s) \leq \mathbf{C}$, thus $\mu\int_{a\in\mathcal{A}} \pi_\mu^{*^2}(a|s)da = \mu\mathbb{E}\pi_\mu^*(a|s) \in [0, \mathbf{C}]$. Therefore,

$$\eta(s) := \tilde{\eta}(s) - V_\mu^*(s) \in [-\mu\mathbf{C} - \mathbf{C}\int_{a\in\mathcal{A}} \varpi_2(s,a)da, -\mathbf{C}\int_{a\in\mathcal{A}} \varpi_2(s,a)da].$$

The stationary condition can be reformulated as

$$\mathbb{E}_{S^{t+1}|s,a}\left[R(S^{t+1},s,a)+\gamma V_\mu^*(S^{t+1})\right]-\mu\text{prox}^\circ(\pi_\mu^*(a|s))-\eta(s)+\varpi_1(s,a)-\varpi_2(s,a)-V_\mu^*(s) = 0. \tag{17}$$

Obviously, $(\pi_\mu^*, V_\mu^*)$ is a solution for the above equation for some $\eta(s), \varpi_1(s,a)$, and $\varpi_2(s,a)$, such that

$$\varpi_1(s,a) \geq 0, \varpi_2(s,a) \geq 0, \varpi_1(s,a)\cdot\pi_\mu(a|s) = 0, \varpi_2(s,a)\cdot(\mathbf{C}-\pi_\mu(a|s)) = 0$$

$$\text{and } \eta(s) \in \left[-\mu\mathbf{C} - \mathbf{C}\int_{a\in\mathcal{A}} \varpi_2(s,a)da, -\mathbf{C}\int_{a\in\mathcal{A}} \varpi_2(s,a)da\right].$$

When $\sigma(\mathcal{W}_{s,2}) = 0$, we have $\varpi_2(s,a) = 0$.

Plugging in to equation (17), and denote $\mathcal{W}_{s,1}$ as $\mathcal{W}_s$, we have the exact form of (9).

### B.3 Proofs on Kernel Representation

#### B.3.1 Proof of Theorem S.2

**Theorem S.2.** *We define the optimal weight function as $u^* = \arg\max_{u\in L^2(C_0)} \mathcal{L}^2(V_\mu, \pi_\mu, \eta, \varpi, u)$. Let $\mathbb{C}(\mathcal{S}\times\mathcal{A})$ be all continuous functions on $\mathcal{S}\times\mathcal{A}$. For any $(s,a) \in \mathcal{S}\times\mathcal{A}$ and $s' \in \mathcal{S}$, the optimal weight function $u^*(S^t, A^t) \in L^2(C_0)\cap\mathbb{C}(\mathcal{S}\times\mathcal{A})$ and is unique if the reward function $R(s', s, a)$ and the transition kernel $\mathbf{P}(s'|s,a)$ are continuous over $(s,a)$.*

**Proof:** Denote $\tilde{u} = \mathcal{G}_{V_\mu,\pi_\mu}(S^t, A^t, S^{t+1}) - \eta(S^t) + \varpi(S^t, A^t) - V_\mu(S^t)$. It follows from the definition of $\mathcal{L}^2(V_\mu, \pi_\mu, \eta, \varpi, u)$, we have that

$$\min_{V_\mu,\pi_\mu,\eta,\varpi}\max_u \quad \mathcal{L}^2(V_\mu, \pi_\mu, \eta, \varpi, u)$$

$$= \min_{V_\mu,\pi_\mu,\eta,\varpi}\max_u \left(\mathbb{E}_{S^t,A^t,S^{t+1}}\left[(\mathcal{G}_{V_\mu,\pi_\mu}(S^t, A^t, S^{t+1}) - \eta(S^t) + \varpi(S^t, A^t)) - V_\mu(S^t))u(S^t, A^t)\right]\right)^2$$

$$= \min_{V_\mu,\pi_\mu,\eta,\varpi}\max_u \left\langle(\mathcal{G}_{V_\mu,\pi_\mu}(S^t, A^t, S^{t+1}) - \eta(S^t) + \varpi(S^t, A^t)) - V_\mu(S^t)), u(S^t, A^t)\right\rangle^2$$

$$= \min_{V_\mu,\pi_\mu,\eta,\varpi} \left\langle(\mathcal{G}_{V_\mu,\pi_\mu}(S^t, A^t, S^{t+1}) - \eta(S^t) + \varpi(S^t, A^t)) - V_\mu(S^t)), \frac{\sqrt{C_0}\tilde{u}}{\|\tilde{u}\|_{L^2}})\right\rangle^2$$

$$= \min_{V_\mu,\pi_\mu,\eta,\varpi} \left\langle(\mathcal{G}_{V_\mu,\pi_\mu}(S^t, A^t, S^{t+1}) - \eta(S^t) + \varpi(S^t, A^t)) - V_\mu(S^t)), (\mathcal{G}_{V_\mu,\pi_\mu}(S^t, A^t, S^{t+1}) - \eta(S^t)+\right.$$

$$\left.\varpi(S^t, A^t)) - V_\mu(S^t))\right\rangle \cdot \left\langle\frac{C_0\tilde{u}}{\|\tilde{u}\|_{L^2}}, \frac{\tilde{u}}{\|\tilde{u}\|_{L^2}}\right\rangle$$

$$= \min_{V_\mu,\pi_\mu,\eta,\varpi} \left\langle(\mathcal{G}_{V_\mu,\pi_\mu}(S^t, A^t, S^{t+1}) - \eta(S^t) + \varpi(S^t, A^t)) - V_\mu(S^t)), (\mathcal{G}_{V_\mu,\pi_\mu}(S^t, A^t, S^{t+1}) - \eta(S^t)+\right.$$

$$\left.\varpi(S^t, A^t)) - V_\mu(S^t))\right\rangle$$

$$= \min_{V_\mu,\pi_\mu,\eta,\varpi} \mathbb{E}_{S^t,A^t}\left[\sqrt{C_0}\left(\mathcal{G}_{V_\mu,\pi_\mu}(S^t, A^t, S^{t+1}) - \eta(S^t) + \varpi(S^t, A^t)) - V_\mu(S^t)\right)\right]^2,$$

where the third equality is obtained by maximization condition of the inner product between $u$ and $\mathcal{G}_{V_\mu,\pi_\mu}(S^t, A^t, S^{t+1}) - \eta(S^t) + \varpi(S^t, A^t) - V_\mu(S^t)$ is that the two terms should have the same direction; the fourth equality is obtained by the equality condition of the Cauchy-Schwartz inequality.

Such finding indicates that there exists a closed form solution of the the optimal weight function $u^*$, such that

$$u^*(s, a) = \mathcal{G}_{V_\mu^*, \pi_\mu^*}(s, a, s') - \eta(s) + \varpi(s, a) - V_\mu^*(s),$$

which is equal to $\widetilde{u}$ when $(V_\mu, \pi_\mu) = (V_\mu^*, \pi_\mu^*)$.

Notice that for a given $\mu$, $\mathcal{W}_s$ is fully determined by $Q_\mu^*(s, a)$, thus by Equation (3),(4), we have that $\pi_\mu^*(a|s), V_\mu^*(s)$ is continuous over $Q_\mu^*(s, a)$. Additionally, by the complete slackness and stationary condition in Theorem S.1, we have

$$- \eta(s) + \varpi(s, a) = -Q_\mu^*(s, a) - \mu + V_\mu^*(s), \qquad \text{if } \varpi(s, a) \neq 0;$$
$$- \eta(s) = -Q_\mu^*(s, a) - \mu + 2\mu\pi_\mu^*(a|s) + V_\mu^*(s), \quad \text{if } \varpi(s, a) = 0.$$

Since $V_\mu^*, \pi_\mu^*$ can be represented by functions of $Q_\mu^*(s, a)$, the Lagrange multipliers $-\eta(s) + \varpi(s, a)$ can also be represented by a function of $Q_\mu^*(s, a)$, and is also continuous over $Q_\mu^*(s, a)$.

As $\pi_\mu^*(a|s), V_\mu^*(s), -\eta(s) + \varpi(s, a)$ are all continuous over $Q_\mu^*(s, a)$, we only need to prove that $Q_\mu^*(s, a)$ is continuous over $(s, a)$. By the stationarity equation in Theorem 3.3, $\mathbb{E}_{s'|s,a}[R(s', s, a)] = g(Q_\mu^*(s, a))$. Since the reward function $R(s', s, a)$ and the transition kernel $\mathbf{P}(s'|s, a)$ are continuous over $(s, a)$ by assumption, $Q_\mu^*(s, a)$ is continuous for any $(s, a)$ as $\mathbb{E}_{s'|s,a}[R(s', s, a)]$ is continuous for any $(s, a)$. Therefore, the optimal weight function $u^*(s, a)$ is continuous over any arbitrary state-action pair $(s, a)$.

### B.3.2 PROOF OF THEOREM S.3

**Theorem S.3.** *Suppose $u^* \in \mathcal{H}_\mathcal{K}^{C_0}$ is reproduced by a universal kernel $K(\cdot, \cdot)$, then the minimax optimizer* (10) *can be decoupled to a single-stage minimization problem as*

$$\min_{V_\mu, \pi_\mu, \eta, \varpi} \mathcal{L}_U = \mathbb{E}_{S^t, \widetilde{S}^t, A^t, \widetilde{A}^t, S^{t+1}, \widetilde{S}^{t+1}} \left[ \left( \mathcal{G}_{V_\mu, \pi_\mu}\left(S^t, A^t, S^{t+1}\right) - \eta\left(S^t\right) + \varpi\left(A^t \mid S^t\right) - V_\mu\left(S^t\right) \right) \right.$$
$$\left. \cdot C_0 K\left(S^t, A^t; \widetilde{S}^t, \widetilde{A}^t\right) \left( \mathcal{G}_{V_\mu, \pi_\mu}(\widetilde{S}^t, \widetilde{A}^t, \widetilde{S}^{t+1}) - \eta(\widetilde{S}^t) + \varpi(\widetilde{A}^t \mid \widetilde{S}^t) - V_\mu(\widetilde{S}^t) \right) \right],$$

*where $(\widetilde{S}^t, \widetilde{A}^t, \widetilde{S}^{t+1})$ is an independent copy of the transition pair $(S^t, A^t, S^{t+1})$.*

**Proof:** Let $\tilde{u} = \mathbb{E}_{S^t, A^t}\left[ \left( \mathcal{G}_{V_\mu, \pi_m u}(S^t, A^t, S^{t+1}) - \eta(S^t) + \varpi(A^t|S^t) - V_\mu(S^t) \right) K(\cdot, \{S^t, A^t\}) \right]$., and define the inner product $\langle \cdot, \cdot \rangle_{\mathcal{H}_\text{RKHS}}$ in $\mathcal{H}_\mathcal{K}^{C_0}$. It follows from the definition of $\mathcal{L}(V_\mu, \pi_\mu, \eta, \varpi, u)$ and kernel reproducing property we have,

$$\min_{V_\mu, \pi_\mu, \eta, \varpi} \max_u \mathcal{L}^2\left(V_\mu, \pi_\mu, \eta, \varpi, u\right)$$

$$= \min_{V_\mu, \pi_\mu, \eta, \varpi} \max_u \left( \mathbb{E}_{S^t, A^t}\left[ \left( \mathcal{G}_{V_\mu, \pi_\mu}\left(S^t, A^t, S^{t+1}\right) - \eta\left(S^t\right) + \varpi\left(A^t \mid S^t\right) - V_\mu\left(S^t\right) \right) u\left(S^t, A^t\right) \right] \right)^2$$

$$= \min_{V_\mu, \pi_\mu, \eta, \varpi} \max_u \left( \mathbb{E}_{S^t, A^t}\left[ \left\langle \left( \mathcal{G}_{V_\mu, \pi_\mu}\left(S^t, A^t, S^{t+1}\right) - \eta\left(S^t\right) + \varpi\left(A^t \mid S^t\right) - V_\mu\left(S^t\right) \right) \cdot \right. \right. \right.$$
$$\left. \left. \left. K\left(\cdot\,; S^t, A^t\right), u\left(S^t, A^t\right) \right\rangle_{\mathcal{H}_\text{RKHS}} \right] \right)^2$$

$$= \min_{V_\mu, \pi_\mu, \eta, \varpi} \max_u \left\langle \mathbb{E}_{S^t, A^t}\left[ \left( \mathcal{G}_{V_\mu, \pi_\mu}\left(S^t, A^t, S^{t+1}\right) - \eta\left(S^t\right) + \varpi\left(A^t \mid S^t\right) - V_\mu\left(S^t\right) \right) \cdot \right. \right.$$
$$\left. \left. K\left(\cdot\,; S^t, A^t\right) \right], u\left(S^t, A^t\right) \right\rangle_{\mathcal{H}_\text{RKHS}}^2$$

$$= \min_{V_\mu, \pi_\mu, \eta, \varpi} \left\langle \mathbb{E}_{S^t, A^t}\left[ \left( \mathcal{G}_{V_\mu, \pi_\mu}\left(S^t, A^t, S^{t+1}\right) - \eta\left(S^t\right) + \varpi\left(A^t \mid S^t\right) - V_\mu\left(S^t\right) \right) \cdot \right. \right.$$
$$\left. \left. K\left(\cdot\,; S^t, A^t\right) \right], \frac{\sqrt{C_0}\widetilde{u}}{\|\widetilde{u}\|_{\mathcal{H}_\text{RKHS}}} \right\rangle_{\mathcal{H}_\text{RKHS}}^2,$$

where the last equality holds because of the maximization of inner product between $\tilde{u}$ and $\mathbb{E}_{S^t,A^t}\left[\left(\mathcal{G}_{V_\mu,\pi_\mu}(S^t,A^t,S^{t+1}) - \eta(S^t) + \varpi(A^t|S^t) - V_\mu(S^t)\right)K(\cdot\,;S^t,A^t)\right]$ should have the same direction. Then we have,

$$
\min_{V_\mu,\pi_\mu,\eta,\varpi}\left\langle \mathbb{E}_{S^t,A^t}\left[\left(\mathcal{G}_{V_\mu,\pi_\mu}\left(S^t,A^t,S^{t+1}\right) - \eta\left(S^t\right) + \varpi\left(A^t\mid S^t\right) - V_\mu\left(S^t\right)\right)\cdot K\left(\cdot\,;S^t,A^t\right)\right],\right.
$$
$$
\left.\sqrt{C_0}\tilde{u}/\|\tilde{u}\|_{\mathcal{H}_{\mathrm{RKHS}}}\right\rangle^2_{\mathcal{H}_{\mathrm{RKHS}}}
$$
$$
= \min_{V_\mu,\pi_\mu,\eta,\varpi}\left\langle \mathbb{E}_{S^t,A^t}\left[\left(\mathcal{G}_{V_\mu,\pi_\mu}\left(S^t,A^t,S^{t+1}\right) - \eta\left(S^t\right) + \varpi\left(A^t\mid S^t\right) - V_\mu\left(S^t\right)\right)\cdot K(\cdot\,;S^t,A^t)\right],\right.
$$
$$
\left.\mathbb{E}_{S^t,A^t}\left[\left(\mathcal{G}_{V_\mu,\pi_\mu}\left(S^t,A^t,S^{t+1}\right) - \eta\left(S^t\right) + \varpi\left(A^t\mid S^t\right) - V_\mu\left(S^t\right)\right)\cdot K(\cdot\,;S^t,A^t)\right]\right\rangle
$$
$$
\cdot\left\langle \frac{\tilde{u}}{\|\tilde{u}\|_{\mathcal{H}_{\mathrm{RKHS}}}},\frac{C_0\tilde{u}}{\|\tilde{u}\|_{\mathcal{H}_{\mathrm{RKHS}}}}\right\rangle_{\mathcal{H}_{\mathrm{RKHS}}}
$$
$$
= \min_{V_\mu,\pi_\mu,\eta,\varpi}\left\langle \mathbb{E}_{S^t,A^t}\left[\left(\mathcal{G}_{V_\mu,\pi_\mu}\left(S^t,A^t,S^{t+1}\right) - \eta\left(S^t\right) + \varpi\left(A^t\mid S^t\right) - V_\mu\left(S^t\right)\right)\cdot K\left(\cdot\,;S^t,A^t\right)\right],\right.
$$
$$
\left.C_0\mathbb{E}_{\widetilde{S}^t,\widetilde{A}^t}\left[\left(\mathcal{G}_{V_\mu,\pi_\mu}\left(\widetilde{S}^t,\widetilde{A}^t,\widetilde{S}^{t+1}\right) - \eta\left(\widetilde{S}^t\right) + \varpi(\widetilde{A}^t\mid \widetilde{S}^t) - V_\mu(\widetilde{S}^t)\right)\cdot K\left(\cdot\,;\widetilde{S}^t,\widetilde{A}^t\right)\right]\right\rangle_{\mathcal{H}_{\mathrm{RKHS}}},
$$

where the first equality is by the equality condition of Cauchy-Schwarz inequality, i.e. $\tilde{u}/\|\tilde{u}\|_{\mathcal{H}_{\mathrm{RKHS}}}$ is linear dependent of $\mathbb{E}_{S^t,A^t}\left[\left(\mathcal{G}_{V_\mu,\pi_\mu}(S^t,A^t,S^{t+1}) - \eta(S^t) + \varpi(A^t|S^t) - V_\mu(S^t)\right)K(\cdot\,;S^t,A^t)\right]$. Then, by the reproducing property of $K(S^t,A^t;\tilde{S}^t,\tilde{A}^t)$, we have

$$
\min_{V_\mu,\pi_\mu,\eta,\varpi}\max_{u\in\mathcal{H}_\mathcal{K}^{C_0}}\mathcal{L}^2(V_\mu,\pi_\mu,\eta,\varpi,u)
$$
$$
= \min_{V_\mu,\pi_\mu,\eta,\varpi}\mathbb{E}_{S^t,\widetilde{S}^t,A^t,\widetilde{A}^t}\left[\left(\mathcal{G}_{V_\mu,\pi_\mu}(S^t,A^t,S^{t+1}) - \eta(S^t) + \varpi(S^t,A^t)\right) - V_\mu(S^t)\right)
$$
$$
C_0\left\langle K\left(S^t,A^t;\cdot\right),K\left(\widetilde{S}^t,\widetilde{A}^t;\cdot\right)\right\rangle_{\mathcal{H}_{\mathrm{RKHS}}}\left(\mathcal{G}_{V_\mu,\pi_\mu}(\widetilde{S}^t,\widetilde{A}^t,\widetilde{S}^{t+1}) - \eta(\widetilde{S}^t) + \varpi(\widetilde{A}^t|\widetilde{S}^t) - V_\mu(\widetilde{S}^t)\right)\right]
$$
$$
= \min_{V_\mu,\pi_\mu,\eta,\varpi}\mathbb{E}_{S^t,\widetilde{S}^t,A^t,\widetilde{A}^t}C_0\left[\left(\mathcal{G}_{V_\mu,\pi_\mu}(S^t,A^t,S^{t+1}) - \eta(S^t) + \varpi(S^t,A^t)\right) - V_\mu(S^t)\right)
$$
$$
K\left(S^t,A^t;\widetilde{S}^t,\widetilde{A}^t\right)\left(\mathcal{G}_{V_\mu,\pi_\mu}(\widetilde{S}^t,\widetilde{A}^t,\widetilde{S}^{t+1}) - \eta(\widetilde{S}^t) + \varpi(\widetilde{A}^t|\widetilde{S}^t) - V_\mu(\widetilde{S}^t)\right)\right]
$$
$$
= \min_{V_\mu,\pi_\mu,\eta,\varpi}\mathbb{E}_{S^t,\widetilde{S}^t,A^t,\widetilde{A}^t,S^{t+1},\widetilde{S}^{t+1}}C_0\left[\left(\widetilde{\mathcal{G}}_{V_\mu,\pi_\mu}(S^t,A^t,S^{t+1}) - \eta(S^t) + \varpi(S^t,A^t)\right) - V_\mu(S^t)\right)
$$
$$
K\left(S^t,A^t;\widetilde{S}^t,\widetilde{A}^t\right)\left(\widetilde{\mathcal{G}}_{V_\mu,\pi_\mu}(\widetilde{S}^t,\widetilde{A}^t,\widetilde{S}^{t+1}) - \eta(\widetilde{S^t}) + \varpi(\widetilde{A}^t|\widetilde{S}^t) - V_\mu(\widetilde{S}^t)\right)\right].
$$

Thus, we finish the proof.

## B.4 PROOFS ON GENERIC PROPERTIES OF QUASI-OPTIMAL BELLMAN OPERATOR

### B.4.1 PROOF OF PROPOSITION S.1

**Proposition S.1.** *The quasi-optimal Bellman operator $\mathcal{B}_\mu$ is $\gamma$-contractive with respect to the supreme norm over $\mathcal{S}$. That is $\|\mathcal{B}_\mu V - \mathcal{B}_\mu V'\|_\infty \leq \gamma\|V - V'\|_\infty$, for any generic value functions $\{V,V':\mathcal{S}\to\mathbb{R}\}$.*

Proposition S.1 justifies that there exists a unique fixed point of $\mathcal{B}_\mu$, i.e., $V_\mu^*$, indicating that the quasi-optimal value function $V_\mu^*$ and the induced policy $\pi_\mu^*$ are well defined and unique.

**Proof:** By the definition of $\mathcal{B}_\mu$, the explicit form corresponding to $V$ is as follows:

$$
\mathcal{B}_\mu V(s) = \max_\pi\mathbb{E}_{a\sim\pi(\cdot|s)}\left[\mathbb{E}_{S^{t+1}|s,a}[R(S^{t+1},s,a) + \gamma V(S^{t+1})] + \mu\mathrm{prox}^\circ(\pi(a|s))\right].
$$

For any two arbitrary value functions $V$ and $V'$, we have

$$
\|\mathcal{B}_\mu V(s) - \mathcal{B}_\mu V'(s)\|_\infty
$$

$$
= \max_{\pi_1} \mathbb{E}_{a \sim \pi(\cdot|s)} \Big[ \mathbb{E}_{S^{t+1}|s,a}[R(S^{t+1}, s, a) + \gamma V(S^{t+1})] + \mu \mathrm{prox}^\circ(\pi_1(a|s)) \Big] -
$$

$$
\max_{\pi_2} \mathbb{E}_{a \sim \pi(\cdot|s)} \Big[ \mathbb{E}_{S^{t+1}|s,a}[R(S^{t+1}, s, a) + \gamma V'(S^{t+1})] + \mu \mathrm{prox}^\circ(\pi_2(a|s)) \Big]
$$

$$
\leq \max_\pi \Bigg\{ \mathbb{E}_{a \sim \pi(\cdot|s)} \Big[ \mathbb{E}_{S^{t+1}|s,a}[R(S^{t+1}, s, a) + \gamma V(S^{t+1})] + \mu \mathrm{prox}^\circ(\pi(a|s)) \Big] -
$$

$$
\mathbb{E}_{a \sim \pi(\cdot|s)} \Big[ \mathbb{E}_{S^{t+1}|s,a}[R(S^{t+1}, s, a) + \gamma V'(S^{t+1})] + \mu \mathrm{prox}^\circ(\pi(a|s)) \Big] \Bigg\}
$$

$$
= \max_\pi \gamma \mathbb{E}_{a \sim \pi(\cdot|s), S^{t+1}|s,a} \Big[ \big( V(S^{t+1}) - V'(S^{t+1}) \big) \Big]
$$

$$
\leq \gamma \|V(s) - V'(s)\|_\infty.
$$

### B.4.2 PROOF OF PROPOSITION S.2

**Proposition S.2.** *For any $s \in \mathcal{S}$, the performance error between $V_\mu^*(s)$ and $V^*(s)$ satisfies*

$$
\|V_\mu^* - V^*\|_\infty \leq \frac{\mu \cdot \max\{|1 - \mathbf{C}|, 1\}}{1 - \gamma},
$$

*where $\mathbf{C}$ is the upper bound for induced policy $\pi_\mu$.*

**Proof of Proposition S.2:**

$$
\|V_\mu^* - V^*\|_\infty = \|\mathcal{B}_\mu V_\mu^* - \mathcal{B} V^*\|_\infty
$$

$$
\leq \|\mathcal{B}_\mu V_\mu^* - \mathcal{B}_\mu V^*\|_\infty + \|\mathcal{B}_\mu V^* - \mathcal{B} V^*\|_\infty.
$$

Notice that $\|\mathcal{B}_\mu V_\mu^* - \mathcal{B}_\mu V^*\|_\infty \leq \gamma \|V_\mu^* - V^*\|_\infty$ by Theorem S.1, and $\|\mathcal{B}_\mu V^* - \mathcal{B} V^*\|_\infty \leq \mu \cdot \max\{|1 - \mathbf{C}|, 1\}$ by Proposition 3.1. Therefore,

$$
(1 - \gamma)\|V_\mu^* - V^*\|_\infty \leq \mu \cdot \max\{|1 - \mathbf{C}|, 1\}.
$$

We finish the proof.

### B.5 PROOF OF THEOREM 4.1

**Proof of Theorem 4.1:** We first prove that when $\mu \to \infty$, $\pi_\mu^*$ would degenerate to uniform distribution over $\mathcal{A}$. By (4), we only need to prove that for arbitrary small $\epsilon > 0$

$$
\Big| \frac{Q_\mu^*(s, a)}{2\mu} - \frac{\int_{a \in \mathcal{W}_s} Q_\mu^*(s, a) da}{2\mu \sigma(\mathcal{W}_s)} + \frac{1}{\sigma(\mathcal{W}_s)} - \frac{1}{\sigma(\mathcal{A})} \Big| < \epsilon.
$$

Lower bound:

$$
\frac{Q_\mu^*(s, a)}{2\mu} - \frac{\int_{a \in \mathcal{W}_s} Q_\mu^*(s, a)}{2\mu \sigma(\mathcal{W}_s)} + \frac{1}{\sigma(\mathcal{W}_s)} \geq \frac{Q_\mu^*(s, a)}{2\mu} - \frac{\sigma(\mathcal{W}_s) \max_{a'} Q_\mu^*(s, a')}{2\mu \sigma(\mathcal{W}_s)} + \frac{1}{\sigma(\mathcal{W}_s)} \quad (18)
$$

$$
\geq \frac{Q_\mu^*(s, a)}{2\mu} - \frac{\max_{a'} Q_\mu^*(s, a')}{2\mu} + \frac{1}{\sigma(\mathcal{A})} \quad (19)
$$

Thus, we aim to prove that

$$
\Big| \frac{Q_\mu^*(s, a) - \max_{a'} Q_\mu^*(s, a')}{2\mu} \Big| \to 0.
$$

Let $V^*$ be the unique fixed point of (1), and $H_{\max} = \max H(\pi)$, where

$$
H(\pi) = \mathbb{E}_{a \sim \pi(\cdot|s)}[1 - \pi(a|s)].
$$

Let $r(s,a) := \mathbb{E}_{S^{t+1}|s,a}[R(S^{t+1}, s, a)]$, by the definition of $Q^*_\mu$, we have

$$\frac{Q^*_\mu(s,a)}{2\mu} - \frac{\gamma\mathbb{E}_{S^{t+1}|s,a}\left[V^*_\mu\left(S^{t+1}\right)\right]}{2\mu} = \frac{r(s,a)}{2\mu}$$

$$\frac{Q^*_\mu(s,a)}{2\mu} - \frac{\gamma\mathbb{E}_{S^{t+1}|s,a}\left[V^*_\mu\left(S^{t+1}\right) - V^*\left(S^{t+1}\right)\right]}{2\mu} - \frac{\gamma\mathbb{E}_{S^{t+1}|s,a}\left[V^*\left(S^{t+1}\right)\right]}{2\mu} = \frac{r(s,a)}{2\mu}.$$

Therefore,

$$\begin{aligned}\frac{Q^*_\mu(s,a)}{2\mu} - \frac{\mu\gamma H_{\max}}{2(1-\gamma)} &\leq \frac{r(s,a)}{2\mu} + \frac{\gamma\mathbb{E}_{s'|s,a}\left[V^*\left(s'\right)\right]}{2\mu}, \\ \frac{Q^*_\mu(s,a)}{2\mu} - \frac{\mu\gamma H_{\max}}{2(1-\gamma)} &\leq \frac{R_{\max}}{2(1-\gamma)\mu}.\end{aligned} \tag{20}$$

Meanwhile, from another perspective, the proximal Bellman operator (2) can be treated as a new MDP with the immediate reward $r(s,a) + \mu H(\pi(\cdot|s))$ for given $s, a$. Combine with the fact that

$$\frac{\gamma\mu H_{\max}}{1-\gamma} = \max_\pi \mathbb{E}_\pi\Big[\sum_{t=2}^\infty \gamma^{t-1}(\mu - \mu\pi(A^t|S^t))|S^1 = s, A^1 = a\Big].$$

Let $\pi_H = \operatorname{argmax}_\pi H(\pi(a|s))$, then

$$\begin{aligned}\frac{Q^*_\mu(s,a)}{2\mu} - \frac{\mu\gamma H_{\max}}{2(1-\gamma)} &= \frac{Q^*_\mu(s,a)}{2\mu} - \max_\pi \mathbb{E}_\pi\left[\sum_{t=2}^\infty \gamma^{t-1}\left(\mu - \mu\pi\left(A^t \mid S^t\right)\right) \mid S^1 = s, A^1 = a\right] \\ &\geq \frac{Q^{\pi_H}_\mu(s,a)}{2\mu} - \mathbb{E}_{\pi_H}\left[\sum_{t=2}^\infty \gamma^{t-1}\left(\mu - \mu\pi_H\left(A^t \mid S^t\right)\right) \mid S^1 = s, A^1 = a\right] \\ &= \mathbb{E}_{\pi_H}\left[\sum_{t=1}^\infty \gamma^{t-1}\frac{r\left(S^t, A^t\right)}{2\mu} \mid S^1 = s, A^1 = a\right] \\ &\geq -\frac{R_{\max}}{2(1-\gamma)\mu}.\end{aligned} \tag{21}$$

Based on (20) and (21), we have

$$\begin{aligned}\frac{Q^*_\mu(s,a)}{2\mu} - \frac{\max_{a'} Q^*_\mu(s,a')}{2\mu} &= \frac{Q^*_\mu(s,a)}{2\mu} - \frac{\gamma H_{\max}}{2(1-\gamma)} + \frac{\gamma H_{\max}}{2(1-\gamma)} - \frac{\max_{a'} Q^*_\mu(s,a')}{2\mu} \\ &\geq -\frac{R_{\max}}{(1-\gamma)\mu}.\end{aligned} \tag{22}$$

Similarly, we also have

$$\frac{Q^*_\mu(s,a)}{2\mu} - \frac{\max_{a'} Q^*_\mu(s,a')}{2\mu} \leq \frac{R_{\max}}{(1-\gamma)\mu}. \tag{23}$$

Therefore, we have the lower bound approaching to $\frac{1}{\sigma(\mathcal{A})}$.

For the upper bound, we have $\int_{a\in\mathcal{A}} \pi^*_\mu(a|s)da = 1$, thus

$$\begin{aligned}\int_{a\in\mathcal{A}} &\left\{\frac{Q^*_\mu(s,a)}{2\mu} - \frac{\int_{a'\in\mathcal{W}_s} Q^*_\mu(s,a')da'}{2\mu\sigma(\mathcal{W}_s)} + \frac{1}{\sigma(\mathcal{W}_s)}\right\}^+ da \\ &\geq \int_{a\in\mathcal{A}}\left\{\frac{\min_{a''} Q^*_\mu(s,a'')}{2\mu} - \frac{\int_{a'\in\mathcal{W}_s} Q^*_\mu(s,a')da'}{2\mu\sigma(\mathcal{W}_s)} + \frac{1}{\sigma(\mathcal{W}_s)}\right\}da \\ \frac{1}{\sigma(\mathcal{A})} &\geq \frac{\min_{a''} Q^*_\mu(s,a'')}{2\mu} - \frac{\int_{a'\in\mathcal{W}_s)} Q^*_\mu(s,a')da'}{2\mu} + \frac{1}{\sigma(\mathcal{W}_s)}.\end{aligned}$$

By (23), we then have

$$
\frac{Q_\mu^*(s,a)}{2\mu} - \frac{\int_{a\in\mathcal{W}_s} Q_\mu^*(s,a)da}{2\mu\sigma(\mathcal{W}_s)} + \frac{1}{\sigma(\mathcal{W}_s)} = \frac{Q_\mu^*(s,a)}{2\mu} - \frac{\max_{a''} Q_\mu^*(s,a'')}{2\mu} \tag{24}
$$

$$
+ \frac{\max_{a''} Q_\mu^*(s,a'')}{2\mu} - \frac{\int_{a'\in\mathcal{W}_s} Q_\mu^*(s,a')da'}{2\mu} + \frac{1}{\sigma(\mathcal{W}_s)}
$$

$$
\leq \frac{1}{\sigma(\mathcal{A})} + \frac{R_{\max}}{(1-\gamma)\mu} \tag{25}
$$

Therefore, by the lower bound and upper bound, we conclude that $\pi_\mu(a|s)$ will decay to the uniform distribution on $\mathcal{A}$ as $\mu \to \infty$.

For the case when $\mu \to 0$, we prove that $\pi_\mu$ would converge to the uniform distribution with the length of the support set equal to $\frac{1}{\mathbf{C}}$. Therefore, when $\mathbf{C} \to \infty$, it will converge to the point mass. According to (15), we only need to prove $\sigma(\mathcal{W}_{s,1}) \to 0$ as $\mu \to 0$. Meanwhile by Theorem (S.1), $a \in \mathcal{W}_{s,1}$, if

$$
\sigma(\mathcal{W}_{s,1})Q_\mu^*(s,a) - \left( \int_{a'\in\mathcal{W}_{s,1}} Q_\mu^*(s,a')da' - 2\mu + 2\mu\mathbf{C}\sigma(\mathcal{W}_{s,2}) \right) \in (0, 2\mu\mathbf{C}\sigma(\mathcal{W}_{s,1})).
$$

As $\mu \to 0$, $(0, 2\mu\mathbf{C}\sigma(\mathcal{W}_{s,1})) \to 0$. Thus, by squeeze theorem, we have $\sigma(\mathcal{W}_{s,1})Q_\mu^*(s,a) - \left( \int_{a'\in\mathcal{W}_{s,1}} Q_\mu^*(s,a')da' - 2\mu + 2\mu C\sigma(\mathcal{W}_{s,2}) \right) \to 0$ as $\mu \to 0$, which is equivalent to

$$
\sigma(\mathcal{W}_{s,1})Q_\mu^*(s,a) - \int_{a'\in\mathcal{W}_{s,1}} Q_\mu^*(s,a')da' \to 0 \quad \text{for all } a \in \mathcal{W}_{s,1}.
$$

Therefore, $\mathcal{W}_{s,1}$ could only include $a$ with the same value of $Q_\mu^*(s,a)$, which should only be a series of points rather than an interval. Thus, $\sigma(\mathcal{W}_{s,1}) = 0$, and $\pi_\mu^*(a|s)$ would converge to uniform distribution with interval length $\frac{1}{\mathbf{C}}$.

### B.6   PROOF OF LEMMA S.1

Before we prove the main result, we first provide a helper lemma for studying the boundedness of the symmetric kernel in the U-statistic.

**Lemma S.1.** *Under Assumption 1, for any $s \in \mathcal{S}, a \in \mathcal{A}$ and $\mu \in (0, \infty)$, we have that*

$$
\sup_{s\in\mathcal{S},a\in\mathcal{A}} \left| \mathcal{G}_{V_\mu,\pi_\mu}(s,a,s') - \eta(s) + \varpi(s,a) - V_\mu(s) \right| \leq M_{\max},
$$

*where $M_{\max} = \frac{4}{1-\gamma}R_{\max} + \mu\mathbf{C}$.*

**Proof of Lemma S.1:**

$$
\mathcal{G}_{V_\mu,\pi_\mu}(s,a,s') - \eta(s) + \varpi(s,a) - V_\mu(s)
$$
$$
= R(s',s,a) + \gamma V_\mu(s') + \mu - 2\mu\pi_\mu(a|s) - \eta(s) + \varpi(s,a) - V_\mu(s)
$$
$$
\leq R_{\max} + \mu + \mu\mathbf{C} + \gamma V_\mu(s') - V_\mu(s) \underbrace{-2\mu\pi_\mu(a|s) + \varpi(s,a)}_{(a)}.
$$

By checking the KKT conditions, we can further simplify the term (a). Specifically,

1. If $\pi_\mu = 0$, then $\varpi \geq 0$. By the stationarity equation (9), we have

$$
(a) = \varpi(s,a)
$$
$$
= \eta(s) - Q_\mu(s,a) - \mu + V_\mu(s)
$$
$$
\leq R_{\max} + \gamma\frac{R_{\max} - \mu H}{1-\gamma} - \mu + \frac{R_{\max} + \mu H}{1-\gamma} \quad \left( H := \mathbb{E}_{a\sim\pi_\mu(\cdot|s)}(1 - \pi_\mu(a|s)) \right)
$$
$$
\leq \frac{2}{1-\gamma}R_{\max} - \mu + \mu H
$$
$$
\leq \frac{2}{1-\gamma}R_{\max}.
$$

2. If $\pi_\mu \in (0, \mathbf{C}]$, then $\varpi = 0$

$$(a) = -2\mu\pi_\mu(a|s) < 0.$$

Therefore,

$$\mathcal{G}_{\pi_\mu}(s, a, s') - \eta(s) + \varpi(s, a) - V_\mu(s)$$

$$\leq R_{\max} + \mu + \mu\mathbf{C} + \gamma V_\mu(s') - V_\mu(s) + \frac{2}{1-\gamma}R_{\max}$$

$$\leq R_{\max} + \mu + \mu\mathbf{C} + \gamma\frac{R_{\max} + \mu H}{1-\gamma} - \frac{-R_{\max} + \mu H}{1-\gamma} + \frac{2}{1-\gamma}R_{\max}$$

$$\leq \frac{4}{1-\gamma}R_{\max} + \mu\mathbf{C} + \mu - \mu H$$

$$\leq \frac{4}{1-\gamma}R_{\max} + \mu\mathbf{C}.$$

Thus, we gain the upper bound. For the lower bound, the same technique is applied, and we can also gain that

$$\mathcal{G}_{V_\mu, \pi_\mu}(s, a, s') - \eta(s) + \varpi(s, a) - V_\mu(s) \geq -\frac{4}{1-\gamma}R_{\max} - \mu\mathbf{C}.$$

Therefore, this completes the proof.

## B.7 PROOF OF THEOREM 4.2

**Proof of Theorem 4.2:** We first define an operator $\mathcal{P}$ from $\mathcal{G}_{V_\mu, \pi_\mu}(S^k, A^k, S^{k+1})$ to $\mathcal{G}_{V_\mu, \pi_\mu}(S^k, A^k, S^{k+1}) - \eta(S^k) + \varpi(S^k, A^k)$ to simplify the expression, such that

$$\mathcal{P}\mathcal{G}_{V_\mu, \pi_\mu}(S^k, A^k, S^{k+1}) := \mathcal{G}_{V_\mu, \pi_\mu}(S^k, A^k, S^{k+1}) - \eta(S^k) + \varpi(A^k|S^k),$$

We further define several other notations

$$U_T := \binom{T}{2}^{-1} \sum_{1 \leq j \neq k \leq T} K(S^j, A^j; S^k, A^k)\{\mathcal{P}\mathcal{G}_{V_\mu, \pi_\mu}(S^j, A^j, S^{j+1}) - V_\mu(S^j)\}\cdot$$

$$\{\mathcal{P}\mathcal{G}_{V_\mu, \pi_\mu}(S^k, A^k, A^{k+1}) - V_\mu(S^k)\}$$

$$\tilde{K}\left(S^t, A^t, S^{t+1}; \widetilde{S}^t, \widetilde{A}^t, \widetilde{S}^{t+1}\right)$$

$$:= K\left(S^t, A^t; \widetilde{S}^t, \widetilde{A}^t\right)\{\mathcal{P}\mathcal{G}_{V_\mu, \pi_\mu}\left(S^t, A^t, S^{t+1}\right) - V_\mu\left(S^t\right)\}\{\mathcal{P}\mathcal{G}_{V_\mu, \pi_\mu}\left(\widetilde{S}^t, \widetilde{A}^t, \widetilde{S}^{t+1}\right) - V_\mu\left(\widetilde{S}^t\right)\}.$$

Let the expectation with respect to stationary trajectory and i.i.d training set as $\mathbb{E}_T$ and $\mathbb{E}$ respectively. For any finite threshold parameter $\mu < \infty$ and any $\epsilon > 0$, we have

$$\mathbb{P}\left(\left|\widehat{\mathcal{L}_U} - \mathcal{L}_U\right| > \epsilon\right) = \mathbb{P}\left(\left|\widehat{\mathcal{L}_U} - \mathbb{E}\left(U_T\right) + \mathbb{E}\left(U_T\right) - \mathcal{L}_U\right| > \epsilon\right)$$

$$\leq \underbrace{\mathbb{P}\left(\left|\widehat{\mathcal{L}_U} - \mathbb{E}\left(U_T\right)\right| > \frac{\epsilon}{2}\right)}_{(i)} + \underbrace{\mathbb{P}\left(\left|\mathbb{E}\left(U_T\right) - \mathcal{L}_U\right| > \frac{\epsilon}{2}\right)}_{(ii)}.$$

For $(i)$, since the Gaussian kernel satisfy that $|K(\cdot; \cdot)| \leq 1$, then by Lemma S.1, we have

$$\tilde{K}\left(s, a, s'; \tilde{s}, \tilde{a}, \tilde{s}'\right) \leq M_{\max}^2,$$

for any $s, \tilde{s}, a, \tilde{a}$. By Hoeffding's inequality, we have

$$(i) \leq 2\exp\left\{-\frac{n\epsilon^2}{2M_{\max}^4}\right\}. \tag{26}$$

For the term $(ii)$, the expectation of $U_T$ as $\mathbb{E}_T(U_T)$ can be calculated as follows:

$$\mathbb{E}_T(U_T) = \binom{T}{2}^{-1} \sum_{1 \leq j \neq k \leq T} \mathbb{E}_T\left[K(S^j, A^j; S^k, A^k)\{\mathcal{P}\mathcal{G}_{V_\mu, \pi_\mu}(S^j, A^j, S^{j+1}) - V_\mu(S^j)\}\cdot\right.$$

$$\left.\{\mathcal{P}\mathcal{G}_{V_\mu, \pi_\mu}(S^k, A^k, S^{k+1}) - V_\mu(S^j)\}\right].$$

If with-in trajectory samples are independent, then it is obvious that

$$\mathbb{E}_T(U_T) = \mathbb{E}_T\left[\tilde{K}\left(S^t, A^t, S^{t+1}; \widetilde{S}^t, \widetilde{A}^t, \widetilde{S}^{t+1}\right)\right] := U^*.$$

However, for weakly dependent data, dependency may introduce an additional bias term $\mathbb{E}_T(U_T) - U^*$, thus we further decompose the term $(ii)$ as

$$(ii) = \mathbb{P}(\underbrace{|\mathbb{E}(U_T) - \mathbb{E}[\mathbb{E}_T(U_T)]|}_{(iii)} + \underbrace{|\mathbb{E}[\mathbb{E}_T(U_T)] - \mathbb{E}U^\star)|}_{(iv)} > \frac{\epsilon}{2}).$$

For the term $(iii)$, we follow a similar idea to use a novel decomposition of the variance term of U-statistic from Han (2018). The idea is to break down the summation of U-statistic into numerous parts, where the current time is affected by randomness, and the historical time will be canceled out after conditioning on the future.

As $|\tilde{K}(\cdot\ ;\ \cdot)|$ is bounded by $M_{\max}^2$, under the mixing condition of Assumption 4.2, the exponential inequality from Merlevède et al. (2009) can be applied to to bound each decomposition part.

Then we follow the Theorem 3.1 from Han (2018) that for any $\epsilon_0$,

$$\mathbb{P}(|\mathbb{E}(U_T) - \mathbb{E}[\mathbb{E}_T(U_T)]| > \epsilon_0) \leq 2\exp\left\{-\left(\frac{M_{\max}^4}{T\epsilon_0^2 C_1'} + \frac{M_{\max}^2 \log\log(4T)\log T}{T\epsilon_0 C_1'}\right)^{-1}\right\}, \quad (27)$$

where $C_1'$ is some constant.

Then, we proceed to bound the term $(iv)$. By Hoeffding decomposition of kernel function $\tilde{K}\left(S^t, A^t, S^{t+1}; \tilde{S}^t, \tilde{A}^t, \tilde{S}^{t+1}\right)$, there exist kernel functions $\tilde{K}_1(S^t, A^t, S^{t+1})$ and $\tilde{K}_2\left(S^t, A^t, S^{t+1}; \tilde{S}^t, \tilde{A}^t, \tilde{S}^{t+1}\right)$ such that

$$\tilde{K}_1(s, a, s') = \mathbb{E}_T\tilde{K}\left(s, a, s'; \widetilde{S}^t, \widetilde{A}^t, \widetilde{S}^{t+1}\right) - U^*,$$

$$\tilde{K}_2\left(s, a, s'; \widetilde{s}, \widetilde{a}, \widetilde{s}'\right) = \tilde{K}\left(s, a, s'; \widetilde{s}, \widetilde{a}, \widetilde{s}'\right) - \tilde{K}_1(s, a, s') - \tilde{K}_1\left(\widetilde{s}, \widetilde{a}, \widetilde{s}'\right) - U^*,$$

and $\mathbb{E}_T\tilde{K}_1(S^t, A^t, S^{t+1}) = 0$, $\mathbb{E}_T\tilde{K}_2\left(S^t, A^t, S^{t+1}; \tilde{S}^t, \tilde{A}^t, \tilde{S}^{t+1}\right) = 0$. Then by Hoeffding decomposition of $U_T$, we have

$$U_T = U^* + \frac{2}{n}\sum_{t=1}^T \tilde{K}_1(S^t, A^t, S^{t+1}) + U_{\tilde{K}_2}.$$

Taking the expectation from both sides:

$$\mathbb{E}_T[U_T] = U^* + \frac{2}{n}\sum_{k=1}^T \mathbb{E}_T\tilde{K}_1(S^t, A^t, S^{t+1}) + \mathbb{E}_T[U_{\tilde{K}_2}]$$
$$= U^* + \mathbb{E}_T[U_{\tilde{K}_2}]$$

Therefore, by Lyapunov inequality, we can bound the bias term

$$|\mathbb{E}_T[U_T] - U^\star| = \left|\mathbb{E}_T\left[U_{\tilde{K}_2}\right]\right| \leq \mathbb{E}_T\left[|U_{\tilde{K}_2}|\right] \leq \sqrt{\mathbb{E}_T\left[U_{\tilde{K}_2}^2\right]}$$

$$= \sqrt{\sum_{1\leq h_1\leq l_1\leq T, 1\leq h_2\leq l_2\leq T} \mathbb{E}_T\left[\tilde{K}_2\left(S^{h_1}, A^{h_1}, S^{h_1+1}; S^{l_1}, A^{l_1}, S^{l_1+1}\right)\right.} \qquad (28)$$
$$\overline{\left.\cdot\tilde{K}_2\left(S^{h_2}, A^{h_2}, S^{h_2+1}; S^{l_2}, A^{l_2}, S^{l_2+1}\right)\right]\frac{4}{T^2(T-1)^2}}.$$

We proceed by the discussing the relationship between $h_1, h_2, l_1, l_2$.

**Case 1.1:** If $1 \leq h_1 \leq h_2 \leq l_1 \leq l_2 \leq T$ and $l_2 - l_1 \leq h_1 - h_2$.
Under the mixing condition assumption, and by Generalized Correlation inequality in Lemma 2 of, we have

$$\left| \mathbb{E}_T \left[ \tilde{K}_2 \left( S^{h_1}, A^{h_1}, S^{h_1+1}; S^{l_1}, A^{l_1}, S^{l_1+1} \right) \tilde{K}_2 \left( S^{h_2}, A^{h_2}, S^{h_2+1}; S^{l_2}, A^{l_2}, S^{l_2+1} \right) \right] \right|$$

$$\leq 4 \left( M_{\max}^{2r} \right)^{1/r} \beta^{1/s} \left( h_2 - h_1 \right),$$

where $1/r + 1/s = 1, s > -1$.

**Case 1.2:** If $1 \leq h_1 \leq h_2 \leq l_1 \leq l_2 \leq T$ and $h_1 - h_2 \leq l_2 - l_1$.
Similar as Case 1.1, we have

$$\left| \mathbb{E}_T \left[ \tilde{K}_2 \left( S^{h_1}, A^{h_1}, S^{h_1+1}; S^{l_1}, A^{l_1}, S^{l_1+1} \right) \tilde{K}_2 \left( S^{h_2}, A^{h_2}, S^{h_2+1}; S^{l_2}, A^{l_2}, S^{l_2+1} \right) \right] \right|$$

$$\leq 4 \left( M_{\max}^{2r} \right)^{1/r} \beta^{1/s} \left( l_2 - l_1 \right).$$

Combine Case 1.1 and Case 1.2, we apply the bounded inequalities (2.17-2.21) from Yoshihara (1976), and have the following result

$$\left| \sum_{1 \leq h_1 \leq h_2 \leq l_1 \leq l_2 \leq T} \mathbb{E}_T \left[ \tilde{K}_2 \left( S^{h_1}, A^{h_1}, S^{h_1+1}; S^{l_1}, A^{l_1}, S^{l_1+1} \right) \right.\right.$$

$$\left.\left. \tilde{K}_2 \left( S^{h_2}, A^{h_2}, S^{h_2+1}; S^{l_2}, A^{l_2}, S^{l_2+1} \right) \right] \right|$$

$$\leq \sum_{\substack{l_2 - l_1 \leq h_2 - h_1 \\ 1 \leq h_1 \leq h_2 \leq l_1 \leq l_2 \leq T}} \left| \mathbb{E}_T \left[ \tilde{K}_2 \left( S^{h_1}, A^{h_1}, S^{h_1+1}; S^{l_1}, A^{l_1}, S^{l_1+1} \right) \right.\right.$$

$$\left.\left. \cdot \tilde{K}_2 \left( S^{h_2}, A^{h_2}, S^{h_2+1}; S^{l_2}, A^{l_2}, S^{l_2+1} \right) \right] \right| +$$

$$\sum_{\substack{h_2 - h_1 \leq l_2 - l_2 \\ 1 \leq h_1 \leq h_2 \leq l_1 \leq l_2 \leq T}} \left| \mathbb{E}_T \left[ \tilde{K}_2 \left( S^{h_1}, A^{h_1}, S^{h_1+1}; S^{l_1}, A^{l_1}, S^{l_1+1} \right) \right.\right.$$

$$\left.\left. \tilde{K}_2 \left( S^{h_2}, A^{h_2}, S^{h_2+1}; S^{l_2}, A^{l_2}, S^{l_2+1} \right) \right] \right|$$

$$\leq M_{\max}^2 T^2 \sum_{j=1}^{T} (j+1) \beta^{1/s}(j) = \mathcal{O} \left( M_{\max}^2 T^{3-\tau} \right),$$

where

$$\tau = \frac{\left( \frac{2}{s+1} - \frac{2}{1-\delta_1} \right)}{\left( \frac{1}{\delta_1 - 1} \right) \left( 1 + \frac{1}{s+1} \right)}. \tag{29}$$

**Case 2:** If $1 \leq h_1 \leq l_1 \leq h_2 \leq l_2 \leq T$.
Using similar technique as Case 1.1 and 1.2, we have

$$\left| \sum_{1 \leq h_1 \leq l_1 \leq h_2 \leq l_2 \leq T} \mathbb{E}_T \left[ \tilde{K}_2 \left( S^{h_1}, A^{h_1}, S^{h_1+1}; S^{l_1}, A^{l_1}, S^{l_1+1} \right) \right.\right.$$

$$\left.\left. \tilde{K}_2 \left( S^{h_2}, A^{h_2}, S^{h_2+1}; S^{l_2}, A^{l_2}, S^{l_2+1} \right) \right] \right|$$

$$\leq \sum_{\substack{l_2 - h_2 \leq l_1 - h_1 \\ 1 \leq h_1 \leq l_1 \leq h_2 \leq l_2 \leq T}} \left| \mathbb{E}_T \left[ \tilde{K}_2 \left( S^{h_1}, A^{h_1}, S^{h_1+1}; S^{l_1}, A^{l_1}, S^{l_1+1} \right) \right.\right.$$

$$\left.\left. \cdot \tilde{K}_2 \left( S^{h_2}, A^{h_2}, S^{h_2+1}; S^{l_2}, A^{l_2}, S^{l_2+1} \right) \right] \right| +$$

$$\sum_{\substack{l_1 - h_1 \leq l_2 - h_2 \\ 1 \leq h_1 \leq l_1 \leq h_1 \leq l_2 \leq T}} \left| \mathbb{E}_T \left[ \tilde{K}_2 \left( S^{h_1}, A^{h_1}, S^{h_1+1}; S^{l_1}, A^{l_1}, S^{l_1+1} \right) \right.\right.$$

$$\left.\left. \tilde{K}_2 \left( S^{h_2}, A^{h_2}, S^{h_2+1}; S^{l_2}, A^{l_2}, S^{l_2+1} \right) \right] \right|$$

$$= \mathcal{O} \left( M_{\max}^2 T^{3-\tau} \right)$$

**Case 3:** If $1 \leq h_1 \leq l_1 \leq T$ and $1 \leq h_2 = l_2 \leq T$.
Following the same technique, we have

$$
\Bigg| \sum_{1 \leq h_2 = l_2 \leq T} \sum_{1 \leq h_1 \leq l_1 \leq T} \mathbb{E}_T \Big[ \tilde{K}_2 \left( S^{h_1}, A^{h_1}, S^{h_1+1}; S^{l_1}, A^{l_1}, S^{l_1+1} \right)
$$
$$
\cdot \tilde{K}_2 \left( S^{h_2}, A^{h_2}, S^{h_2+1}; S^{l_2}, A^{l_2}, S^{l_2+1} \right) \Big] \Bigg|
$$
$$
\leq \sum_{1 \leq h_1 = l_1 \leq T} \sum_{1 \leq h_2 = l_2 \leq T} \Big| \mathbb{E}_T \Big[ \tilde{K}_2 \left( S^{h_1}, A^{h_1}, S^{h_1+1}; S^{l_1}, A^{l_1}, S^{l_1+1} \right)
$$
$$
\tilde{K}_2 \left( S^{h_2}, A^{h_2}, S^{h_2+1}; S^{l_2}, A^{l_2}, S^{l_2+1} \right) \Big] \Big| +
$$
$$
2 \sum_{1 \leq h_1 < l_1 \leq T} \sum_{1 \leq h_2 = l_2 \leq T} \Big| \mathbb{E}_T \Big[ \tilde{K}_2 \left( S^{h_1}, A^{h_1}, S^{h_1+1}; \S^{l_1}, A^{l_1}, S^{l_1+1} \right)
$$
$$
\cdot \tilde{K}_2 \left( S^{h_2}, A^{h_2}, S^{h_2+1}; S^{l_2}, A^{l_2}, S^{l_2+1} \right) \Big] \Big|
$$
$$
\leq U_{\max}^2 T^2 + M_{\max}^2 T^2 \sum_{j=1}^{T} \beta^{1/s}(j) = \mathcal{O} \left( M_{\max}^2 T^2 \right).
$$

**Case 4:** If $1 \leq h_1 = l_1 \leq T$ and $1 \leq h_2 \leq l_2 \leq T$.
Using the same technique, we can obtain the same rate as follows:

$$
\Bigg| \sum_{1 \leq h_1 = l_1 \leq T} \sum_{1 \leq h_2 \leq l_2 \leq T} \mathbb{E}_T \Big[ \tilde{K}_2 \left( S^{h_1}, A^{h_1}, S^{h_1+1}; S^{l_1}, A^{l_1}, S^{l_1+1} \right)
$$
$$
\cdot \tilde{K}_2 \left( S^{h_2}, A^{h_2}, S^{h_2+1}; S^{l_2}, A^{l_2}, S^{l_2+1} \right) \Big] \Bigg|
$$
$$
= \mathcal{O} \left( M_{\max}^2 T^2 \right).
$$

Combine Case 1-4 with the equation (28), we conclude that
$$
|\mathbb{E} U_T - U^*| \leq C_0' M_{\max}^2 T^{-\frac{1+\tau}{2}} \quad a.s.
$$
We further use the continuous mapping theorem to conclude that
$$
\left| \mathbb{E}[\mathbb{E}_T(U_T)] - \mathbb{E} U^* \right| \leq C_0' M_{\max}^2 T^{-\frac{1+\tau}{2}} \quad a.s., \tag{30}
$$
where $\tau$ is defined in (29) and $C_0'$ is a constant.

As $\tau > 0$, we have $T^{-\frac{1+\tau}{2}} < T^{-\frac{1}{2}}$. Combine (27) and (30), for sufficiently large $T$, we have

$$
(ii) = \mathbb{P} \left( |\mathbb{E}\left(U_T\right) - \mathbb{E}\left[\mathbb{E}_T\left(U_T\right)\right]| + |\mathbb{E}\left[\mathbb{E}_T\left(U_T\right)\right] - \mathbb{E} U^\star| > \frac{\epsilon}{2} \right)
$$
$$
\leq 2 \exp \left( -\frac{T C_1' \left( \epsilon/2 - C_0' M_{\max}^2 T^{-(1+\tau)/2} \right)^2}{M_{\max}^4 + M_{\max}^2 \left( \epsilon/2 - C_0' M_{\max}^2 T^{-(1+\tau)/2} \right) \log T \log \log 4T} \right)
$$
$$
= 2 \exp \left( -\frac{T C_1' \epsilon^2/4 - T c_1 \epsilon C_0' M_{\max}^2 T^{-(1+\tau)/2} + T C_1' C_0'^2 M_{\max}^4 T^{-(1+\tau)}}{M_{\max}^4 + M_{\max}^2 \left( \epsilon/2 - C_0' M_{\max}^2 T^{-(1+\tau)/2} \right) \log T \log \log 4T} \right) \tag{31}
$$
$$
= 2 \exp \left( -\frac{T c_1 \epsilon^2/4 - T T^{-(1+\tau)/2} c_1 \epsilon C_0' M_{\max}^2 + c_1 C_0'^2 M_{\max}^4 T^{-\tau}}{M_{\max}^4 + M_{\max}^2 \left( \epsilon/2 - C_0' M_{\max}^2 T^{-(1+\tau)/2} \right) \log T \log \log 4T} \right)
$$

Then by the monotonicity of $\exp(\cdot)$,

$$
\frac{T T^{-(1+\tau)/2} C_1' \epsilon C_0' M_{\max}^2 - T^{-\tau} C_1' C_0'^2 M_{\max}^4 - T C_1' \epsilon^2/4}{M_{\max}^4 + \log T \log \log 4T M_{\max}^2 \epsilon/2 - T - (1+\tau)/2 \log T \log \log 4T C_0' M_{\max}^4}
$$
$$
\leq -\frac{T C_1' \epsilon^2/4 - T^{1/2} C_1' \epsilon C_0' M_{\max}^2 + T^{-\tau} C_1' C_0'^2 M_{\max}^4}{M_{\max}^4 + \log T \log \log 4T M_{\max}^2 \epsilon/2 - T^{-1/2} \log T \log \log 4T C_0' M_{\max}^4} \tag{32}
$$
$$
\leq -\frac{c C_1' \epsilon^2 T/4 - C_0' C_1' \epsilon M_{\max}^2 \sqrt{T}}{M_{\max}^2 \left( \epsilon/2 - C_0' M_{\max}^2/\sqrt{T} \right) \log T \log \log 4T + M_{\max}^4}
$$

where $C_1'$ is a constant. Combine (26) and (32), we simplify the terms and then

$$\mathbb{P}(|\widehat{\mathcal{L}}_U - \mathcal{L}_U| > \epsilon) \le C_1 \exp\left(-\frac{\epsilon^2 T - C_2\epsilon M_{\max}^2\sqrt{T}}{M_{\max}^2 + (\frac{\epsilon}{2} - \frac{C_2 M_{\max}^2}{\sqrt{T}})\log T \log\log(T)}\right) + C_3\exp\left(\frac{-n\epsilon^2}{M_{\max}^4}\right),$$

where $C_1, C_2, C_3$ are some constants depending on $\delta_1$ respectively, and $M_{\max} = \frac{4}{1-\gamma}R_{\max} + \mu\mathbf{C}$.

## B.8 PROOF OF THEOREM 4.3

*Proof of Theorem 4.3.* To bound the performance error, we first decompose it as

$$\|\widehat{V}_\mu^{\theta_1,k} - V^*\|_{L^2}^2 \le \|\widehat{V}_\mu^{\theta_1} - V_\mu^{\theta_1,k}\|_{L^2}^2 + \|\widehat{V}_\mu^{\theta_1} - V^*\|_{L^2}^2 + \epsilon_{\text{approximation error}}$$

where the first term is the optimization error and the last term is the approximation error. Then we proceed to bound

$$\|\widehat{V}_\mu^{\theta_1} - V^*\|_{L^2}^2 \le \left| \|\underbrace{\widehat{V}_\mu^{\theta_1} - V_\mu^{\widetilde{\pi}_\mu}\|_{L^2}}_{\Delta_1} + \underbrace{\|V_\mu^{\widetilde{\pi}_\mu} - V^*\|_{L^2}}_{\Delta_2} \right|^2.$$

where $V_\mu^{\widetilde{\pi}_\mu}$ satisfying the stationarity equation (9) and $V^*$ is the unique fixed point of $\mathcal{B}$. First, we move to bound $\Delta_1$. Follow a similar kernel reproducing property and a eigen decomposition spirit in Bertsekas (1997); Sutton & Barto (2018); Zhou et al. (2022), we have

$$\frac{2}{\kappa_{\min}}\left(\mathcal{L}_U(\widehat{V}_\mu^{\theta_1}, \widehat{\pi}_\mu^{\theta_2}, \eta^{\xi_1}, \varpi^{\xi_2}) - \mathcal{L}_U(V_\mu^{\widetilde{\pi}_\mu}, \widetilde{\pi}_\mu, \eta, \varpi)\right)+$$

$$2\|\left(\mu\text{prox}^\circ(\widehat{\pi}_\mu^{\theta_2}(A^t|S^t)) - \mu\text{prox}^\circ(\widetilde{\pi}_\mu(A^t|S^t))\right) - \left(\widehat{\eta}^{\xi_1}(S^t) - \eta(S^t)\right) + \left(\widehat{\varpi}^{\xi_2}(S^t, A^t) - \varpi(S^t, A^t)\right)\|_{L^2}^2$$

$$\ge \|\gamma\left(\mathbb{E}_{S^{t+1}|S^t,A^t}[\widehat{V}_\mu^{\theta_1}(S^{t+1})] - \mathbb{E}_{S^{t+1}|S^t,A^t}[V_\mu^{\widetilde{\pi}_\mu}(S^{t+1})]\right) - \left(\widehat{V}_\mu^{\theta_1}(S^t) - V_\mu^{\widetilde{\pi}_\mu}(S^t)\right)\|_{L^2}^2.$$

Then by

$$\|\mu\text{prox}^\circ(\widehat{\pi}_\mu^{\theta_2}(A^t|S^t)) - \mu\text{prox}^\circ(\widetilde{\pi}_\mu(A^t|S^t))\|_{L^2}^2 \le \mu^2\|\widehat{\pi}_\mu^{\theta_2}(A^t|S^t) - \widetilde{\pi}_\mu(A^t|S^t)\|_{L^2}^2 \le \mathbf{C}\mu^2.$$

and the auxiliary functions $\eta^{\xi_1}(s) \in [-\mathbf{C}\mu, 0]$ for any $s \in \mathcal{S}$, then

$$\|\widehat{\eta}^{\xi_1}(S^t) - \eta(S^t)\|_{L^2}^2 \le (\mathbf{C}\mu + \mathbf{C}\mu)^2 = (\mathbf{C}\mu)^2$$

$$\|\widehat{\eta}_1^{\xi_1}(S^t) - \eta_1(S^t)\|_{L^2}^2 \le \frac{2}{\kappa_{\min}}\left(\mathcal{L}_U(\widehat{V}_\mu^{\theta_1}, \widehat{\pi}_\mu^{\theta_2}, \widehat{\eta}^{\xi_1}, \widehat{\varpi}^{\xi_2}) - \mathcal{L}_U(V_\mu^{\widetilde{\pi}_\mu}, \widetilde{\pi}_\mu, \eta, \varpi)\right)$$

Then we conclude that

$$\|\widehat{V}_\mu^{\theta_1}(S^t) - V_\mu^{\widetilde{\pi}_\mu}(S^t)\|_{L^2}^2 \le \frac{C_5(\mathcal{L}_U(\widehat{V}_\mu^{\theta_1}, \widehat{\pi}_\mu^{\theta_2}, \widehat{\eta}^{\xi_1}, \widehat{\varpi}^{\xi_2}) - \mathcal{L}_U(V_\mu^{\widetilde{\pi}_\mu}, \widetilde{\pi}_\mu, \eta, \varpi))}{\kappa_{\min}(1-\gamma)^2} + \frac{C_6\mu^2}{(1-\gamma)^2}$$

$$\le \frac{C_5(\mathcal{L}_U(\widehat{V}_\mu^{\theta_1}, \widehat{\pi}_\mu^{\theta_2}, \widehat{\eta}^{\xi_1}, \widehat{\varpi}^{\xi_2}) - \mathcal{L}_U^*)}{\kappa_{\min}(1-\gamma)^2} + \frac{C_6\mu^2}{(1-\gamma)^2}$$

where $C_5$ and $C_6$ are some constants, and

$$\mathcal{L}_U^* := \inf_{\{V_\mu, \pi_\mu, \eta, \varpi\}} \mathcal{L}_U(V_\mu, \pi_\mu, \eta, \varpi)$$

Now, we have the remainder term $\Delta_2$ to bound.

$$\Delta_2 \le \underbrace{\|V_\mu^{\widetilde{\pi}_\mu} - V_\mu^*\|_{L^2}}_{\Delta_2^1} + \underbrace{\|V_\mu^* - V^*\|_{L^2}}_{\Delta_2^2}$$

We first bound $\Delta_2^1$. For any $s \in \mathcal{S}$, then we have that

$$\mathcal{B}_\mu V_\mu^{\widetilde{\pi}_\mu}(s) = \max_\pi \mathbb{E}_{a\sim\pi(\cdot|s),\,S^{t+1}|s,a}\left[R(S^{t+1}, s, a) + \gamma V_\mu^{\widetilde{\pi}_\mu}(S^{t+1}) + \mu\text{prox}(\pi(a|s))\right]$$

$$= \mathbb{E}_{a\sim\widetilde{\pi}_\mu(\cdot|s),\,S^{t+1}|s,a}\left[R(S^{t+1}, s, a) + \gamma V_\mu^{\widetilde{\pi}_\mu}(S^{t+1}) + \mu\text{prox}(\widetilde{\pi}_\mu(a|s))\right]$$

$$= \mathbb{E}_{a\sim\widetilde{\pi}_\mu(\cdot|s),\,S^{t+1}|s,a}\left[R(S^{t+1}, s, a) + \gamma V_\mu^{\widetilde{\pi}_\mu}(S^{t+1}) + \mu(1 - \widetilde{\pi}_\mu(a|s))\right]$$

$$= \mathbb{E}_{a\sim\widetilde{\pi}_\mu(\cdot|s),\,S^{t+1}|s,a}\left[R(S^{t+1}, s, a) + \gamma V_\mu^{\widetilde{\pi}_\mu}(S^{t+1}) + \mu - \mu\widetilde{\pi}_\mu(a|s)\right]+$$

$$\mathbb{E}_{a\sim\widetilde{\pi}_\mu(\cdot|s)}\left[\mu\widetilde{\pi}_\mu(a|s)\right].$$

As $(V_\mu^{\widetilde{\pi}_\mu}, \widetilde{\pi}_\mu)$ is the solution of the stationarity equation,

$$\mathbb{E}_{a\sim\widetilde{\pi}_\mu(\cdot|s),\ S^{t+1}|s,a}\Big[R(S^{t+1},s,a)+\gamma V_\mu^{\widetilde{\pi}_\mu}(S^{t+1})+\mu-\mu\widetilde{\pi}_\mu(a|s)\Big]\leq V_\mu^{\widetilde{\pi}_\mu}(s)$$

and since $\mathbb{E}_{a\sim\widetilde{\pi}_\mu(\cdot|s)}\Big[\mu\widetilde{\pi}_\mu(a|s)\Big]\leq\mu$, then we have

$$\mathcal{B}_\mu V_\mu^{\widetilde{\pi}_\mu}(s)\leq V_\mu^{\widetilde{\pi}_\mu}(s)+\mu\mathbf{C}.$$

For the lower bound, as

$$\mathbb{E}_{a\sim\widetilde{\pi}_\mu(\cdot|s),\ S^{t+1}|s,a}\Big[R(S^{t+1},s,a)+\gamma V_\mu^{\widetilde{\pi}_\mu}(S^{t+1})+\mu-\mu\widetilde{\pi}_\mu(a|s)-V_\mu^{\widetilde{\pi}_\mu}(s)\mid S^t=s\Big]\geq-\mathbf{C}\mu$$

so similarly, we conclude that

$$\mathbf{C}\mu+\mathcal{B}_\mu V_\mu^{\widetilde{\pi}_\mu}(s)\geq V_\mu^{\widetilde{\pi}_\mu}(s).$$

If follows the definition of the proximal Bellman operator $\mathcal{B}_\mu$ and due to the monotonicity of the Bellman operator that $\mathcal{B}_\mu V_1(s)\geq\mathcal{B}_\mu V_2(s)$ for generic value functions $V_1(s)\geq V_2(s)$, and the $\mathcal{B}_\mu V(s)\geq\mathcal{B}_\mu^{\widetilde{\pi}_\mu}V(s)$ for any generic value function $V$, where $\mathcal{B}_\mu^{\widetilde{\pi}_\mu}$ is the proximal Bellman evaluation operator, i.e.,

$$\mathcal{B}_\mu^{\widetilde{\pi}_\mu}V(s):=\mathbb{E}_{a\sim\widetilde{\pi}_\mu(\cdot|s),\ S^{t+1}|s,a}\Big[R(S^{t+1},s,a)+\gamma V(S^{t+1})+\mu\mathrm{prox}\big(\widetilde{\pi}_\mu(a|s)\big)\Big].$$

Note that, $V_\mu^{\widetilde{\pi}_\mu}$ is unique fixed point of the Bellman operator $\mathcal{B}_\mu^{\widetilde{\pi}_\mu}$, thus $\lim_{i\to\infty}(\mathcal{B}_\mu^{\widetilde{\pi}_\mu})^i V_\mu^{\widetilde{\pi}_\mu}(s)=V_\mu^{\widetilde{\pi}_\mu}(s)$, where $i\in\mathbb{Z}^+$. And for any initial value function. e.g., $V_\mu^{\widetilde{\pi}_\mu}$, $\lim_{i\to\infty}(\mathcal{B}_\mu)^i V_\mu^{\widetilde{\pi}_\mu}(s)=V_\mu^*(s)$ holds. Therefore the following inequality holds that

$$V_\mu^{\widetilde{\pi}_\mu}(s)=\lim_{i\to\infty}(\mathcal{B}_\mu^{\widetilde{\pi}_\mu})^i V_\mu^{\widetilde{\pi}_\mu}(s)\leq\lim_{i\to\infty}(\mathcal{B}_\mu^{\widetilde{\pi}_\mu})^i\Big(V_\mu^{\widetilde{\pi}_\mu}+\mathbf{C}\mu\Big)(s)\leq\lim_{i\to\infty}(\mathcal{B}_\mu)^i\Big(V_\mu^{\widetilde{\pi}_\mu}+\mathbf{C}\mu\Big)(s)$$

$$\implies V_\mu^{\widetilde{\pi}_\mu}(s)\leq\lim_{i\to\infty}(\mathcal{B}_\mu)^i V_\mu^{\widetilde{\pi}_\mu}(s)+\sum_{i=1}^\infty\mathbf{C}\mu\gamma^{i-1}\leq V_\mu^*(s)+\frac{\mathbf{C}\mu}{(1-\gamma)}. \tag{33}$$

We repeatedly apply a similar procedure, without loss of generality. We first show one step that

$$\mathcal{B}_\mu(\mathcal{B}_\mu V_\mu^{\widetilde{\pi}_\mu}(s))\leq\mathcal{B}_\mu(V_\mu^{\widetilde{\pi}_\mu}(s)+\mathbf{C}\mu)=\mathcal{B}_\mu(V_\mu^{\widetilde{\pi}_\mu}(s))+\mathbf{C}\mu\gamma\leq V_\mu^{\widetilde{\pi}_\mu}(s)+\mathbf{C}\mu+\mathbf{C}\mu\gamma.$$

Then we apply infinite many time $\mathcal{B}_\mu$, then we can have that

$$V_\mu^*(s)=\lim_{i\to\infty}(\mathcal{B}_\mu)^i V_\mu^{\widetilde{\pi}_\mu}(s)\leq V_\mu^{\widetilde{\pi}_\mu}(s)+\sum_{i=1}^\infty\mathbf{C}\mu\gamma^{i-1}=V_\mu^{\widetilde{\pi}_\mu}(s)+\frac{\mathbf{C}\mu}{(1-\gamma)}. \tag{34}$$

Combine with the inequalities (33)-(34), we immediately have that

$$\|V_\mu^*-V_\mu^{\widetilde{\pi}_\mu}\|_{L^2}\leq\frac{\mathbf{C}\mu}{(1-\gamma)}$$

Next, by Proposition S.2, we have

$$\|V_\mu^*-V^*\|_\infty\leq\frac{\mu\cdot\max\{|1-\mathbf{C}|,1\}}{1-\gamma},$$

Now, we need to bound the excess risk. The excess risk can be decomposed into approximation error and estimation error, i.e.

$$\mathcal{L}_U(\widehat{V}_\mu^{\theta_1},\widehat{\pi}_\mu^{\theta_2},\widehat{\eta}^{\xi_1},\widehat{\varpi}^{\xi_2})-\mathcal{L}_U^*=\underbrace{\left(\inf_{(V_\mu^{\theta_1},\pi_\mu^{\theta_2},\eta^{\xi_1},\varpi^{\xi_2})\in\Theta_1\times\Theta_2\times\Xi_1\times\Xi_2}\mathcal{L}_U(V_\mu^{\theta_1},\pi_\mu^{\theta_2},\eta^{\xi_1},\varpi^{\xi_2})-\mathcal{L}_U^*\right)}_{\Delta_{\mathrm{approx}}}$$

$$+\underbrace{\left(\mathcal{L}_U\left(\widehat{V}_\mu^{\theta_1},\widehat{\pi}_\mu^{\theta_2},\widehat{\eta}^{\xi_1},\widehat{\varpi}^{\xi_2}\right)-\inf_{(V_\mu^{\theta_1},\pi_\mu^{\theta_2},\eta^{\xi_1},\varpi^{\xi_2})\in\Theta_1\times\Theta_2\times\Xi_1\times\Xi_2}\mathcal{L}_U(V_\mu^{\theta_1},\pi_\mu^{\theta_2},\eta^{\xi_1},\varpi^{\xi_2})\right)}_{\Delta_{\mathrm{est}}},$$

where $\Delta_{\text{approx}}$ is the approximation error and $\Delta_{\text{est}}$ is the estimation error. The approximation error is assumed to be zero in our proof for simplicity. At first, we consider to bound the estimation error.

$$\mathcal{L}_U\left(\widehat{V}_\mu^{\theta_1}, \widehat{\pi}_\mu^{\theta_2}, \widehat{\eta}^{\xi_1}, \widehat{\varpi}^{\xi_2}\right) - \inf_{(V_\mu^{\theta_1}, \pi_\mu^{\theta_2}, \eta^{\xi_1}, \varpi^{\xi_2}) \in \Theta_1 \times \Theta_2 \times \Xi_1 \times \Xi_2} \mathcal{L}_U(V_\mu^{\theta_1}, \pi_\mu^{\theta_2}, \eta^{\xi_1}, \varpi^{\xi_2})$$

$$:= \mathcal{L}_U\left(\widehat{V}_\mu^{\theta_1}, \widehat{\pi}_\mu^{\theta_2}, \widehat{\eta}^{\xi_1}, \widehat{\varpi}^{\xi_2}\right) - \mathcal{L}_U(V_\mu^{\pi_\mu^\circ}, \pi_\mu^\circ, \eta^{\xi_1}, \varpi^{\xi_2})$$

$$\leq \mathcal{L}_U\left(\widehat{V}_\mu^{\theta_1}, \widehat{\pi}_\mu^{\theta_2}, \widehat{\eta}^{\xi_1}, \widehat{\varpi}^{\xi_2}\right) - \mathcal{L}_U(V_\mu^{\pi_\mu^\circ}, \pi_\mu^\circ, \eta^{\xi_1}, \varpi^{\xi_2}) + \widehat{\mathcal{L}_U}(V_\mu^{\pi_\mu^\circ}, \pi_\mu^\circ, \eta^{\xi_1}, \varpi^{\xi_2}) - \widehat{\mathcal{L}_U}\left(\widehat{V}_\mu^{\theta_1}, \widehat{\pi}_\mu^{\theta_2}, \widehat{\eta}^{\xi_1}, \widehat{\varpi}^{\xi_2}\right)$$

$$\leq \left(\mathcal{L}_U\left(\widehat{V}_\mu^{\theta_1}, \widehat{\pi}_\mu^{\theta_2}, \widehat{\eta}^{\xi_1}, \widehat{\varpi}^{\xi_2}\right) - \widehat{\mathcal{L}_U}\left(\widehat{V}_\mu^{\theta_1}, \widehat{\pi}_\mu^{\theta_2}, \widehat{\eta}^{\xi_1}, \widehat{\varpi}^{\xi_2}\right)\right) - \left(\mathcal{L}_U(V_\mu^{\pi_\mu^\circ}, \pi_\mu^\circ, \eta^{\xi_1}, \varpi^{\xi_2}) - \widehat{\mathcal{L}_U}(V_\mu^{\pi_\mu^\circ}, \pi_\mu^\circ, \eta^{\xi_1}, \varpi^{\xi_2})\right)$$

$$\leq 2 \sup_{(V_\mu^{\theta_1}, \pi_\mu^{\theta_2}, \eta^{\xi_1}, \varpi^{\xi_2}) \in \Theta_1 \times \Theta_2 \times \Xi_1 \times \Xi_2} \left|\mathcal{L}_U(V_\mu^{\theta_1}, \pi_\mu^{\theta_2}, \eta^{\xi_1}, \varpi^{\xi_2}) - \widehat{\mathcal{L}_U}(V_\mu^{\theta_1}, \pi_\mu^{\theta_2}, \eta^{\xi_1}, \varpi^{\xi_2})\right|.$$

where $\eta^{\xi_1}, \varpi^{\xi_2}$ are Lagrange multipliers satisfying minimal Bayes risk associated with $V_\mu^{\pi_\mu^\circ}, \pi_\mu^\circ$ for the rest of this proof. Observe that the randomness of $\sup_{(V_\mu^{\theta_1}, \pi_\mu^{\theta_2}, \eta^{\xi_1}, \varpi^{\xi_2}) \in \Theta_1 \times \Theta_2 \times \Xi_1 \times \Xi_2} |\mathcal{L}_U(V_\mu^{\theta_1}, \pi_\mu^{\theta_2}, \eta^{\xi_1}, \varpi^{\xi_2}) - \widehat{\mathcal{L}_U}(V_\mu^{\theta_1}, \pi_\mu^{\theta_2}, \eta^{\xi_1}, \varpi^{\xi_2})|$ can be decomposed into two parts, one is from the $n$ number of i.i.d. trajectories and another one is from the dependent transition within each trajectory. For each single trajectory, we define the quantity

$$U^\star(V_\mu^{\theta_1}, \pi_\mu^{\theta_2}, \eta^{\xi_1}, \varpi^{\xi_2}) = \Lambda_{V_\mu^{\theta_1}, \pi_\mu^{\theta_2}}(S_i^t, A_i^t, S_i^{t+1}) K(S_i^t, A_i^t; \widetilde{S}_i^t, \widetilde{A}_i^t) \Lambda_{V_\mu^{\theta_1}, \pi_\mu^{\theta_2}}(\widetilde{S}_i^t, \widetilde{A}_i^t, V_\mu^{\theta_1}),$$

where $\mathbb{E}_T$ is defined as taking expectation to single stationary trajectory and $\mathbb{E}$ is defined as taking expectation to i.i.d. trajectory random variable $\mathcal{D}_1$, respectively. Without loss of generality, we assume $C_0 = 1$. The U-statistic approximation for $\mathbb{E}_T(U^\star)$ is as follows:

$$U_T(V_\mu^{\theta_1}, \pi_\mu^{\theta_2}, \eta^{\xi_1}, \varpi^{\xi_2})$$

$$:= \frac{2}{T(T-1)} \sum_{1 \leq j \neq k \leq T} \left[\left(\Lambda_{V_\mu^{\theta_1}, \pi_\mu^{\theta_2}}(S_i^j, A_i^j, S_i^{j+1}) K(S_i^j, A_i^j; S_i^k, A_i^k)\left(\Lambda_{V_\mu^{\theta_1}, \pi_\mu^{\theta_2}}(S_i^k, A_i^k, V_\mu^{\theta_1})\right)\right].$$

Then the uniform process is bounded by

$$\sup_{(V_\mu^{\theta_1}, \pi_\mu^{\theta_2}, \eta^{\xi_1}, \varpi^{\xi_2}) \in \Theta_1 \times \Theta_2 \times \Xi_1 \times \Xi_2} \left|\mathcal{L}_U(V_\mu^{\theta_1}, \pi_\mu^{\theta_2}, \eta^{\xi_1}, \varpi^{\xi_2}) - \widehat{\mathcal{L}_U}(V_\mu^{\theta_1}, \pi_\mu^{\theta_2}, \eta^{\xi_1}, \varpi^{\xi_2})\right|$$

$$\leq \sup_{(V_\mu^{\theta_1}, \pi_\mu^{\theta_2}, \eta^{\xi_1}, \varpi^{\xi_2}) \in \Theta_1 \times \Theta_2 \times \Xi_1 \times \Xi_2} \left|\mathcal{L}_U(V_\mu^{\theta_1}, \pi_\mu^{\theta_2}, \eta^{\xi_1}, \varpi^{\xi_2}) - \mathbb{P}_n^{(\mathcal{D}_{i:n})} \mathbb{E}_T[U^\star(V_\mu^{\theta_1}, \pi_\mu^{\theta_2}, \eta^{\xi_1}, \varpi^{\xi_2})]\right|$$

$$+ \sup_{(V_\mu^{\theta_1}, \pi_\mu^{\theta_2}, \eta^{\xi_1}, \varpi^{\xi_2}) \in \Theta_1 \times \Theta_2 \times \Xi_1 \times \Xi_2} \left|\mathbb{P}_n^{(\mathcal{D}_{i:n})} \mathbb{E}_T[U^\star(V_\mu^{\theta_1}, \pi_\mu^{\theta_2}, \eta^{\xi_1}, \varpi^{\xi_2})] - \mathbb{P}_n^{(\mathcal{D}_{i:n})} U_T(V_\mu^{\theta_1}, \pi_\mu^{\theta_2}, \eta^{\xi_1}, \varpi^{\xi_2})]\right|$$

$$\leq \underbrace{\sup_{(V_\mu^{\theta_1}, \pi_\mu^{\theta_2}, \eta^{\xi_1}, \varpi^{\xi_2}) \in \Theta_1 \times \Theta_2 \times \Xi_1 \times \Xi_2} \left|\mathcal{L}_U(V_\mu^{\theta_1}, \pi_\mu^{\theta_2}, \eta^{\xi_1}, \varpi^{\xi_2}) - \mathbb{P}_n^{(\mathcal{D}_{i:n})} \mathbb{E}_T[U^\star(V_\mu^{\theta_1}, \pi_\mu^{\theta_2}, \eta^{\xi_1}, \varpi^{\xi_2})]\right|}_{\Delta_1}$$

$$+ \frac{1}{n} \sum_{i=1}^n \sup_{(V_\mu^{\theta_1}, \pi_\mu^{\theta_2}, \eta^{\xi_1}, \varpi^{\xi_2}) \in \Theta_1 \times \Theta_2 \times \Xi_1 \times \Xi_2} \left|\mathbb{E}_T[U^\star(V_\mu^{\theta_1}, \pi_\mu^{\theta_2}, \eta^{\xi_1}, \varpi^{\xi_2})] - U_T(V_\mu^{\theta_1}, \pi_\mu^{\theta_2}, \eta^{\xi_1}, \varpi^{\xi_2})]\right|,$$

where $\mathbb{P}_n^{(\mathcal{D}_{i:n})}$ is the empirical measure with respect to $\mathcal{D}_{i:n} = \{\mathcal{D}_i\}_{i=1}^n$ and we simply denotes it as $\mathbb{P}_n$ in the following proof. The last term is the bound for uniform process w.r.t sum of trajectories. In this sense, it is necessary to bound

$$\Delta_2 = \sup_{(V_\mu^{\theta_1}, \pi_\mu^{\theta_2}, \eta^{\xi_1}, \varpi^{\xi_2}) \in \Theta_1 \times \Theta_2 \times \Xi_1 \times \Xi_2} \left|\mathbb{E}_T[U^\star(V_\mu^{\theta_1}, \pi_\mu^{\theta_2}, \eta^{\xi_1}, \varpi^{\xi_2})] - U_T(V_\mu^{\theta_1}, \pi_\mu^{\theta_2}, \eta^{\xi_1}, \varpi^{\xi_2})]\right|,$$

since the trajectories $\{\mathcal{D}_i\}_{i=1}^n$ are i.i.d. Now, we process to bound $\Delta_1$. $\Delta_1$ can be re-expressed as the empirical process of $\{\mathcal{D}_i\}_{i=1}^n$ w.r.t. the probability space $(\Omega_N, \mathcal{F}_N, \mathbb{P})$ equipped with empirical measure $\mathbb{P}_n$ such that

$$\Delta_1 = \sup_{(V_\mu^{\theta_1}, \pi_\mu^{\theta_2}, \eta^{\xi_1}, \varpi^{\xi_2}) \in \Theta_1 \times \Theta_2 \times \Xi_1 \times \Xi_2} \left|\mathbb{E}(\mathbb{E}_T[U^\star(V_\mu^{\theta_1}, \pi_\mu^{\theta_2}, \eta^{\xi_1}, \varpi^{\xi_2})]) - \mathbb{P}_n \mathbb{E}_T[U^\star(V_\mu^{\theta_1}, \pi_\mu^{\theta_2}, \eta^{\xi_1}, \varpi^{\xi_2})]\right|$$

$$= \sup_{(V_\mu^{\theta_1}, \pi_\mu^{\theta_2}, \eta^{\xi_1}, \varpi^{\xi_2}) \in \Theta_1 \times \Theta_2 \times \Xi_1 \times \Xi_2} \left|\mathbb{E}G(V_\mu^{\theta_1}, \pi_\mu^{\theta_2}, \eta^{\xi_1}, \varpi^{\xi_2}; \mathcal{D}_i) - \mathbb{P}_n G(V_\mu^{\theta_1}, \pi_\mu^{\theta_2}, \eta^{\xi_1}, \varpi^{\xi_2}; \mathcal{D}_i)\right|,$$

where $G(V_\mu^{\theta_1}, \pi_\mu^{\theta_2}, \eta^{\xi_1}, \varpi^{\xi_2}; \mathcal{D}_i)$ is the random function associated with random variable $\mathcal{D}_i$. To bound $\Delta_1$, it is needed to calculate the covering number $\mathcal{N}(\epsilon, \mathcal{F}_{\theta,\xi}, \{\mathcal{D}_i\}_{i=1}^n)$ by Pollard's tail inequality (Pollard, 2012), where the function space is the composite space $\mathcal{F}_{\theta,\xi} = G(\Theta_1 \times \Theta_2 \times \Xi_1 \times \Xi_2)$. Specifically, $G(V_\mu^{\theta_1}, \pi_\mu^{\theta_2}, \eta^{\xi_1}, \varpi^{\xi_2}; \mathcal{D}_i) = \mathbb{E}_T[M(V_\mu^{\theta_1}, \pi_\mu^{\theta_2}, \eta^{\xi_1}, \varpi^{\xi_2}; \mathcal{D}_i) K(\{S_i^t, A_i^t\}, \{\widetilde{S}_i^t, \widetilde{A}_i^t\})$ $\widetilde{M}(V_\mu^{\theta_1}, \pi_\mu^{\theta_2}, \eta^{\xi_1}, \varpi^{\xi_2}; \mathcal{D}_i)]$, where $M(V_\mu^{\theta_1}, \pi_\mu^{\theta_2}, \eta^{\xi_1}, \varpi^{\xi_2}; \mathcal{D}_i) = \mu_{V_\mu^{\theta_1}, \pi_\mu^{\theta_2}}(S_i^t, A_i^t, S_i^{t+1})$ and $\widetilde{M}(V_\mu^{\theta_1}, \pi_\mu^{\theta_2}, \eta^{\xi_1}, \varpi^{\xi_2}; \mathcal{D}_i) = \Lambda_{V_\mu^{\theta_1}, \pi_\mu^{\theta_2}}(\widetilde{S}_i^t, \widetilde{A}_i^t, \widetilde{S}_i^{t+1})$. Next, we proceed to bound the distance in composite space $\mathcal{F}_{\theta,\xi}$. In particular, let $(V_\mu^{\theta_1}, \pi_\mu^{\theta_2}, \eta^{\xi_1}, \varpi^{\xi_2})$, $(V_\mu^{\theta_1}{}', \pi_\mu^{\theta_2}{}', \eta^{\xi_1}{}', \varpi^{\xi_2}{}') \in \Theta_1 \times \Theta_2 \times \Xi_1 \times \Xi_2$ are two arbitrary functions, then the empirical distance w.r.t. $\{\mathcal{D}_i\}_{i=1}^n$ for the two function can be upper bounded by

$$\mathbb{P}_n \left| G(V_\mu^{\theta_1}, \pi_\mu^{\theta_2}, \eta^{\xi_1}, \varpi^{\xi_2}; \mathcal{D}_i) - G(V_\mu^{\theta_1}{}', \pi_\mu^{\theta_2}{}', \eta^{\xi_1}{}', \varpi^{\xi_2}{}'; \mathcal{D}_i) \right|$$

$$=\mathbb{P}_n \Big| \mathbb{E}_T \big[ M(V_\mu^{\theta_1}, \pi_\mu^{\theta_2}, \eta^{\xi_1}, \varpi^{\xi_2}; \mathcal{D}_i) K(\{S^t, A^t\}, \{\widetilde{S}^t, \widetilde{A}^t\}) \widetilde{M}(V_\mu^{\theta_1}, \pi_\mu^{\theta_2}, \eta^{\xi_1}, \varpi^{\xi_2}; \mathcal{D}_i) \big] -$$

$$\mathbb{E}_T \big[ M(V_\mu^{\theta_1}{}', \pi_\mu^{\theta_2}{}', \eta^{\xi_1}{}', \varpi^{\xi_2}{}'; \mathcal{D}_i) K(S^t, A^t; \widetilde{S}^t, \widetilde{A}^t) \widetilde{M}(V_\mu^{\theta_1}{}', \pi_\mu^{\theta_2}{}', \eta^{\xi_1}{}', \varpi^{\xi_2}{}'; \mathcal{D}_i) \big] \Big|$$

$$=\mathbb{P}_n \Big| \mathbb{E}_T \big[ K(S^t, A^t; \widetilde{S}^t, \widetilde{A}^t) \big( M(V_\mu^{\theta_1}, \pi_\mu^{\theta_2}, \eta^{\xi_1}, \varpi^{\xi_2}; \mathcal{D}_i) \widetilde{M}(V_\mu^{\theta_1}, \pi_\mu^{\theta_2}, \eta^{\xi_1}, \varpi^{\xi_2}; \mathcal{D}_i) -$$

$$M(V_\mu^{\theta_1}{}', \pi_\mu^{\theta_2}{}', \eta^{\xi_1}{}', \varpi^{\xi_2}{}'; \mathcal{D}_i) \widetilde{M}(V_\mu^{\theta_1}{}', \pi_\mu^{\theta_2}{}', \eta^{\xi_1}{}', \varpi^{\xi_2}{}'; \mathcal{D}_i) \big) \big] \Big|$$

$$=\mathbb{P}_n \Big\{ \mathbb{E}_T \Big| K(S^t, A^t; \widetilde{S}^t, \widetilde{A}^t) \big( M(V_\mu^{\theta_1}, \pi_\mu^{\theta_2}, \eta^{\xi_1}, \varpi^{\xi_2}; \mathcal{D}_i) \widetilde{M}(V_\mu^{\theta_1}, \pi_\mu^{\theta_2}, \eta^{\xi_1}, \varpi^{\xi_2}; \mathcal{D}_i) -$$

$$M(V_\mu^{\theta_1}{}', \pi_\mu^{\theta_2}{}', \eta^{\xi_1}{}', \varpi^{\xi_2}{}'; \mathcal{D}_i) \widetilde{M}(V_\mu^{\theta_1}{}', \pi_\mu^{\theta_2}{}', \eta^{\xi_1}{}', \varpi^{\xi_2}{}'; \mathcal{D}_i) \big) \Big| \Big\}$$

$$\leq \mathbb{P}_n \Big\{ \mathbb{E}_T \Big| K(S^t, A^t; \widetilde{S}^t, \widetilde{A}^t) \Big| \cdot \mathbb{E}_T \Big| \big( M(V_\mu^{\theta_1}, \pi_\mu^{\theta_2}, \eta^{\xi_1}, \varpi^{\xi_2}; \mathcal{D}_i) + M(V_\mu^{\theta_1}{}', \pi_\mu^{\theta_2}{}', \eta^{\xi_1}{}', \varpi^{\xi_2}{}'; \mathcal{D}_i) \big) \Big| \cdot$$

$$\mathbb{E}_T \Big| \big( \widetilde{M}(V_\mu^{\theta_1}, \pi_\mu^{\theta_2}, \eta^{\xi_1}, \varpi^{\xi_2}; \mathcal{D}_i) - \widetilde{M}(V_\mu^{\theta_1}{}', \pi_\mu^{\theta_2}{}', \eta^{\xi_1}{}', \varpi^{\xi_2}{}'; \mathcal{D}_i) \big) \Big| \Big\}$$

$$\leq M_{\max,1} \Big( \mathbb{P}_n \mathbb{E}_T |\eta^{\xi_1} - \eta^{\xi_1}{}'| + \mu \mathbb{P}_n \mathbb{E}_T |\mathrm{prox}^\circ(\pi_\mu^{\theta_2}) - \mathrm{prox}^\circ(\pi_\mu^{\theta_2}{}')|$$

$$+ \mathbb{P}_n \mathbb{E}_T |(\gamma V_\mu^{\theta_1} - V_\mu^{\theta_1}) - (\gamma V_\mu^{\theta_1}{}' - V_\mu^{\theta_1}{}')| + \mathbb{P}_n \mathbb{E}_T |\psi^\theta - \varpi^{\xi_2}{}'| \Big)$$

$$=M_{\max,1} \big( \mathbb{P}_n \mathbb{E}_T |\eta^{\xi_1} - \eta^{\xi_1}{}'| + \mu \mathbb{P}_n \mathbb{E}_T |\pi_\mu^{\theta_2} - \pi_\mu^{\theta_2}{}'| + (1+\gamma) \mathbb{P}_n \mathbb{E}_T |V_\mu^{\theta_1} - V_\mu^{\theta_1}{}'| + \mathbb{P}_n \mathbb{E}_T |\psi^\theta - \varpi^{\xi_2}{}'| \big)$$

$$\leq M_{\max,1} \big( \mathbb{P}_n \|\eta^{\xi_1} - \eta^{\xi_1}{}'\|_\infty + \mu \mathbb{P}_n \|\pi_\mu^{\theta_2} - \pi_\mu^{\theta_2}{}'\|_\infty + (1+\gamma) \mathbb{P}_n \|V_\mu^{\theta_1} - V_\mu^{\theta_1}{}'\|_\infty + \mathbb{P}_n \|\varpi^{\xi_2} - \varpi^{\xi_2}{}'\|_\infty \big),$$

where $M_{\max,1} = 2M_{\max}$. Therefore, as the proximal parameter $0 \leq \mu \leq \mu_{\max} < \infty$, for any $\varepsilon > 0$ the metric entropy $\log \mathcal{N}((\mu_{\max} + 4)M_{\max,1}\varepsilon, \mathcal{F}_{\theta,\xi}, \{\mathcal{D}_i\}_{i=1}^n)$ can be bound with respect to separate metric entropy of $(\Theta_1, \Theta_2, \Xi_1, \Xi_2)$. Denote $\min(2(\mu_{\max} + 4)M_{\max}, 1)$ as $\widetilde{C}$, then

$$\mathcal{N}\Big( (\mu_{\max} + 4)M_{\max,1}\varepsilon, \mathcal{F}_{\theta,\xi}, \{\mathcal{D}_i\}_{i=1}^n \Big)$$

$$\leq \mathcal{N}\Big( \widetilde{C}\varepsilon, \Theta_1, \{\mathcal{D}_i\}_{i=1}^n \Big) \mathcal{N}\Big( \widetilde{C}\varepsilon, \Theta_2, \{\mathcal{D}_i\}_{i=1}^n \Big) \mathcal{N}\Big( \widetilde{C}\varepsilon, \Xi_1, \{\mathcal{D}_i\}_{i=1}^n \Big) \mathcal{N}\Big( \widetilde{C}\epsilon, \Xi_2, \{\mathcal{D}_i\}_{i=1}^n \Big)$$

To bound these factors, we first introduce a idea of pseudo-dimension, that is, for any set $\mathcal{X}$, any points $x^{1:N} \in \mathcal{X}^N$, any class $\mathcal{F}$ of functions on $\mathcal{X}$ taking values in $[0, C]$ with pseudo-dimension $D_{\mathcal{F}} < \infty$ and any $\epsilon > 0$, we have

$$\mathcal{N}\left( \epsilon, \mathcal{F}, x^{1:N} \right) \leqslant e\left( D_{\mathcal{F}} + 1 \right) \left( \frac{2eC}{\epsilon} \right)^{D_{\mathcal{F}}}$$

Therefore, we have

$$\mathcal{N}\Big( 2(\mu_{\max} + 4)M_{\max}\epsilon, \mathcal{F}_{\theta,\xi}, \{\mathcal{D}_i\}_{i=1}^n \Big)$$

$$\leq e^4 \left( D_{\Theta_1} + 1 \right) \left( D_{\Theta_2} + 1 \right) \left( D_{\Xi_1} + 1 \right) \left( D_{\Xi_2} + 1 \right) \left( \frac{2eM_{\max}}{\widetilde{C}\epsilon} \right)^{D_{\Theta_1} + D_{\Theta_2} + D_{\Xi_1} + D_{\Xi_2}}$$

which implies

$$\mathcal{N}\left(\frac{\epsilon}{2}, \mathcal{F}_{\theta,\xi}, \{\mathcal{D}_i\}_{i=1}^n\right)$$

$$\leq e^4 \left(D_{\Theta_1}+1\right)\left(D_{\Theta_2}+1\right)\left(D_{\Xi_1}+1\right)\left(D_{\Xi_2}+1\right)\left(\frac{8(\mu_{\max}+4)M_{\max}^3 e}{\widetilde{C}\epsilon}\right)^{D_{\Theta_1}+D_{\Theta_2}+D_{\Xi_1}+D_{\Xi_2}}$$

$$:=C_1\left(\frac{1}{\epsilon}\right)^{D_{\mathcal{F}_{\theta,\xi}}}$$

where $C_1 = e^4 \left(D_{\Theta_1}+1\right)\left(D_{\Theta_2}+1\right)\left(D_{\Xi_1}+1\right)\left(D_{\Xi_2}+1\right)\left(\frac{8(\mu_{\max}+4)M_{\max}^3 e}{\widetilde{C}}\right)^{D_{\Theta_1}+D_{\Theta_2}+D_{\Xi_1}+D_{\Xi_2}}$
and $D_{\mathcal{F}_{\theta,\xi}} = D_{\Theta_1} + D_{\Theta_2} + D_{\Xi_1} + D_{\Xi_2}$ i.e., the "effective" psuedo dimension.

Then we apply Pollard tail inequality, for any $n \geq 32/\epsilon^2$, we have

$$\mathbb{P}\left(\sup_{(V_\mu^{\theta_1}, \pi_\mu^{\theta_2}, \eta^{\xi_1}, \varpi^{\xi_2}) \in \Theta_1 \times \Theta_2 \times \Xi_1 \times \Xi_2} \left|\mathbb{E}G(V_\mu^{\theta_1}, \pi_\mu^{\theta_2}, \eta^{\xi_1}, \varpi^{\xi_2}; \mathcal{D}_i) - \mathbb{P}_n G(V_\mu^{\theta_1}, \pi_\mu^{\theta_2}, \eta^{\xi_1}, \varpi^{\xi_2}; \mathcal{D}_i)\right| \geq \frac{\epsilon}{2}\right)$$

$$\leq 8C_1 \left(\frac{1}{\epsilon}\right)^{D_{\mathcal{F}_{\theta,\xi}}} \exp\left(-\frac{n\epsilon^2}{512 M_{\max}^2}\right)$$

Then we can obtain

$$\mathbb{E}\left[\sup_{(V_\mu^{\theta_1}, \pi_\mu^{\theta_2}, \eta^{\xi_1}, \varpi^{\xi_2}) \in \Theta_1 \times \Theta_2 \times \Xi_1 \times \Xi_2} \left|\mathbb{E}G(V_\mu^{\theta_1}, \pi_\mu^{\theta_2}, \eta^{\xi_1}, \varpi^{\xi_2}; \mathcal{D}_i) - \mathbb{P}_n G(V_\mu^{\theta_1}, \pi_\mu^{\theta_2}, \eta^{\xi_1}, \varpi^{\xi_2}; \mathcal{D}_i)\right|^2\right]$$

$$= \int_0^\infty \mathbb{P}\left(\sup_{(V_\mu^{\theta_1}, \pi_\mu^{\theta_2}, \eta^{\xi_1}, \varpi^{\xi_2}) \in \Theta_1 \times \Theta_2 \times \Xi_1 \times \Xi_2} \left|\mathbb{E}G(V_\mu^{\theta_1}, \pi_\mu^{\theta_2}, \eta^{\xi_1}, \varpi^{\xi_2}; \mathcal{D}_i) - \mathbb{P}_n G(V_\mu^{\theta_1}, \pi_\mu^{\theta_2}, \eta^{\xi_1}, \varpi^{\xi_2}; \mathcal{D}_i)\right|^2 \geq t\right)dt$$

$$= \int_0^u \mathbb{P}\left(\sup_{(V_\mu^{\theta_1}, \pi_\mu^{\theta_2}, \eta^{\xi_1}, \varpi^{\xi_2}) \in \Theta_1 \times \Theta_2 \times \Xi_1 \times \Xi_2} \left|\mathbb{E}G(V_\mu^{\theta_1}, \pi_\mu^{\theta_2}, \eta^{\xi_1}, \varpi^{\xi_2}; \mathcal{D}_i) - \mathbb{P}_n G(V_\mu^{\theta_1}, \pi_\mu^{\theta_2}, \eta^{\xi_1}, \varpi^{\xi_2}; \mathcal{D}_i)\right|^2 \geq t\right)dt$$

$$+ \int_u^\infty \mathbb{P}\left(\sup_{(V_\mu^{\theta_1}, \pi_\mu^{\theta_2}, \eta^{\xi_1}, \varpi^{\xi_2}) \in \Theta_1 \times \Theta_2 \times \Xi_1 \times \Xi_2} \left|\mathbb{E}G(V_\mu^{\theta_1}, \pi_\mu^{\theta_2}, \eta^{\xi_1}, \varpi^{\xi_2}; \mathcal{D}_i) - \mathbb{P}_n G(V_\mu^{\theta_1}, \pi_\mu^{\theta_2}, \eta^{\xi_1}, \varpi^{\xi_2}; \mathcal{D}_i)\right|^2 \geq t\right)dt$$

$$\leq u + \int_u^\infty 8C_1 \left(\frac{1}{t}\right)^{D_{\mathcal{F}_{\theta,\xi}}} \exp\left(-\frac{nt^2}{512 M_{\max}^2}\right)dt$$

$$= u + \frac{64 C_1 \left(\frac{1}{u}\right)^{D_{\mathcal{F}_{\theta,\xi}}}}{n} \exp\left(-\frac{nu}{512 M_{\max}^2}\right)$$

With probability $1 - \delta$, minimizing the RHS with respect to $u$, and plug the minimizer in, we have

$$\mathbb{E}\left[\sup_{(V_\mu^{\theta_1}, \pi_\mu^{\theta_2}, \eta^{\xi_1}, \varpi^{\xi_2})} \left|\mathbb{E}G(V_\mu^{\theta_1}, \pi_\mu^{\theta_2}, \eta^{\xi_1}, \varpi^{\xi_2}; \mathcal{D}_i) - \mathbb{P}_n G(V_\mu^{\theta_1}, \pi_\mu^{\theta_2}, \eta^{\xi_1}, \varpi^{\xi_2}; \mathcal{D}_i)\right|^2\right]$$

$$\leq \frac{64 D_{\mathcal{F}_{\theta,\xi}} \log(8C_1\left(\frac{1}{\delta}\right))}{n},$$

where $C_2 = 8C_1$. Therefore, we conclude that, with probability $1 - \delta$, we have

$$\Delta_1 \leq \sqrt{\frac{64 D_{\mathcal{F}_{\theta,\xi}} \log\left(\frac{8C_1}{\delta}\right)}{n}} := \sqrt{\frac{C_3 D_{\mathcal{F}_{\theta,\xi}} \log\left(\frac{8C_1}{\delta}\right)}{n}}$$

Next, we proceed to bound $\Delta_2$. To simply the notation, we denote the U-statistic kernel as

$$\bar{K}(S^t, A^t; \widetilde{S}^t, \widetilde{A}^t) := \Lambda_{V_\mu^{\theta_1}, \pi_\mu^{\theta_2}}(S_i^t, A_i^t, S_i^{t+1}) K\left(S_i^t, A_i^t; \widetilde{S}_i^t, \widetilde{A}_i^t\right) \Lambda_{V_\mu^{\theta_1}, \pi_\mu^{\theta_2}}(\widetilde{S}_i^t, \widetilde{A}_i^t, S_i^{t+1}).$$

By Hoeffding's decomposition of kernel function $\bar{K}(S^t, A^t; \widetilde{S}^t, \widetilde{A}^t)$, there exists kernel functions $\bar{K}_1(S^t, A^t)$ and $\bar{K}_2(S^t, A^t; \widetilde{S}^t, \widetilde{A}^t)$ that $\mathbb{E}_T \bar{K}_1(\widetilde{S}^t, \widetilde{A}^t) = 0$ and $\mathbb{E}_T \bar{K}_2(s, a; \widetilde{S}^t, \widetilde{A}^t) = 0$. The U-statistic $U_T$ can be decomposed into

$$U_T = \mathbb{E}_T[U^\star] + \frac{2}{T}\sum_{t=1}^T \bar{K}_1\left(S^t, A^t\right) + U_{\bar{K}_2} \quad \text{and} \quad \mathbb{E}_T[U_T] = \mathbb{E}_T[U^\star] + \mathbb{E}_T[U_{\bar{K}_2}],$$

where $U_{\bar{K}_2} := U_{\bar{K}_2}(V_\mu^{\theta_1}, \pi_\mu^{\theta_2}, \eta^{\xi_1}, \varpi^{\xi_2})$ is defined similarly as in the proof of Theorem 4.2. The details of the decomposition can be seen in the proof of Theorem 4.2. The term $\Delta_2$ can be immediately decomposed as follows

$$\Delta_2 = \sup_{(V_\mu^{\theta_1}, \pi_\mu^{\theta_2}, \eta^{\xi_1}, \varpi^{\xi_2}) \in \Theta_1 \times \Theta_2 \times \Xi_1 \times \Xi_2} \left| U_T(V_\mu^{\theta_1}, \pi_\mu^{\theta_2}, \eta^{\xi_1}, \varpi^{\xi_2})] - \mathbb{E}_T[U_T(V_\mu^{\theta_1}, \pi_\mu^{\theta_2}, \eta^{\xi_1}, \varpi^{\xi_2})] \right| +$$

$$\sup_{(V_\mu^{\theta_1}, \pi_\mu^{\theta_2}, \eta^{\xi_1}, \varpi^{\xi_2}) \in \Theta_1 \times \Theta_2 \times \Xi_1 \times \Xi_2} \left| \mathbb{E}_T[U^\star(V_\mu^{\theta_1}, \pi_\mu^{\theta_2}, \eta^{\xi_1}, \varpi^{\xi_2})] - \mathbb{E}_T[U_T(V_\mu^{\theta_1}, \pi_\mu^{\theta_2}, \eta^{\xi_1}, \varpi^{\xi_2})] \right|$$

$$= \underbrace{\sup_{(V_\mu^{\theta_1}, \pi_\mu^{\theta_2}, \eta^{\xi_1}, \varpi^{\xi_2}) \in \Theta_1 \times \Theta_2 \times \Xi_1 \times \Xi_2} \left| U_T(V_\mu^{\theta_1}, \pi_\mu^{\theta_2}, \eta^{\xi_1}, \varpi^{\xi_2}) - \mathbb{E}_T[U_T(V_\mu^{\theta_1}, \pi_\mu^{\theta_2}, \eta^{\xi_1}, \varpi^{\xi_2})] \right|}_{\Delta_2^1} +$$

$$\underbrace{\sup_{(V_\mu^{\theta_1}, \pi_\mu^{\theta_2}, \eta^{\xi_1}, \varpi^{\xi_2}) \in \Theta_1 \times \Theta_2 \times \Xi_1 \times \Xi_2} \left| \mathbb{E}_T[U_{\bar{K}_2}(V_\mu^{\theta_1}, \pi_\mu^{\theta_2}, \eta^{\xi_1}, \varpi^{\xi_2})] \right|}_{\Delta_2^2}.$$

Note that the second term is not exactly zero since the samples are weakly dependent. But next, we will show that $\Delta_2^2$ converges to zero. First, we check the conditions of Lemma 3.1 in Arcones & Yu (1994). Observe that $K(\cdot, \cdot) \leq 1$ and according to Lemma S.1, then

$$\sup_{(V_\mu^{\theta_1}, \pi_\mu^{\theta_2}, \eta^{\xi_1}, \varpi^{\xi_2}) \in \Theta_1 \times \Theta_2 \times \Xi_1 \times \Xi_2} \left| \bar{K}\left(S^t, A^t; \widetilde{S}^t, \widetilde{A}^t\right) \right| \leq M_{\max}^2 K\left(S^t, A^t; \widetilde{S}^t, \widetilde{A}^t\right) \leq M_{\max}^2.$$

Therefore, the kernel $\bar{K}$ is a uniformly bounded function. Under Assumption 4.2 that $\beta(m) \lesssim m^{-\delta_1}$ for $\delta_1 > 1$. Therefore, $\beta(m) m^{\delta_1} \to 0$. By using a similar technique of calculating the metric entropy, for any $\epsilon > 0$, we have the covering number that

$$\mathcal{N}\left(\varepsilon, \mathcal{F}_{\theta,\xi}, \|\cdot\|_{L^2}\right) \leq \mathcal{N}\left(\varepsilon, \mathcal{F}_{\theta,\xi}, \{\mathcal{D}_i\}_{i=1}^n\right) < \infty$$

Then the conditions of Lemma 3.1 in Arcones & Yu (1994) are satisfied, we have

$$\sup_{(V_\mu^{\theta_1}, \pi_\mu^{\theta_2}, \eta^{\xi_1}, \varpi^{\xi_2}) \in \Theta_1 \times \Theta_2 \times \Xi_1 \times \Xi_2} |\sqrt{T} U_{\bar{K}_2}(V_\mu^{\theta_1}, \pi_\mu^{\theta_2}, \eta^{\xi_1}, \varpi^{\xi_2})| = o_p(1)$$

$$\implies \sup_{(V_\mu^{\theta_1}, \pi_\mu^{\theta_2}, \eta^{\xi_1}, \varpi^{\xi_2}) \in \Theta_1 \times \Theta_2 \times \Xi_1 \times \Xi_2} |U_{\bar{K}_2}(V_\mu^{\theta_1}, \pi_\mu^{\theta_2}, \eta^{\xi_1}, \varpi^{\xi_2})| = o_p(1). \tag{35}$$

Since $U_{\bar{K}_2}(V_\mu^{\theta_1}, \pi_\mu^{\theta_2}, \eta^{\xi_1}, \varpi^{\xi_2})$ is uniformly bounded, then

$$\sup_{(V_\mu^{\theta_1}, \pi_\mu^{\theta_2}, \eta^{\xi_1}, \varpi^{\xi_2})} |U_{\bar{K}_2}(V_\mu^{\theta_1}, \pi_\mu^{\theta_2}, \eta^{\xi_1}, \varpi^{\xi_2})| < \infty$$

As $x, T \to \infty$, then

$$\mathbb{E}\left[\sup_{(V_\mu^{\theta_1}, \pi_\mu^{\theta_2}, \eta^{\xi_1}, \varpi^{\xi_2})} |U_{\bar{K}_2}(V_\mu^{\theta_1}, \pi_\mu^{\theta_2}, \eta^{\xi_1}, \varpi^{\xi_2})| I\{\sup_{(V_\mu^{\theta_1}, \pi_\mu^{\theta_2}, \eta^{\xi_1}, \varpi^{\xi_2})} |U_{\bar{K}_2}(V_\mu^{\theta_1}, \pi_\mu^{\theta_2}, \eta^{\xi_1}, \varpi^{\xi_2})| > x\}\right] \to 0,$$

which means $\sup_{(V_\mu^{\theta_1}, \pi_\mu^{\theta_2}, \eta^{\xi_1}, \varpi^{\xi_2})} |U_{\bar{K}_2}(V_\mu^{\theta_1}, \pi_\mu^{\theta_2}, \eta^{\xi_1}, \varpi^{\xi_2})|$ is uniformly integrable. Combine with the weak convergence in (35), then as $T \to \infty$

$$\Delta_2^2 = \sup_{(V_\mu^{\theta_1}, \pi_\mu^{\theta_2}, \eta^{\xi_1}, \varpi^{\xi_2})} \mathbb{E}|U_{\bar{K}_2}(V_\mu^{\theta_1}, \pi_\mu^{\theta_2}, \eta^{\xi_1}, \varpi^{\xi_2})| \leq \mathbb{E} \sup_{(V_\mu^{\theta_1}, \pi_\mu^{\theta_2}, \eta^{\xi_1}, \varpi^{\xi_2})} |U_{\bar{K}_2}(V_\mu^{\theta_1}, \pi_\mu^{\theta_2}, \eta^{\xi_1}, \varpi^{\xi_2})| \to 0.$$

Then we move to bound $\Delta_2^1$. The U-statistic $U_T$ is not degenerate, so we adopt Hoeffding's representation (Hoeffding, 1994) such that it reduces the problem to a "first-order" analysis. Specifically, let $\sigma(T)$ is the collection of all permutations of $\{1, 2, ..., T\}$, the U-statistic $U_T$ can be re-expressed as

$$U_T\left(V_\mu^{\theta_1}, \pi_\mu^{\theta_2}, \eta^{\xi_1}, \varpi^{\xi_2}\right) = \frac{1}{T!} \sum_{\sigma(T)} \frac{1}{T_0} \sum_{t=1}^{T_0} \bar{K}\left(X^{\sigma(t)}, X^{\sigma(T_0+t)}\right),$$

where $T_0 = \lfloor T/2 \rfloor$. By the trick, we have the following inequality

$$
\Delta_2^1 = \sup_{(V_\mu^{\theta_1}, \pi_\mu^{\theta_2}, \eta^{\xi_1}, \varpi^{\xi_2})} \left| \frac{2}{T(T-1)} \sum_{1 \le i \ne j \le T} \bar{K}\left(X^i, X^j\right) - \mathbb{E}_T[U_T(V_\mu^{\theta_1}, \pi_\mu^{\theta_2}, \eta^{\xi_1}, \varpi^{\xi_2})] \right|
$$

$$
= \sup_{(V_\mu^{\theta_1}, \pi_\mu^{\theta_2}, \eta^{\xi_1}, \varpi^{\xi_2})} \left| \frac{1}{T!} \sum_{\sigma(T)} \frac{1}{T_0} \sum_{t=1}^{T_0} \bar{K}\left(X^{\sigma(t)}, X^{\sigma(T_0+t)}\right) - \mathbb{E}_T[U_T(V_\mu^{\theta_1}, \pi_\mu^{\theta_2}, \eta^{\xi_1}, \varpi^{\xi_2})] \right|
$$

$$
= \sup_{(V_\mu^{\theta_1}, \pi_\mu^{\theta_2}, \eta^{\xi_1}, \varpi^{\xi_2})} \left| \frac{1}{T!} \sum_{\sigma(T)} \frac{1}{T_0} \sum_{i=t}^{T_0} \bar{K}\left(X^{\sigma(t)}, X^{\sigma(T_0+t)}\right) - \frac{1}{T!} \sum_{\sigma(T)} \mathbb{E}_T[U_T(V_\mu^{\theta_1}, \pi_\mu^{\theta_2}, \eta^{\xi_1}, \varpi^{\xi_2})] \right|
$$

$$
\le \sup_{(V_\mu^{\theta_1}, \pi_\mu^{\theta_2}, \eta^{\xi_1}, \varpi^{\xi_2})} \frac{1}{T!} \sum_{\sigma(T)} \left| \frac{1}{T_0} \sum_{t=1}^{T_0} \bar{K}\left(X^{\sigma(t)}, X^{\sigma(T_0+t)}\right) - \mathbb{E}_T[U_T(V_\mu^{\theta_1}, \pi_\mu^{\theta_2}, \eta^{\xi_1}, \varpi^{\xi_2})] \right|
$$

$$
\le \frac{1}{T!} \sum_{\sigma(T)} \mathbb{E}_T \sup_{(V_\mu^{\theta_1}, \pi_\mu^{\theta_2}, \eta^{\xi_1}, \varpi^{\xi_2})} \left| \frac{1}{T_0} \sum_{t=1}^{T_0} \bar{K}\left(X^{\sigma(t)}, X^{\sigma(T_0+t)}\right) - \mathbb{E}_T[U_T(V_\mu^{\theta_1}, \pi_\mu^{\theta_2}, \eta^{\xi_1}, \varpi^{\xi_2})] \right|
$$

$$
= \sup_{(V_\mu^{\theta_1}, \pi_\mu^{\theta_2}, \eta^{\xi_1}, \varpi^{\xi_2})} \left| \frac{1}{T_0} \sum_{t=1}^{T_0} \bar{K}\left(X^t, X^{(T_0+t)}\right) - \mathbb{E}_T[U_T(V_\mu^{\theta_1}, \pi_\mu^{\theta_2}, \eta^{\xi_1}, \varpi^{\xi_2})] \right|
$$

$$
= \sup_{(V_\mu^{\theta_1}, \pi_\mu^{\theta_2}, \eta^{\xi_1}, \varpi^{\xi_2})} \left| \frac{1}{T_0} \sum_{t=1}^{T_0} \bar{K}\left(X^t, X^{(T_0+t)}\right) - \mathbb{E}_T\left[ \frac{1}{T_0} \sum_{t=1}^{T_0} \bar{K}\left(X^t, X^{(T_0+t)}\right) \right] \right|
$$

$$
= \sup_{(V_\mu^{\theta_1}, \pi_\mu^{\theta_2}, \eta^{\xi_1}, \varpi^{\xi_2})} \left| \frac{1}{T_0} \sum_{t=1}^{T_0} \bar{G}\left(\widetilde{X}^t\right) - \mathbb{E}_T\left[ \frac{1}{T_0} \sum_{t=1}^{T_0} \bar{G}\left(\widetilde{X}^t\right) \right] \right|,
$$

where $\widetilde{X}^t = \left(X^t, X^{(T_0+t)}\right)$ which itself is a two-dimensional stationary sequences under mixing condition. Note that the last term is the expectation of the suprema of the empirical process $1/T_0 \sum_{t=1}^{T_0} \bar{G}(\widetilde{X}^t) - \mathbb{E}_T[1/T_0 \sum_{t=1}^{T_0} \bar{G}(\widetilde{X}^t)]$ on the space $\bar{\mathcal{G}}_{\theta,\xi}$. The distance in $\bar{\mathcal{G}}_{\theta,\xi}$ can be bounded by the following,

$$
\mathcal{N}\left( \min\{(2\mu_{\max} + 4)M_{\max}, 1\}\varepsilon, \bar{\mathcal{G}}_{\theta,\xi}, \{\widetilde{X}^t\}_{t=1}^{T_0} \right)
$$

$$
\le \mathcal{N}\left( \widetilde{C}\varepsilon, \Theta_1, \{\mathcal{D}_i\}_{i=1}^n \right) \mathcal{N}\left( \widetilde{C}\varepsilon, \Theta_2, \{\mathcal{D}_i\}_{i=1}^n \right) \mathcal{N}\left( \widetilde{C}\epsilon, \Xi_1, \{\mathcal{D}_i\}_{i=1}^n \right) \mathcal{N}\left( \widetilde{C}\epsilon, \Xi_2, \{\mathcal{D}_i\}_{i=1}^n \right)
$$

$$
= e^4 \left(D_{\Theta_1} + 1\right) \left(D_{\Theta_2} + 1\right) \left(D_{\Xi_1} + 1\right) \left(D_{\Xi_2} + 1\right) \left( \frac{2eM_{\max}}{\widetilde{C}\epsilon} \right)^{D_{\Theta_1} + D_{\Theta_2} + D_{\Xi_1} + D_{\Xi_2}}
$$

which implies

$$
\mathcal{N}\left( \frac{\epsilon}{16}, \bar{\mathcal{G}}_{\theta,\xi}, \{\widetilde{X}^t\}_{t=1}^{T_0} \right)
$$

$$
\le e^4 \left(D_{\Theta_1} + 1\right) \left(D_{\Theta_2} + 1\right) \left(D_{\Xi_1} + 1\right) \left(D_{\Xi_2} + 1\right) \left( \frac{64(_{\max} + 32)U_{\max}^2 e}{\widetilde{C}\epsilon} \right)^{D_{\Theta_1} + D_{\Theta_2} + D_{\Xi_1} + D_{\Xi_2}}
$$

$$
:= C_3 \left( \frac{1}{\epsilon} \right)^{D_{\bar{\mathcal{G}}_{\theta,\xi}}}
$$

where $D_{\bar{\mathcal{G}}_{\theta,\xi}} = D_{\Theta_1} + D_{\Theta_2} + D_{\Xi_1} + D_{\Xi_2}$. First, without loss of generality, let $T_0 = 2m_{T_0} k_{T_0}$ for appropriate positive integers $m_{T_0} k_{T_0}$ as in (Yu, 1994). Follow Lemma 5 in Antos et al. (2008), we obtain that

$$
\mathbb{P}\left( \sup_{(V_\mu^{\theta_1}, \pi_\mu^{\theta_2}, \eta^{\xi_1}, \varpi^{\xi_2})} \left| \frac{1}{T_0} \sum_{t=1}^{T_0} \bar{G}\left(\widetilde{X}^t\right) - \mathbb{E}_T\left[ \frac{1}{T_0} \sum_{t=1}^{T_0} \bar{G}\left(\widetilde{X}^t\right) \right] \right| \ge \frac{\epsilon}{2} \right)
$$

$$
\le C_3 \left( \frac{1}{\epsilon} \right)^{D_{\bar{\mathcal{G}}_{\theta,\xi}}} \exp\left( -4C_4 m_{T_0} \epsilon^2 \right) + 2m_{T_0} \beta(k_{T_0})
$$

where $C_4 = \frac{1}{2}\left(\frac{1}{8M_{\max}^2}\right)^2$. If $D_{\bar{\mathcal{G}}} \geq 2$, and let $\beta(m) \lesssim \exp(-\delta_1 m), T \geq 1, m_T = \left\lceil (C_4 T_0 \epsilon^2/\delta_1)^{\frac{1}{2}} \right\rceil, m_{T_0} = T_0/(2k_{T_0})$, where $D_{\bar{\mathcal{G}}_{\theta,\xi}} \geq 2, C_3, C_4, \delta_1$, we apply Lemma 14 in Antos et al. (2008), then

$$2m_{T_0}\beta_{k_{T_0}} + C_1\left(\frac{1}{\epsilon}\right)^{D_{\bar{\mathcal{G}}_{\theta,\xi}}}\exp\left(-4C_2 m_{T_0}\epsilon^2\right) \leq \delta$$

and we have, with probability $1 - \delta$,

$$\Delta_2^1 \leq \sqrt{\frac{2\Delta(\Delta/\delta_1 \vee 1)}{C_4 T_0}}$$

$$\implies \Delta_2^1 \leq \sqrt{\frac{2\Delta(\Delta/\delta_1 \vee 1)}{C_4 \lfloor T/2 \rfloor}}$$

where

$$\Delta = (D_{\bar{\mathcal{G}}_{\theta,\xi}}/2)\log T_0 + \log(e/\delta) + \log^+\left(C_3 C_4^{D_{\bar{\mathcal{G}}_{\theta,\xi}}/2}\right)$$

$$\implies \Delta = (D_{\bar{\mathcal{G}}_{\theta,\xi}}/2)\log(T/2) + \log(e/\delta) + \log^+\left(C_3 C_4^{D_{\bar{\mathcal{G}}_{\theta,\xi}}/2}\right)$$

Now, we conclude that

$$\|\widehat{V}_\mu^{\theta_1,k} - V^*\|_{L^2}^2 \leq \frac{C_1}{\kappa_{\min}(1-\gamma)^2}\left(\sqrt{\frac{C_3 D \log\left(\frac{8C_1}{\delta}\right)}{n}} + \sqrt{\frac{2\Delta(\Delta/\delta_1 \vee 1)}{C_4\lfloor T/2\rfloor}}\right) +$$

$$C_2\frac{\mu^2(\mathbf{C} + |1-\mathbf{C}| \vee 1)^2}{(1-\gamma)^2} + C_5\left\|\widehat{V}_\mu^{\theta_1} - \widehat{V}_\mu^{\theta_1,k}\right\|_{L^2}^2 + \epsilon_{\text{approximation error}}$$

where $\Delta = (D_{\bar{\mathcal{G}}_{\theta,\xi}}/2)\log(\lfloor T/2\rfloor) + \log(e/\delta) + \log^+\left(C_3 C_4^{D_{\bar{\mathcal{G}}_{\theta,\xi}}}/2\right)$, $D_{\bar{\mathcal{G}}_{\theta,\xi}} = \text{P-dim}(\Theta_1) + \text{P-dim}(\Theta_2) + \text{P-dim}(\Xi_1) + \text{P-dim}(\Xi_2)$, and $C_1, ..., C_5$ are some constants. Adapt the notations for the constants number from Theorem 4.2. By some algebra, we conclude that

$$\|\widehat{V}_\mu^{\theta_1,k} - V^{\pi^*}\|_{L^2}^2 \leq \underbrace{\frac{C_4}{\kappa_{\min}(1-\gamma)^2}\left(\sqrt{\frac{C_5 D_{\text{P-dim}}\log\left(\frac{8C_4}{\delta}\right)}{n}} + \sqrt{\frac{2(\frac{\bar{\Delta}}{\delta_1}\vee 1)\bar{\Delta}}{C_6\lfloor T/2\rfloor}}\right)}_{\text{generalization error}} +$$

$$\underbrace{C_7\frac{\mu^2(\mathbf{C} + |1-\mathbf{C}| \vee 1)^2}{(1-\gamma)^2}}_{\text{proximal bias}} + \underbrace{C_8\left\|\widehat{V}_\mu^{\theta_1} - \widehat{V}_\mu^{\theta_1,k}\right\|_{L^2}^2}_{\text{optimization error}} + \epsilon_{\text{approximation error}}$$

where $\bar{\Delta} = \frac{D_{\text{P-dim}}\log(\lfloor T/2\rfloor)}{2} + \log(\frac{e}{\delta}) + \log^+\left(\frac{C_5 C_6^{D_{\text{P-dim}}}}{2}\right)$, $D_{\text{P-dim}} = \text{P-dim}(\Theta_1) + \text{P-dim}(\Theta_2) + \text{P-dim}(\Xi_1) + \text{P-dim}(\Xi_2)$, and $C_4, ..., C_8$ are some constants. $\qquad\square$

## B.9 PROOF OF THEOREM 4.4

We note that SGD converges has a global convergence to a stationary point with a sublinear rate in the case of convexity. However, the resulting dose not typically holds for the non-convex analysis. The intuition behind the proof is that our quasi-optimal algorithm can be regarded as a special case of the randomized stochastic descent (RSD) algorithm for solving the non-convex minimization problem.

The convergence analysis of for randomized stochastic descent algorithm has been established in Corollary 2.2 of (Ghadimi & Lan, 2013). That is, RSD is provably convergent to a stationary point. Follow Theorem 3 in (Drori & Shamir, 2020), an unbiased SGD algorithm, i.e., the quasi-optimal

algorithm with diminishing learning rate and evaluated on Euclidean distance. Therefore, it suffices to show that the gradient of the loss is unbiased.

Now we show that the gradient is unbiased, as follows

$$
\begin{aligned}
\nabla_{\theta_1}\mathcal{L}_U = \mathbb{E}\Big[ &\nabla_{\theta_1}(\gamma V_\mu^{\theta_1}(S^{t+1}) - V_\mu^{\theta_1}(S^t))K(S^t, A^t; \tilde{S}^t, \tilde{A}^t)\Lambda_{V_\mu, \pi_\mu}(\tilde{S}^t, \tilde{A}^t, \tilde{S}^{t+1}) \\
&+ \Lambda_{V_\mu, \pi_\mu}(S^t, A^t, S^{t+1})K(S^t, A^t; \tilde{S}^t, \tilde{A}^t)\nabla_{\theta_1}(\gamma V_\mu^{\theta_1}(\tilde{S}^{t+1}) - V_\mu^{\theta_1}(\tilde{S}^t))\Big]
\end{aligned}
$$

$$
\begin{aligned}
\nabla_{\theta_2}\mathcal{L}_U = \mathbb{E}\Big[ &- 2\mu(\nabla_{\theta_2}\pi_\mu^{\theta_2}(A^t|S^t))K(S^t, A^t; \tilde{S}^t, \tilde{A}^t)\Lambda_{V_\mu, \pi_\mu}(\tilde{S}^t, \tilde{A}^t, \tilde{S}^{t+1}) \\
&- \Lambda_{V_\mu, \pi_\mu}(S^t, A^t, S^{t+1})K(S^t, A^t; \tilde{S}^t, \tilde{A}^t)(2\mu(\nabla_{\theta_2}\pi_\mu^{\theta_2}(\tilde{A}^t|S^t)))\Big]
\end{aligned}
$$

$$
\begin{aligned}
\nabla_{\xi_1}\mathcal{L}_U = \mathbb{E}\Big[ &\nabla_{\xi_1}\varpi(S^t, A^t)K(S^t, A^t; \tilde{S}^t, \tilde{A}^t)\Lambda_{V_\mu, \pi_\mu}(\tilde{S}^t, \tilde{A}^t, \tilde{S}^{t+1}) \\
&+ \Lambda_{V_\mu, \pi_\mu}(S^t, A^t, S^{t+1})K(S^t, A^t; \tilde{S}^t, \tilde{A}^t)\nabla_{\xi_1}\varpi(\tilde{S}^t, \tilde{A}^t)\Big]
\end{aligned}
$$

$$
\begin{aligned}
\nabla_{\xi_2}\mathcal{L}_U = \mathbb{E}\Big[ &- \nabla_{\xi_2}\eta(S^t)K(S^t, A^t; \tilde{S}^t, \tilde{A}^t)\Lambda_{V_\mu, \pi_\mu}(\tilde{S}^t, \tilde{A}^t, \tilde{S}^{t+1}) \\
&- \Lambda_{V_\mu, \pi_\mu}(S^t, A^t, S^{t+1})K(S^t, A^t; \tilde{S}^t, \tilde{A}^t)\nabla_{\xi_2}\eta(\tilde{S}^t)\Big]
\end{aligned}
$$

We conclude that the gradient estimator is unbiased. Follow Theorem 3 in (Drori & Shamir, 2020), under the conditions stated in Theorem 4.4, we adapt Corollary 2.2 to our quasi-optimal algorithm, it completes the proof.

## C  PRACTICAL IMPLEMENTATION

In practice, $\{V_\mu^*, \pi_\mu^*, \eta, \varpi\}$ needs to be parameterized for practical implementation. However, noticing that $V_\mu^*$ and $\pi_\mu^*$ are both associated with $Q_\mu^*$ with closed-form expressions (3)(4). Thus, we propose to represent $(V_\mu^*, \pi_\mu^*)$ by modeling $Q_\mu^*$. Additionally, by modeling $Q_\mu^*$ as a quadratic function, the induced policy would follow a $q$-Gaussian distribution. Therefore, we model the coefficients associated with the quadratic form as a linear combination of basis function $\varphi(s)$ such that $Q_\mu^*(s, a; \theta) = -\exp\{\theta_1^T\varphi(s)\}a^2 + \theta_2^T\varphi(s)a + \theta_3^T\varphi(s)$, where $\varphi(s) = [\varphi_1(s), \varphi_2(s), ..., \varphi_m(s)]^T$ is the $m$-dimensional basis function, and $\theta = [\theta_1, \theta_2, \theta_3]^T$ is the $3m$-dimensional parameters we need to estimate. The advantage of such parametrization lies in that the parameter space could be reduced.

To solve the constrained optimization problem, we propose a computationally efficient algorithm by transforming the original constrained optimization problem into an unconstrained minimization problem. Specifically, we impose restrictions on the representation of Lagrangian multipliers $(\eta(s), \varpi(s, a))$ so that they satisfy their constraints automatically. Although such re-parametrization may sacrifice model flexibility, it gains great computational advantage as the unconstrained optimization problem would be much simpler. To be specific, we parametrize $\varpi$ as

$$
\varpi(s, a; \theta) = \max\left(0, -\frac{Q_\mu^*(s, a; \theta)}{2\mu} + \frac{\int_{a \in \mathcal{W}_1(s)} Q_\mu^*(s, a; \theta)da}{2\mu\sigma(\mathcal{W}_s)} - \frac{1}{\sigma(\mathcal{W}_s)}\right), \tag{36}
$$

Therefore, $\varpi(S^t, A^t) \geq 0$ and $\pi_\mu^*(A^t|S^t) \cdot \varpi(S^t, A^t) = 0$ are automatically satisfied. Also, by specifying the expression of Lagrangian multipliers, $\varpi(s, a)$ share the same set of parameters $\theta$ as $(V_\mu^*, \pi_\mu^*)$. We also define

$$
\eta(s; \xi) = \left\{\frac{-\mu\mathbf{C}}{1 + \exp(-k_0(\xi^T s - b_0))}\right\}, \tag{37}
$$

where $b_0$ is the sigmoid's midpoint and $k_0$ is the logistic growth rate. By flipping the sigmoid function to parametrize $\eta(s; \xi)$, the constraint $\eta(s) \in [-\mu\mathbf{C}, 0]$ is also automatically satisfied.

# D   EXPERIMENT DETAILS AND ADDITIONAL RESULTS

For the reproducing purpose, we include our code for all the experiments and the guideline for access to the Ohio Type I Diabetes dataset in an anonymous GitHub link `https://anonymous.4open.science/r/Quasi-optimal-Learning-with-Continuous-Treatments-9B88`.

## D.1   DETAILS OF SIMULATION SETTINGS AND REAL DATA ANALYSIS

The details of the data generative model of each environment in Section 6 are stated below:

**Environment I:** We consider a bounded action space where $\mathcal{A} = [0, 1]$, and a 2-dimensional state space. $A_i^t \overset{iid}{\sim} \text{Unif}(0, 1)$, the state transition function is defined as
$S_{i,1}^{t+1} = \frac{1-\exp(-A_i^t)}{1+\exp(-A_i^t)} S_{i,1}^t + 0.25 S_{i,1}^t S_{i,2}^t + \epsilon_{i,1}^t, \ S_{i,2}^{t+1} = -\frac{1-\exp(-A_i^t)}{1+\exp(-A_i^t)} S_{i,2}^t + 0.25 S_{i,1}^t S_{i,2}^t + \epsilon_{i,2}^t,$
where $\epsilon_{i,1}^t, \epsilon_{i,2}^t \overset{iid}{\sim} N(0, 0.5^2)$, and the reward function is defined as

$$R_i^t = 3 \left( -\exp(S_{i,1}^{t+1} - S_{i,2}^{t+1})(A_i^t)^2 + (S_{i,1}^{t+1} + S_{i,2}^{t+1} + 0.5)A_i^t + S_{i,1}^{t+1} + S_{i,2}^{t+1} \right).$$

**Environment II:** We consider a bounded action space where $\mathcal{A} = [0, 1]$, and a 2-dimensional state space. $A_i^t \overset{iid}{\sim} \text{Unif}(0, 1)$, the state transition function is defined as
$S_{i,1}^{t+1} = 0.75(2A_i^t - 1) \cdot S_{i,1}^t + 0.25 S_{i,1}^t S_{i,2}^t + \epsilon_{i,1}^t, \ S_{i,2}^{t+1} = 0.75(1 - 2A_i^t)S_{i,2}^t + 0.25 S_{i,1}^t S_{i,2}^t + \epsilon_{i,2}^t.$
where $\epsilon_{i,1}^t, \epsilon_{i,2}^t \overset{i.i.d}{\sim} N(0, 0.5^2)$, and $R_i^t = 0.25(S_{i,1}^{t+1})^3 + 2S_{i,1}^{t+1} + 0.5(S_{i,2}^{t+1})^3 + S_{i,2}^{t+1} + 0.25(2A_i^t - 1)$.

**Environment III:** We consider an unbounded action space where $\mathcal{A} = (-\infty, \infty)$, and a 8-dimensional state space. We sampled action uniformly from a bounded space, $A_i^t \overset{iid}{\sim} \text{Unif}(-100, 100)$, while it is allowed to select actions on $\mathbb{R}$ for the learned policy. The state transition function is defined as, $S_i^{t+1} \sim N(\mu_i^{t+1}, \Sigma)$, where $\Sigma$ is a pre-specified covariance matrix, and $\mu_i^t = [\mu_{i,1}^t, ..., \mu_{i,8}^t]$,

$$\mu_{i,j}^{t+1} = \frac{\exp(A_i^t/100 + \mu_{i,j}^t) - \exp(-(A_i^t/100 + \mu_{i,j}^t))}{\exp(A_i^t/100 + \mu_{i,j}^t) + \exp(-(A_i^t/100 + \mu_{i,j}^t))} \quad \text{for} \quad j = 1, 2, 3, 4,$$

$$\mu_{i,j}^{t+1} = \frac{\exp(-A_i^t/100 + \mu_{i,j}^t) - \exp(-(-A_i^t/100 + \mu_{i,j}^t))}{\exp(-A_i^t/100 + \mu_{i,j}^t) + \exp(-(-A_i^t/100 + \mu_{i,j}^t))} \quad \text{for} \quad j = 5, 6, 7, 8.$$

$R_i^t = -\exp(S_{i,1}^{t+1}/2 + S_{i,5}^{t+1}/2)(A_i^t/100)^2 + 2(S_{i,2}^{t+1} + S_{i,3}^{t+1} + S_{i,6}^{t+1} + S_{i,7}^{t+1} + 0.5)A_i^t/100 + S_{i,4}^{t+1} + S_{i,8}^{t+1}$.

**Environment IV:** This environment shares the same transition kernel as Environment III, the only difference is the reward function here is

$R_i^t = (S_{i,1}^{t+1}/2)^3 + (S_{i,2}^{t+1}/2)^3 + S_{i,3}^{t+1} + S_{i,4}^{t+1} + 2[(S_{i,5}^{t+1}/2)^3 + (S_{i,6}^{t+1}/2)^3] + 0.5(S_{i,7}^{t+1} + S_{i,8}^{t+1}).$

For all four environments, we consider different sample sizes where the number of trajectories $n = \{25, 50\}$, and the length of each trajectory $T = \{24, 36\}$. The discount factor $\gamma$ is set to 0.9.

**Motivation of Synthetic experiment design:** We aim to test the performance of our proposed method on the settings of bounded and unbounded continuous action space with unimodal and multimodal reward functions. The motivation for testing the proposed method in bounded action space is to test if the proposed method could potentially handle the off-support bias, as illustrated in Figure 2. The reason for considering a multimodal synthetic environment is to evaluate the quasi-optimal policy class (q-Gaussian policy class) works in a relatively complex situation. Especially for the q-Gaussian policy distribution which is unimodal, it is necessary to test if the q-Gaussian policy still works and is robust to the scenario where the optimal policy might be multimodally behaving.

We make a summary of the synthetic experiments as follows:

Environment I:

- Setting: Bounded action space and unimodal reward function

- Purpose: To evaluate if the quasi-optimal learning works in the scenario where it might suffer the off-support bias issue as the continuous action space is bounded.

Environment II:

- Setting: Bounded action space and multimodal reward function
- Purpose: In addition to the purpose in Environment I, we aim to implement quasi-optimal learning in a more challenging environment. Also, this is for evaluating the robustness of the unimodal q-Gaussian policy under the scenario that the true optimal policy follows a multimodal probability distribution.

Environment III:

- Setting: High-dimension state space and well-separated reward function. The design of the well-separated reward function causes the effect that the selection of non-optimal or sub-optimal actions greatly damages the rewards and increases the risk.
- Purpose: To evaluate the reliability/safety of quasi-optimal learning. We aim to examine if quasi-optimal learning could perform well in this scenario. As we expect quasi-optimal learning is able to identify the quasi-optimal sub-regions and avoids choosing those non-optimal/sub-optimal actions which greatly damage the performance.

Environment IV:

- Setting: High-dimension state space and complex well-separated reward function.
- Purpose: In addition to the purpose in Environment III, we target to evaluate the quasi-optimal learning in a more complex environment, imposing great challenges on recovering the quasi-optimal regions for the proposed method. Indeed, imposing more complex structures on reward function indicates imposing difficulties on value function learning and thus imposes great challenges on identifying quasi-optimal regions.

**Ohio Type 1 Diabetes Dataset:** For individuals in the first cohort, we treat glucose level , carbonhydrate intake, and acceleration level as state variables, i.e., $S_{i,1}^t, S_{i,2}^t$ and $S_{i,3}^t$ . For individuals in the second cohort, heart rate is used instead of acceleration level as $S_{i,3}^t$. The reward function is defined as

$$R_i^t = -\frac{\mathbb{1}(S_{i,1}^t > 140)(S_{i,1}^t - 140)^{1.1} + \mathbb{1}(S_{i,1}^t < 80)(S_{i,1}^t - 80)^2}{30}.$$

### D.2 ADDITIONAL EXPERIMENT DETAILS

In our implementation, since the objective function, $\hat{\mathcal{L}}_U$ may not be convex with respect to $(\theta, \xi)$. We determine the initial point by randomly generating 200 initial values for all parameters and selecting the one with the smallest objective function value.

For the discretization-based methods, i.e., Greedy-GQ and V-learning, we discretize the original action space into 20 bins for implementation in synthetic experiments and 14 bins for real data analysis. The number of bins is chosen by analyzing the distribution of action and the scale of rewards, where too few bins could not lead to an accurate approximation of the whole dynamic, and too many bins may damage the performance of these methods. We use a radial basis to approximate value functions for these two methods based on the recommendation of the original implementation (Ertefaie & Strawderman, 2018; Luckett et al., 2019).

For the DeepRL-based continuous control methods, i.e., DDPG, SAC, BEAR, CQL and IQN, we implement them mainly based on well-known offline deep reinforcement learning library (Seno & Imai, 2021). For the general optimization and function approximation settings, we use a multi-layer perceptron (MLP) with 2 hidden layers, each with 32 nodes for function approximation. We set the batch size to be 64, and use ReLU function as the activation function. In addition to the summary provided below, the initial learning rate is chosen from the set $\{3 \times 10^{-4}, 1 \times 10^{-4}, 3 \times 10^{-5}\}$. We use Adam (Kingma & Ba, 2014) as the optimizer for learning the neural network parameters. We set the discounted factor to be $\gamma = 0.9$ for all experiments.

To evaluate the policy obtained from the proposed method in synthetic experiments, we generate 100 independent trajectories, each with a length of 100 based on the learned policy. We use rejection sampling (Robert et al., 1999) to randomly sample each action by the induced density $\pi_\mu(a|s)$ and calculate the discounted sum of reward for each trajectory. We compare the discounted return of each method. The boxplot of synthetic experiments results based on 50 runs is presented in Figure 3.

For real data analysis, since the data-generating process is unknown, we follow Luckett et al. (2020) to utilize the Monte Carlo approximation of the estimated V-function of the initial state of each trajectory to evaluate the performance of each method. To better evaluate the stability and performance of each method, we randomly select 10 or 20 trajectories from each individual based on available trajectories 50 times and apply all methods to the selected data. The baseline refers to the observed discounted return. The mean and standard deviation of the improvements on the Monto Carlo discounted returns are presented in Table 1.

We report all hyperparameters used in training and additional experiment results in this section. The value of $\mu$ is selected from the set $\{0.01, 0.05, 0.1, 0.2, 0.3, 0.5\}$. We select $\mu$ by cross-validation for each experiment, specifically we select $\mu$ with the largest fitted V-function value on the initial states of each trajectory, i.e., $\mathbb{P}_n\hat{V}_\mu(S_i^1) - (1-\gamma)^{-1}\mu$, where we mitigate the effect of the threshold parameter $\mu$. In our implementation, we set $\mathbf{C} = 5$ for all synthetic experiments and real data analysis, and check that the induced policy $\pi_\mu$ never reaches the boundary value.

We set the learning rate $\alpha_j$ for the $j$th iteration is be $\frac{\alpha_0}{1+d\sqrt{j}}$, where $\alpha_0$ is the learning rate of the initial iteration, and $d$ is the decay rate of the learning rate. When $n = 25$, we set the batch size to be 5, and when $n = 50$, we set the batch size to be 7. We use the $L_2$ distance of iterative parameters as the stopping criterion for the SGD algorithm. The $\mu$ selected for each experiment, along with the learning rates and their descent rates, are shown in Table 2 3 and 4.

Table 2: Hyperparameters for each synthetic environment

| Hyperparameters | Environment I | Environment II | Environment III | Environment IV |
|---|---|---|---|---|
| $\mu$ | 0.1 | 0.05 | 0.05 | 0.05 |
| Learning Rate | 0.002 | 0.0005 | $10^{-5}$ | $10^{-5}$ |
| Descent Rate | $10^{-4}$ | $10^{-4}$ | $10^{-4}$ | $10^{-4}$ |

Table 3: Hyperparameters for Ohio Type I Diabetes Analysis (Cohort I)

| Patient ID | 540 | 544 | 552 | 567 | 584 | 596 |
|---|---|---|---|---|---|---|
| $\mu$ | 0.1 | 0.1 | 0.1 | 0.05 | 0.05 | 0.1 |
| Learning Rate | 0.001 | 0.001 | 0.001 | 0.0005 | 0.001 | 0.02 |
| Descent Rate | $10^{-4}$ | $10^{-4}$ | $10^{-4}$ | $10^{-4}$ | $10^{-4}$ | $2 \times 10^{-4}$ |

Table 4: Hyperparameters for Ohio Type I Diabetes Analysis (Cohort II)

| Patient ID | 559 | 563 | 570 | 575 | 588 | 591 |
|---|---|---|---|---|---|---|
| $\mu$ | 0.2 | 0.1 | 0.2 | 0.05 | 0.05 | 0.05 |
| Learning Rate | 0.005 | 0.0001 | 0.005 | 0.0001 | 0.0001 | 0.0001 |
| Descent Rate | $10^{-4}$ | $10^{-4}$ | $10^{-4}$ | $10^{-4}$ | $10^{-4}$ | $10^{-4}$ |

Table 5: The mean running time in seconds of each method over 50 experiment runs in Environment I. The synthetic experiments are conducted on a single `2.3 GHz Dual-Core Intel Core i5` CPU
.

| n | T | Proposed | SAC | DDPG | BEAR | Greedy-GQ |
|---|---|---|---|---|---|---|
| 25 | 24 | 23.11 | 22.17 | 14.31 | 35.42 | 11.39 |
|  | 36 | 28.91 | 28.12 | 18.35 | 42.16 | 14.47 |
| 50 | 24 | 28.88 | 29.91 | 19.42 | 46.73 | 15.62 |
|  | 36 | 45.23 | 44.46 | 36.82 | 63.54 | 24.81 |

Table 6: The mean running time in seconds of each method over 50 experiment runs in Environment II. The synthetic experiments are conducted on a single `2.3 GHz Dual-Core Intel Core i5` CPU
.

| n | T | Proposed | SAC | DDPG | BEAR | Greedy-GQ |
|---|---|---|---|---|---|---|
| 25 | 24 | 30.93 | 27.29 | 20.12 | 44.01 | 14.56 |
| | 36 | 39.34 | 36.91 | 26.43 | 52.86 | 19.43 |
| 50 | 24 | 41.12 | 42.25 | 28.42 | 55.17 | 21.56 |
| | 36 | 60.16 | 56.47 | 45.71 | 72.12 | 32.14 |

### D.3 ADDITIONAL EXPERIMENT RESULTS

### D.3.1 SENSITIVITY ANALYSES

To validate the cross-validation procedure in practice and analyze the effect of $\mu$ on model performance, we conduct sensitivity analyses for the change of $\mu$. Results are summerized in Figure 4. This confirms that the cross-validation procedure indeed selects a proper $\mu$ which maximizes the discounted return.

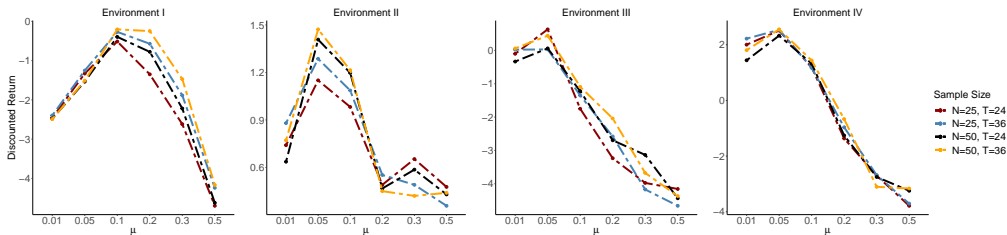

Figure 4: The sensitivity analyses of $\mu$ over 50 repeated experiments

### D.3.2 DISTRIBUTION EVALUATION CRITERION

To measure the performance on safety, we aim to evaluate the distribution of Monte-Carlo discounted sum of rewards for each roll-out trajectory Dabney et al. (2018a), instead of its empirical mean, i.e., discounted return.

In particular, we generate 100 trajectories under the learned policy and record the discounted sum of rewards of each single trajectory. Then we draw the density plots in Figure 5 for all four environments. As shown in Figure 5, the distribution of the quasi-optimal learning shows a thinner tail on the left. This is aligned to two safe RL algorithms IQN and CQL. The phenomenon indicates that there is less chance to enter a low reward trajectory which is damaged by allocating highly-risk actions. However, the non-safe RL approach SAC is more evenly distributed on both extremes; Hence, SAC may enter a low reward trajectory with higher probability (heavier left tail) compared to the quasi-optimal learning and two safe RL baselines. This validates that quasi-optimal learning can avoid risky actions as the other two safe RL baselines.

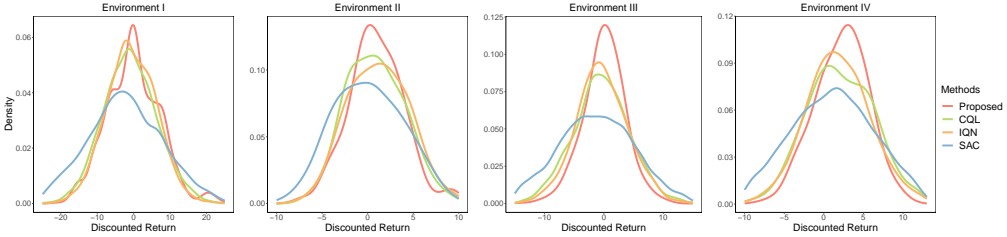

Figure 5: The distribution of Monte-Carlo discounted sum of rewards over 50 repeated experiments.

### D.3.3 MODEL PERFORMANCE ON LARGE DATASET

We evaluate the model performance in large sample size scenarios (10,000 transition pairs ($n = 100, T = 100$) for all four environments. The results are presented in Figure 6. Deep RL baseline methods have some improvement in the model performance and variance reduction with increased training samples. Meanwhile, the quasi-optimal learning still outperforms all competing methods as shown in Figure 6.

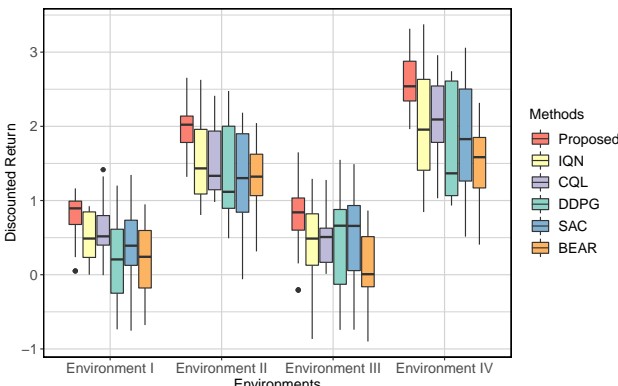

Figure 6: The boxplot of discounted return over 30 repeated experiments with sample size $N = 100, T = 100$.

### D.3.4 SAFE TRANSITIONS AND LEARNED POLICY DISTRIBUTION

**Safe Transition:** We illustrate the safety of the proposed method via evaluating the proportion of safe transition, i.e., from a fixed current state to a safe transition state. The goal of the OhioT1M case study is to maintain the glucose level in a safe range. The safe state in this study is defined as the state where the glucose level is within the range of 80-140 mg/dL. The reward function, i.e., the index of glycemic control (Rodbard, 2009),

$$R_i^t = -\frac{\mathbb{1}(S_{i,1}^t > 140)(140 - S_{i,1}^t)^{1.1} + \mathbb{1}(S_{i,1}^t < 80)(S_{i,1}^t - 80)^2}{30}$$

where $S_{i,1}^t$ is the glucose level of patient $i$ at $t$ decision stage. This reward setting tends to favor the safe range and penalize the risky scenario where the glucose level is out of the range of 80-140 mg/dL.

The details of the evaluation procedure are summarized in the following. In offline OhioT1M dataset, we pick up the observed states which transited to risky states, i.e., the states out of the safe range of glucose level. On the picked-up states, we calculate the proportion of safe transition, in which the corresponding transition states are sampled from the transition kernel under the learned policy. The transition kernel is estimated by maximum likelihood estimation from the offline dataset.

We summarize the results of the safe proportions on 1000 transition samplings in Figure 7. As shown, the quasi-optimal learning achieves 82.2% safe proportions, which outperforms 67.3% in safe RL baseline IQN and 44.6% in non-safe RL baseline SAC. By the results, we may conclude that quasi-optimal learning enjoys a better safety guarantee when applied to the medical domain.

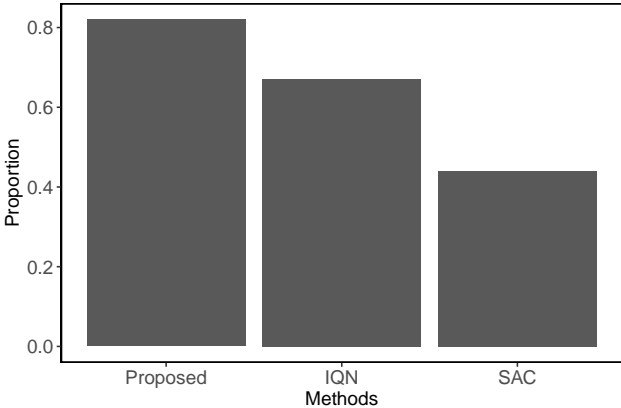

Figure 7: Proportions of Safe Transition from each Method

**The Learned Policy Distribution:** In this dimension, we illustrate the validity of the quasi-optimal policy distribution on a fixed state. In OhioT1M dataset, we select a patient state with a glucose level of 217 mg/dL, which is moderate hyperglycemia. On this state, we draw a density plot in Figure 8 for the policy distribution learned by the quasi-optimal learning, IQN, and SAC. Figure 8 shows that the quasi-optimal learning identified support regions $[3.15, 6.19]$. As the patient is under moderate hyperglycemia, so the moderate insulin dosage, i.e., $[3.15, 6.19]$, works well to decrease the glucose level into a safe range. Meanwhile, it avoids overly dropping the patient's glucose level and causes hypoglycemia. In comparison, SAC is risky as it has a non-negligible probability of assigning too low and too high insulin dosage to the patient. The policy learned by the safe RL algorithm IQN tends to avoid assigning extreme dosage, but it has wider support than the one learned by quasi-optimal learning. Regarding efficiency or safety, the quasi-optimal has certain advantages compared with IQN in this case.

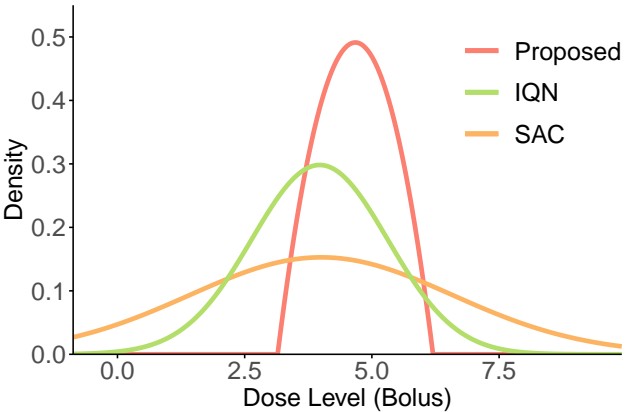

Figure 8: The Learned Policy Distribution of each Method for the Same Given State

