# OpenReview forum: "Quasi-optimal Reinforcement Learning with Continuous Actions"
_ICLR.cc/2023/Conference — ICLR 2023 poster_

### Official Review · Reviewer_xW9e · 2022-10-21

**Confidence:** 5
**Correctness:** 3
**Technical Novelty And Significance:** 3
**Empirical Novelty And Significance:** 3
**Recommendation:** 6

**Clarity, Quality, Novelty And Reproducibility:**

##Clarity
For the most part the paper is well constructed and clearly written. Some of the language connecting the ideas to healthcare could be tightened up to not be so speculative (e.g. "Clinicians are *pretty* interested...", etc). One way to do this is by including actual citations of clinical work to justify the statements and concepts. If the authors are speculating, it's probably isn't appropriate for them to be making as broad of claims as they are.

## Quality
Aside from my concerns about the experiments and the lack of clarity around them (shared in the "Weaknesses" section above), this paper is of sufficiently high quality. The conceptual and theoretical justification for the proposed method are very well set out.

## Novelty
As a core RL method, constraining learning in continuous action-spaces, this paper presents novel methodology. As a RL method proposed for healthcare, there is some cause for concern as some of the concepts and motivations are clearly covered in prior literature.

**Details Of Ethics Concerns:**

No outstanding ethics concern but I disagree with the authors' estimation that there is no risk in their ethics statement. There is abundant risk of someone implementing and using this algorithm without it being wholly validated in a controlled setting.

**Strength And Weaknesses:**

I'll first list the **strengths** of this paper:
 - The problem setting (constraining continuous action spaces to remain safe) is very important for the potential use of contemporary RL algorithms in real-world scenarios.
- The authors have appropriately motivated the design of the approach from a theoretical foundation.
- I found the theoretical justification and design of the quasi-optimal Bellman operator (aided by using q-Gaussian distributions) to be really interesting.
- For the most part the derivation of the methodology is clear and easy to follow (specific examples of where this is not clear will be listed below in the weakness section). The authors have taken great care to list out the failings or limitations of each component and then how the next concepts address this limitation. This made following the line of reasoning and development much easier.

Now for the **weaknesses**:
- The use of $\mu$ is not entirely clear. It's not clearly defined in the paper and shows up as an identifier for the quasi-optimal Bellman operator (first appears in Section 3). It would be helpful if the usage of this parameter (is it a scalar? a stand-in for a sampling policy? etc..) was more precisely discussed when it is first introduced. For example, it's never discussed how $\mu$ is chosen and what is used for that selection process. Since this construction is critical for defining the disjoint regions of admissible actions, this should be better handled throughout the paper.
- Figure 1 is not very informative. There is not enough detail in the main body of the paper to round out the insights that the Figure is trying to reaffirm. For instance, it's unclear what actions are admissible since the volume under the curves primarily correspond to the $<2\mu$ region. I was led to believe (based on the definition of the screening set) that the admissible actions--eg. the support set-- would satisfy the $\leq2\mu$ condition. The term "screening intensity" is undefined".
- The choice of policy class to be concavely quadratic seems to be a pretty strong assumption/constraint? It's unclear whether the component $\alpha_*(s)$ functions are explicitly learned or not. Being more straightforward about this would be helpful, particularly if there's a difference in how the policies (and underlying value functions) are pragmatically used. There's clear benefit theoretically but...
- The assumption that equation 10 can be "solved" with off-policy data also assumes that the data is self-contained and consistent, right? I mean, that the data provides full coverage and there is no stochasticity/confounding? Otherwise, a batch of data would need to be infinite in order to have the value function and corresponding policy fully converge, correct?
- The use of the Kernel representation is not very well motivated or explained. This was perhaps the weakest point of the theoretical derivation of the full method in my eyes. I can understand it's utility and why the choice was made. But the follow through and justification (and then actual implementation) is missing from the paper.
- The experiments are not adequately described nor introduced. I cannot ascribe any significance to the presented results without understanding how the specific experiments were designed to test the proposed method and how the chosen baselines are fairly compared to. I'm not dismissing the results in total but I am not overly enthused by them since it is clear that the DeepRL baselines aren't set up to be real great comparisons (due to dataset limitations as well as the overall design of the algorithms). None of these baseline approaches claim to work in the small data regime so it's no surprise that the observed variance of the results is as high as it is.
- Continuing with the complaints I have about the experiments, it's unclear what the experimental procedure is. Were the algorithms trained in a fully offline, off-policy fashion? If not, why?
- The biggest weakness of this paper in my mind and which causes hesitation for me to consider accepting this work is that it does not adequately frame its contributions among the decades of work in safe RL as well as work applying Reinforcement Learning to healthcare challenges. These omissions overlook relevant contributions made in the community recently that address similar technical considerations raised in this paper. I will describe two such papers that the authors should consider as well as point to a really through survey of the field (as of 2020). Beyond this there are several more papers that should be considered by the authors for framing and adequately justifying the development of algorithms and strategies motivated by and for use in healthcare.
  - First, I'll point to Tang, et al (ICML 2019) "Clinician-in-the-loop decision making". Which constructs set-valued policies of nearly optimal actions, learning to ignore those regions of the action space that are not relevant. While this paper doesn't address continuous action-spaces the theoretical foundations and questions about "quasi-optimal" learning are very similar.
   - Second, I'll point to Fatemi, et al (NeurIPS 2021) "Medical Dead-ends" which introduces a way to re-think learning value functions in data-limited safety-critical environments by assessing regions of risk and recommending that these actions be removed from the list of options an expert can consider.
   - For the survey, I strongly recommend Yu, et al (2021) "Reinforcement Learning in Healthcare: a survey".

**Summary Of The Paper:**

This paper addresses learning in continuous action-spaces where circumspection of "optimal" behavior is necessary, specifically focused on healthcare. The paper proposes a method by which to constrain the policy search space to only those areas that are "quasi-optimal". Detailed theoretical justification and analysis of the proposed quasi-optimal Bellman operator is provided as well as preliminary application of the approach to a diabetes management dataset.

**Summary Of The Review:**

I found this paper to be very well motivated and throughly justified from a theoretical perspective. It's clear that the quasi-optimal learning strategy has some bearing for appropriately constraining policies in continuous action settings. I am less convinced about it's overall use in important safety-critical environments such as healthcare. Conceptually I am aligned with the proposed direction but I cannot currently advocate for the paper's publication at this stage. I would need to see more thorough experimental analysis of how the proposed method actually avoids high-risk or dangerous action decisions in comparison to prior methods to demonstrate that the proposed method is working as intended. The cumulative "score" in Table 1 does not communicate that to me. It's nice to see that the proposed method learns "better" policies but I'm not sure that it actually is fulfilling the design of the approach.

---

> ### Author Response · Authors · 2022-11-18
> **Response to Reviewer xW9e (7/7)**
>
> **Weakness 9: Demonstrate proposed method truly avoids high risk actions.**
>
> We demonstrate the potential of the proposed quasi-optimal learning  for preserving safety and avoiding high-risk actions in the following **three dimensions** with experiments supports
>
> **Dimension 1: Distribution Evaluation Criterion:** To measure the performance on safety, we aim to evaluate the distribution of Monte-Carlo discounted sum of rewards for each roll-out trajectory [12], instead of just its empirical mean, i.e., discounted return.
>
> In particular, we generate $100$ trajectories under the learned policies and record the discounted sum of rewards of each trajectory. Then we draw the density plots in **Figure 5** on page 44 for all four environments. As shown in Figure 5, the distribution of the quasi-optimal learning is left-skewed, which is aligned to two safe RL algos IQN and CQL. The left-skewed phenomenon indicates that there is less chance to follow a low reward trajectory which is damaged by allocating highly-risk actions. However, the non-safe RL algo SAC is no-skewed distributed;  and it rolls out low reward trajectories in higher probability (fat left tail) compared to the quasi-optimal learning and two safe RL baselines. This validates that quasi-optimal learning avoids risky actions as the other two safe RL algos.
>
> **Dimension 2: Safe Transition:** In this dimension,
> we illustrate the safety by evaluating the proportion of safe transition, i.e., from a fixed current state to a safe transition state. In OhioT1M case study, the goal is to maintain the glucose level in a safe range. The safe state in this study is defined as the state where the glucose level is within the range of 80-140 mg/dL. The reward function, i.e., the index of glycemic control [14],
> \$
> R_i^t=-\frac{\mathbb{1}(S_{i,1}^t>140)(S_{i,1}^t-140)^{1.1}+\mathbb{1}(S_{i,1}^t<80)(S_{i,1}^t-80)^2}{30}
> \$
> where $S_{i,1}^t$ is the glucose level of patient $i$ at $t$ decision stage. This setting of reward favor the safe range and penalize the risky scenario where the glucose level is out of the range of 80-140 mg/dL.
>
> The details of the evaluation procedure are summarized as follows. In OhioT1M data, we pick up the observed states which transited to risky states, i.e., the states out of the safe range of glucose level. On the picked-up states, we calculate the proportion of safe transition, in which the corresponding transition states are sampled from the transition kernel under the learned policy. The transition kernel is estimated by maximum likelihood estimation from the offline dataset.
>
> We summarize the results of the safe proportions on 1000 transition samplings in **Figure 7** on page 46. As shown, the quasi-optimal learning achieves **82.2%** safe proportions, which outperforms **67.3%** in safe RL baseline IQN and **44.6%** in non-safe RL baseline SAC. Thus, we may conclude that quasi-optimal learning enjoys a better safety guarantee when applied to the medical domain.
>
> **Dimension 3: The Learned Policy Distribution:** We illustrate the validity of the quasi-optimal policy distribution on a fixed state. In OhioT1M data, we select a patient state with a glucose level of $217$ mg/dL, which is moderate hyperglycemia. In this state, we draw a density plot in **Figure 8** on page 46 for the policy distribution learned by the quasi-optimal learning, IQN and SAC. **Figure 8** shows that the quasi-optimal learning identified a support region **$[3.15,6.19]$**. As the patient is under moderate hyperglycemia, so the moderate insulin dosage, i.e., $[3.15,6.19]$, works well to decrease the glucose level into a safe range. Meanwhile, it avoids overly dropping the patient's glucose level, which causes hypoglycemia. In comparison, SAC is risky as it has a non-negligible probability to assign too low and too high insulin dosage to the patient. The policy learned by the safe RL algorithm IQN has a tendency to avoid assigning extreme dosage, but it has wider support than the one learned by quasi-optimal learning. Regarding efficiency or safety, the quasi-optimal has certain advantages towards to IQN in this case.
>
> **10. Other issues**
>
> Following your suggestion, we have cited additional clinical papers to justify our statements. We also agree that our proposed method needs to be tested in a controlled experiment for practical use in medical settings, and have modified our ethics statement accordingly.
>
> ## References
> [1] Dabney, W., Ostrovski, G., Silver, D. & Munos, R. (2018), Implicit quantile networks for dis-
> tributional reinforcement learning, in ‘International conference on machine learning’, PMLR,
> pp. 1096–1105
>
> [2] Rodbard, D. (2009), ‘Interpretation of continuous glucose monitoring data: glycemic variability
> and quality of glycemic control’, Diabetes technology & therapeutics 11(S1), S–55.

---

> ### Author Response · Authors · 2022-11-18
> **Response to Reviewer xW9e (6/7)**
>
> **Weakness 8. Discussion and comparison of the quasi-optimal learning to safe RL and RL in healthcare**
>
> Thanks for your insightful comment and pointing out the two exciting RL sub-fields, and interesting papers in healthcare RL. In the following, we provide a pronounced discussions on RL in healthcare and safe RL with the connection to our method. Also, additional  experiments for comprehensive comparisons with safe RL algorithms are provided. Besides, we evaluate and illustrate of the quasi-optimal learning in the sense of the safety with experiments supports (please see our response for weakness 9 next).
>
> **RL in healthcare:**
>
> Reinforcement learning has a wide variety of applications in healthcare  [1]. Some of the recent works aim to solve safety issues when applying RL to healthcare domains.
> [2] considers identifying set-valued policies with near-optimal actions, which incorporates the expert knowledge from clinicians to assist in decision-making. As the same rationale in our proposed  quasi-optimal region, [2] also utilizes the value function to threshold a near-optimal action set. In addition, their work and our work share the advantage that the admissible action set accounts for differences in values at different states.
>
> While we can find some common between the two works, our  framework is
> derived from a different way. Here, we discuss several major differences between the two frameworks: i) Our quasi-optimal learning not only identify  a near-optimal set, but induces a generic policy distribution which is directly applicable for action recommendations. In comparison, [2] only constructs a set-valued policy but would require expert knowledge to determine the action selected.
> ii) [2] considers a discrete action space, while our proposed deals with a more challenging continuous control problem. iii) The algorithm proposed by [2] needs to interact with the environment during the training process, while our framework is implemented in a fully offline scenario.
>
> Another interesting work is proposed by [3]. They consider identifying high-risk states in data-constrained offline settings by training two separate Q functions that model the probability of negative outcomes and positive outcomes respectively. They target to identify treatments proportional to their chance of leading to dead-ends, and attain safety by excluding these treatments from consideration.
>
> This method provides an interesting insight into ensuring safety and promising usage in medical settings. However, as they aim to identifies possible “dead-ends” of a state space and treatments, there exists a trade-off between the safety and optimality. In particular, it still has a gap for optimal treatment allocations. Under their framework, the near-optimal regret guarantee is vacuous. In contrast, the quasi-optimal has a potential to balance this tradeoff, i.e.,  ensuring safety with near-optimal guarantee shown by our experiments results and theoretical supports.
>
> Other interesting works in RL for healthcare including:  [4-5] adopt RL algorithms for sepsis treatment recommendations, [6] redefine the state variables and reward function to reflect practical safety concerns in sepsis treatments. We provide a pronounced discussion of RL in healthcare in Section 5 and Appendix.
>
> ## References
>
> [1] Yu, C., Liu, J., Nemati, S. & Yin, G. (2021), ‘Reinforcement learning in healthcare: A survey’,
> ACM Computing Surveys (CSUR) 55(1), 1–36.
>
> [2] Tang, S., Modi, A., Sjoding, M. & Wiens, J. (2020), Clinician-in-the-loop decision making:
> Reinforcement learning with near-optimal set-valued policies, in ‘International Conference on
> Machine Learning’, PMLR, pp. 9387–9396.
>
> [3] Fatemi, M., Killian, T. W., Subramanian, J. & Ghassemi, M. (2021), ‘Medical dead-ends and
> learning to identify high-risk states and treatments’, Advances in Neural Information Process-
> ing Systems 34, 4856–4870.
>
> [4] Henry, K. E., Hager, D. N., Pronovost, P. J. & Saria, S. (2015), ‘A targeted real-time early
> warning score (trewscore) for septic shock’, Science translational medicine 7(299), 299ra122–
> 299ra122.
>
> [5] Komorowski, M., Celi, L. A., Badawi, O., Gordon, A. C. & Faisal, A. A. (2018), ‘The artificial
> intelligence clinician learns optimal treatment strategies for sepsis in intensive care’, Nature
> medicine 24(11), 1716–1720.
>
> [6] Jia, Y., Burden, J., Lawton, T. & Habli, I. (2020), Safe reinforcement learning for sepsis treat-
> ment, in ‘2020 IEEE International Conference on Healthcare Informatics (ICHI)’, IEEE, pp. 1–7.

---

> > ### Author Response · Authors · 2022-11-18
> > **Response continued**
> >
> > **Connection between the safe RL and quasi-optimal learning**
> >
> > Safe reinforcement learning aims at finding an optimal policy while ensuring safety [1]. In the safe-RL framework, the definition of safety and its guarantee varies based on the specific purpose of learning tasks. There are three mainstream works for safe RL.
> >
> > * **Safe Exploration**: ensuring safe action allocations in the exploration process by incorporating prior knowledge, which often exists in online RL settings [2].
> > * **Safety Constraints**: finding an optimal policy that satisfies external user-specified safe constraints [3-4].
> > * **Risk-sensitivity and Conservatism**: finding a policy maximizing the infinite-horizon cumulative discounted reward while incorporating the notion of risk [5-6], e.g., value at risk (quantile), percentile performance, the variance of return.
> >
> > In medical applications, specifying explicit constraints is typically hard to realize in practice [7]. Alternatively, the notion of safety is usually incorporated in the design of reward functions, where high-risk actions lead to significantly low reward [8-9]. Based on these, our quasi-optimal learning is closely related to the risk-sensitive RL framework, which aims to control value at risk to ensure safety. For example, maintaining the discounted return above a certain threshold [10], reducing the variability of performance by avoiding extremely low performance [11], or target to maximize the robust performance criterion, e.g., quantile of the discounted return [12]. Popular risk-sensitive RL algos include CQL [13] and IQN [12]. CQL learns a conservative Q-function such that the expected value of a policy under this Q-function lower-bounds its true value, and thus avoids selecting high-risk actions with over-estimation action-value. IQN models the full quantile function for the state-action return distribution and yields risk-sensitive policies. **For a more comprehensive empirical study, we compare our algorithm to the aforementioned two safe RL baselines, conduct additional numerical experiments and analyze the results from the safety point of view.** We report the discounted return of the two safe RL algos in the updated **Figure 3** and **Table 1** on page 8.  Overall, quasi-optimal learning outperforms the two safe RL algos. The performance gains might come from quasi-optimal regions identification and bypassing off-support bias issues in continuous action domain. In addition to model performance, we demonstrate the safety of quasi-optimal learning in three dimensions with experiments supports in our next response.
> >
> >
> > ## References
> > [1] Garcıa, J. & Fern ́andez, F. (2015), ‘A comprehensive survey on safe reinforcement learning’, Journal of Machine Learning Research 16(1), 1437–1480.
> >
> > [2] Pham, T.-H., De Magistris, G. & Tachibana, R. (2018), Optlayer-practical constrained optimization for deep reinforcement learning in the real world, in ‘2018 IEEE International Conference
> > on Robotics and Automation (ICRA)’, IEEE, pp. 6236–6243.
> >
> > [3] Chow, Y., Nachum, O., Duenez-Guzman, E. & Ghavamzadeh, M. (2018), ‘A lyapunov-based approach to safe reinforcement learning’, Advances in neural information processing systems
> > 31.
> >
> > [4] Gu, S., Yang, L., Du, Y., Chen, G., Walter, F., Wang, J., Yang, Y. & Knoll, A. (2022), ‘A review of safe reinforcement learning: Methods, theory and applications’, arXiv preprint
> > arXiv:2205.10330.
> >
> > [5] Morimura, T., Sugiyama, M., Kashima, H., Hachiya, H. & Tanaka, T. (2010), Nonparametric return distribution approximation for reinforcement learning, in ‘ICML’.
> >
> > [6] Mavrin, B., Yao, H., Kong, L., Wu, K. & Yu, Y. (2019), Distributional reinforcement learning for efficient exploration, in ‘International conference on machine learning’, PMLR, pp. 4424–4434.
> >
> > [7] Vincent, R. (2014), Reinforcement learning in models of adaptive medical treatment strategies, McGill University (Canada).
> >
> > [8] Raghu, A., Komorowski, M., Ahmed, I., Celi, L., Szolovits, P. & Ghassemi, M. (2017), ‘Deep reinforcement learning for sepsis treatment’, arXiv preprint arXiv:1711.09602
> >
> > [9] Jia, Y., Burden, J., Lawton, T. & Habli, I. (2020), Safe reinforcement learning for sepsis treat-
> > ment, in ‘2020 IEEE International Conference on Healthcare Informatics (ICHI)’, IEEE, pp. 1–7.
> >
> > [10] Tamar, A., Glassner, Y. & Mannor, S. (2015), Optimizing the cvar via sampling, in ‘Twenty-
> > Ninth AAAI Conference on Artificial Intelligence’.
> >
> > [11] Ma, X., Xia, L., Zhou, Z., Yang, J. & Zhao, Q. (2020), ‘Dsac: distributional soft actor critic for
> > risk-sensitive reinforcement learning’, arXiv preprint arXiv:2004.14547.
> >
> > [12] Dabney, W., Ostrovski, G., Silver, D. & Munos, R. (2018), Implicit quantile networks for dis-
> > tributional reinforcement learning, in ‘International conference on machine learning’, PMLR,
> > pp. 1096–1105.
> >
> > [13] Kumar, A., Zhou, A., Tucker, G. & Levine, S. (2020), ‘Conservative q-learning for offline rein-
> > forcement learning’, Advances in Neural Information Processing Systems 33, 1179–1191.

---

> ### Author Response · Authors · 2022-11-18
> **Response to Reviewer xW9e (5/7)**
>
> **The motivation for choosing the baseline algorithms**
>
> **SAC**: SAC is the most well-known entropy-regularized algorithm (our proposed method is related to this type of algorithm algorithmically). Also, SAC is one of the  Gaussian policy family methods for continuous action control.
>
> **DDPG**: DDPG is a well-known benchmark for continuous control. It can be categorized into a policy gradient family. In comparison, our proposed method can be categorized into an actor-critic type algorithm. The actor-critic and policy gradient methods are two mainstream types of RL, so we compare our method to a policy gradient type algorithm benefits for comprehensive comparisons.
>
> **BEAR**: BEAR is a SAC-based offline RL algorithm. It is specifically designed for offline settings, and it constrains the support of policy distribution within the data distribution. Therefore, it could avoid off-support bias to some extent, which is also an advantage of our proposed methods. We include this method for a more comprehensive comparison.
>
> **V-learning** The V-learning is a discretized family approach, which is different from the stream directly modeling policy by a continuous probability distribution. The comparison with this discretization method is helpful in the comprehensive comparison sense. Also, V-learning leverage for solving the estimating equation to learning a stochastic optimal policy and has been shown to be stable and convergence in off-policy training under function approximation and has good statistically sound. And V-learning works well in small sample sizes.
>
> **Greedy-GQ** Greedy-GQ learning is also a discretized family approaches. While V-learning would induce a stochastic policy, Greedy-GQ would induce a deterministic policy. It is designed for medical applications and works well in a small sample size.
>
>
> **CQL:** CQL is a well-known benchmark for offline reinforcement learning on continuous action space. Thanks to its conservative nature, it reaches state-of-the-art performance and enjoys safe policy improvement guarantees. It is widely considered a baseline when compared with other risk-sensitive methods. We include this method to compare the overall performance with our proposed methods and validate that the proposed method could indeed gain improvements in avoiding risk.
>
> **IQN:**  IQN models the quantiles of Q-function and can be used for risk-sensitive tasks by maximizing a distorted expectation of Q-function. It demonstrates great performance for risk-sensitive tasks. We include this method as a comparison with other risk-sensitive RL methods. We demonstrate that the proposed method could achieve better performance in avoiding risk by screening out actions with low state-action values.
>
>
> **Weakness 7: Experimental Procedure**
>
> Yes, all the competing methods are off-policy algorithms and trained in a fully offline fashion for fair comparisons. The three newly added baselines, V-learning [1], IQN [2], and CQL [3] are also trained in fully offline and off-policy fashion.
>
>
> For each synthetic experiment run, in the estimation stage, we learned  optimal policies for all methods on the same offline dataset without interaction with the environment; in the evaluation stage, we executed the learned policy on the synthetic dynamics for collecting the Monte-Carlo discounted return as a criterion model performance.
>
> The estimation procedure for the real data analysis is the same as in the synthetic experiments. In the evaluation stage, we follow [1] to compute the value estimates as the model performance criterion.
>
> More details of the experiments are disclosed in the Appendix. We refer the reviewer to page 41 for details.
>
> ## References
> [1] Luckett, D. J., Laber, E. B., Kahkoska, A. R., Maahs, D. M., Mayer-Davis, E. & Kosorok, M. R.
> (2019), ‘Estimating dynamic treatment regimes in mobile health using v-learning’, Journal of
> the American Statistical Association
>
> [2] Dabney, W., Ostrovski, G., Silver, D. & Munos, R. (2018), Implicit quantile networks for dis-
> tributional reinforcement learning, in ‘International conference on machine learning’, PMLR,
> pp. 1096–1105
>
> [3] Kumar, A., Zhou, A., Tucker, G. & Levine, S. (2020), ‘Conservative q-learning for offline rein-
> forcement learning’, Advances in Neural Information Processing Systems 33, 1179–1191.

---

> ### Author Response · Authors · 2022-11-18
> **Response to reviewer xW9e (4/7)**
>
> **Weakness 6: Experiment Details**
>
> Thanks for pointing this out. Due to page limits, we moved the details of numeric experiment settings and the motivation in Appendix on page 41. For your convenience, we discuss the motivations for the four synthetic experiment designs and the choices of the baseline approaches in the following.
>
> **Synthetic experiment design:** We aim to test the performance of our proposed method on the settings of bounded and unbounded continuous action space with unimodal and multimodal reward functions. The motivation for testing the proposed method in bounded action space is to test if the proposed method could potentially handle the off-support bias, as illustrated in **Figure 2**. The reason for considering a multimodal synthetic environment is to evaluate the quasi-optimal policy class (q-Gaussian policy class) works in a relatively complex situation. Especially for the q-Gaussian policy distribution which is unimodal, it is necessary to test if the q-Gaussian policy still works and is robust to the scenario where the optimal policy might be multimodally behaving.
> We make a summary of the synthetic experiments as follows:
>
> Environment  I:
> * Setting: Bounded action space and unimodal reward function
> * Purpose: To evaluate if the quasi-optimal learning works in the scenario where it might suffer the off-support bias issue as the continuous action space is bounded.
>
> Environment  II:
> * Setting: Bounded action space and multimodal reward function
> * Purpose: In addition to the purpose in Environment I, we aim to implement quasi-optimal learning in a more challenging environment. Also, this is for evaluating the robustness of the unimodal q-Gaussian policy under the scenario that the true optimal policy follows a multimodal probability distribution.
>
> Environment  III:
>
> * Setting: High-dimension state space and well-separated reward function. The design of the well-separated reward function causes the effect that the selection of non-optimal or sub-optimal actions greatly damages the rewards and increases the risk.
> * Purpose: To evaluate the reliability/safety of quasi-optimal learning. We aim to examine if quasi-optimal learning could perform well in this scenario. As we expect quasi-optimal learning is able to identify the quasi-optimal sub-regions and avoids choosing those non-optimal/sub-optimal actions which greatly damage the performance.
>
> Environment  IV:
>
> * Setting: High-dimension state space and complex well-separated reward function.
> * Purpose: In addition to the purpose in Environment III, we target to evaluate the quasi-optimal learning in a more complex environment, imposing great challenges on recovering the quasi-optimal regions for the proposed method. Indeed, imposing more complex structures on reward function indicates imposing difficulties on value function learning and thus imposes great challenges on identifying quasi-optimal regions.

---

> ### Author Response · Authors · 2022-11-18
> **Response to Reviewer xW9e (3/7)**
>
> **Weakness 4: Full coverage and no confounding assumption.**
>
> Yes, this is a very insightful comment! We do require full coverage and no unmeasurable confounder assumptions at this stage. As potential future research directions, it would be very interesting to relax the two assumptions for better applying the proposed method to more complex environments. The potential tools include leveraging the pessimism principle [1] for resolving the insufficient data coverage and dealing with unmeasurable confounders via instrumental variable approaches [2-3].
>
>
> **Weakness 5: Motivation of Kernel Representation.**
>
> The motivation for choosing kernel representation is based on our finding on the continuity invariance between the reward and optimal weight function $u^{*}(\cdot)$. We formally characterize this finding in the following theorem. The proof of this theorem is included in Appendix (Theorem S.2). We present the Theorem as follows,
>
> **Theorem S.2**. We define the optimal weight function as $u^*=argmax_{u\in L^2(C_0)}\mathcal{L}^2(V_{\mu},\pi_{\mu} ,{\eta},\varpi,u)$. Let $\mathbb{C}(\mathcal{S} \times \mathcal{A})$ be all continuous functions on $\mathcal{S} \times \mathcal{A}$. For any $(s,a) \in \mathcal{S} \times \mathcal{A}$ and $s^{\prime} \in \mathcal{S}$, the optimal weight function $u^*(S^t,A^t) \in   L^2(C_0)  \cap \mathbb{C}(\mathcal{S} \times \mathcal{A})$ and is unique if the reward function $R(s^{\prime},s,a)$ and the transition kernel $\mathbf{P}(s^{\prime}|s,a)$ are continuous over $(s,a)$.
>
>
> Theorem S.2 implies that the optimal weight function $u^{*}$ is continuous as long as the reward function and transition kernel are continuous, which is widely satisfied in real-world applications. This provides us with a basis for leveraging the kernel representation. As known, when the bounded RKHS $H_{\text{RKHS}}({C_0})$ is reproduced by a universal kernel, it can approximate a continuous function space $\mathbb{C}(\mathcal{S}\times \mathcal{A})$ with arbitrarily small error. Specifically, the approximation error of $H_{\text{RKHS}}({C_0})$,
> $$
> \varepsilon(C\_0):=\\sup\_{u \\in \\mathbb{C}(\\mathcal{S}\\times \\mathcal{A})} \\inf _{f \\in H\_{\\text{RKHS}}({C_0})}\\|f-u\\|\_{\infty},
> $$
> decreases as $C_0$ increases, and vanishes to zero as $C_0$ goes to infinity [4].
>
> For this reason, we propose to model the weight function in a bounded RKHS
>  space.
>
> Under the kernel representation framework, an additional advantage gained is to decouple the two-stage min-max optimization to a one-stage minimization problem. The basic idea is to follow the principle of the maximum mean discrepancy [5], and thus the inner maximization under the kernel representation will have a closed-form solution. We formally characterize this property in Theorem S.3 in Appendix. In comparison to other popular offline RL frameworks such as Bellman residual minimization (BRM, [6-8]) involving solving a minimax problem, reducing the min-max optimization to minimization optimization greatly benefits the computation efficiency and stability in practice.
>
> ## References
>
> [1] Jin, Y., Yang, Z. & Wang, Z. (2021), Is pessimism provably efficient for offline rl?, in ‘Interna-
> tional Conference on Machine Learning’, PMLR, pp. 5084–5096.
>
> [2] Liao, L., Fu, Z., Yang, Z., Wang, Y., Kolar, M. & Wang, Z. (2021), ‘Instrumental variable value
> iteration for causal offline reinforcement learning’, arXiv preprint arXiv:2102.09907.
>
> [3] Fu, Z., Qi, Z., Wang, Z., Yang, Z., Xu, Y. & Kosorok, M. R. (2022), ‘Offline reinforcement
> learning with instrumental variables in confounded markov decision processes’, arXiv preprint
> arXiv:2209.08666.
>
> [4] Bach, F. (2017), ‘Breaking the curse of dimensionality with convex neural networks’, The Journal
> of Machine Learning Research 18(1), 629–681.
>
> [5] Gretton, A., Borgwardt, K. M., Rasch, M. J., Sch ̈olkopf, B. & Smola, A. (2012), ‘A kernel
> two-sample test’, The Journal of Machine Learning Research 13(1), 723–773.
>
> [6] Antos, A., Szepesv ́ari, C. & Munos, R. (2008), ‘Learning near-optimal policies with bellman-
> residual minimization based fitted policy iteration and a single sample path’, Machine Learn-
> ing 71(1), 89–129.
>
> [7] Hoffman, M. W., Lazaric, A., Ghavamzadeh, M. & Munos, R. (2011), Regularized least squares
> temporal difference learning with nested 2 and 1 penalization, in ‘European Workshop on
> Reinforcement Learning’, Springer, pp. 102–114.
>
> [8] Chen, J. & Jiang, N. (2019), Information-theoretic considerations in batch reinforcement learn-
> ing, in ‘International Conference on Machine Learning’, PMLR, pp. 1042–1051.

---

> ### Author Response · Authors · 2022-11-18
> **Response to Reviewer xW9e (2/7)**
>
> **Weakness 2: Interpretation for Figure 1.**
>
> Thank you for pointing this out, and your understanding of Figure 1 is exactly correct. Basically, Figure 1 aims to visually
> demonstrate what's the quasi-optimal sub-regions and how it potentially adapts the values of the threshold parameter $\mu$ in a 1-dim action space. We have revised Figure 1 to make this demonstration more intuitive.
>
> According to the design of the set in Equation (5), the admissible action set is the intervals in X-axis, which are vertically projected from the pink area.
> In this sense, the pink area is determined, then the admissible action set is determined accordingly. The pink area has the volume $2\mu$, indicating that for a specific action $a \in \mathcal{A}$, if it satisfies the condition
> $$
>  \\int_{a^{\\prime}\\in \\mathcal{M}\_{s}(a) }Q\_{\\mu}^{*}(s,a^{\\prime})da^{\\prime}-Q\_{\\mu}^{\*}(s,a)\\sigma(\\mathcal{M}_{s}(a))\\leq 2\\mu,
> $$
> then this action belongs to the admissible action set. Here, when $\mu$ increases, the condition is relatively loose and in general leads to wider quasi-optimal sub-regions. Reversely, the condition is restrictive, and more actions are screened out from the support set, i.e., quasi-optimal region.
>
> The term "screening intensity" describes the strongness of the actions that will be filtered out from the admissible set. For example, a relatively small $\mu$ indicates the screening intensity is stronger since more actions are screened out from the support set.
>
>
> We have provided more details for interpreting **Figure 1** in the updated caption.
>
>
> **Weakness 3: Choice of policy class.**
>
> Thanks for you comment. We would like to first clarify that we do not impose any constraints, e.g., concavity or
> quadratic shape, on the general quasi-optimal policy class defined in Equation (4). And all of our theoretical justifications, in Section 3.3 and 4, are developed for this general quasi-optimal policy class.
>
> For the q-Gaussian policy distribution defined in Theorem 3.2, it was introduced as a special case of the general quasi-optimal policy class. Aligned with the reviewer's statement, the q-Gaussian policy class indeed requires concavity and quadratic property. However, all coefficient parameters in this formulation are data-adaptive and must be solved through optimization. We want to mention that, in the continuous control problems, the concave and quadratic policy classes are widely used for policy distribution modeling, e.g., using Gaussian policy distribution [1-3], Beta policy distribution [4], etc.
>
> On the other hand, the q-Gaussian policy class sacrifices certain model flexibility but gains a great computational advantage. As shown in Equation (8), the q-Gaussian has an analytical solution for the quasi-optimal region. This helps to overcome the calculations on solving the support following the rule as in Equation (5).
>
> The coefficient components, $\alpha_1(\cdot), \alpha_2(\cdot)$ and $\alpha_3(\cdot)$ in q-Gaussian policy class are explicitly learnable functions. Pragmatically, we parameterize the components by cubic splines basis, such that $Q_{\mu}^{*}(s,a; \theta)=-\exp\{\theta_1^T \varphi(s)\} a^2+ \theta_2^T \varphi(s) a+\theta_3^T\varphi(s),$ where $\varphi(s)=[\varphi_1(s),\varphi_2(s),...,\varphi_m(s)]^T$ is the $m$-dimensional cubic-spline basis, and $\theta=[\theta_1,\theta_2,\theta_3]^T$ is the $3 m$-dimensional parameters we need to estimate. We parametrize $\alpha_1(s)=-\exp\{\theta_1^T \varphi(s)\} $ to force it to be negative numerically, thus the resulting optimal policy would follow the q-Gaussian distribution. During the later estimation procedure, the parameter $\theta$ of the cubic spline model is estimated via implementing Algorithm 1.
>
> ## References
>
> [1] Williams, R. J. (1992), ‘Simple statistical gradient-following algorithms for connectionist rein-
> forcement learning’, Machine learning 8(3), 229–256.
>
> [2] Silver, D., Lever, G., Heess, N., Degris, T., Wierstra, D. & Riedmiller, M. (2014), Deterministic
> policy gradient algorithms, in ‘International conference on machine learning’, PMLR, pp. 387–
> 395.
>
> [3] Fujimoto, S., Hoof, H. & Meger, D. (2018), Addressing function approximation error in actor-
> critic methods, in ‘International conference on machine learning’, PMLR, pp. 1587–1596.
>
> [4] Chou, P.-W., Maturana, D. & Scherer, S. (2017), Improving stochastic policy gradients in con-
> tinuous control with deep reinforcement learning using the beta distribution, in ‘International
> conference on machine learning’, PMLR, pp. 834–843.

---

> ### Author Response · Authors · 2022-11-18
> **Response to Reviewer xW9e (1/7)**
>
> We thank the reviewer for constructive suggestions to improve the manuscript. We hope your concerns can be addressed by the following responses.
>
> **Weakness 1: The use of $\mu$**
>
> In our proposed framework, the threshold parameter $\mu$ is a positive scalar, i.e., $\mu \in \mathbb{R}^{+}$.
>
> Upon Legendre-Fenchel transform on the vanilla Bellman operator $\mathcal{B}$, the scalar parameter $\mu$ is introduced for adapting the strength of the approximation. As shown in Theorem 3.1, for any $s\in \mathcal{S}$ and value function $V$,
> \$
> \mathcal{B}_{\mu}V(s)-\mathcal{B}V(s)\in [\mu(1-\mathbf{C}),\mu].
> \$
>
> To understand the use of $\mu$,  we can view it from two perspectives.
>
> **On the smoothing perspective**: The construction of Equation (2) is related to the Nesterov smoothing technique [1], and the scalar parameter $\mu$ controls the degree of the smoothness.
>
> **On the effect towards the quasi-optimal regions**: The parameter $\mu$ plays an important role in affecting the selection of the admissible actions. Based on Equation (5), a large $\mu$ allows more actions to be incorporated in the support set of the derived quasi-optimal policy. Reversely, the screening rule is more restrictive, and smaller action regions are included in the support as the admissible actions. For any fixed state $s \in \mathcal{S}$, the parameter $\mu$ is equivalent to the half of the integral difference between on action-value function $Q_{\mu}^{*}(s,\cdot)$ and on the lowest admissible action-value over the admissible action support set. In mathematical formulation, it is
> $$
> \\mu = \\frac{1}{2}\\left(\\int_{a\\in \\mathcal{W}\_s}Q\_{\\mu}^{\*}(s,a)da - \\sigma(\\mathcal{W}\_s) \\inf\_{a \\in \\mathcal{W}\_s}Q\_{\\mu}^{\*}(s,a)\\right),
> $$
> where $\mathcal{W}_s$ is the support set defined in page 5 underneath Equation (5).
>
> $$
> \\int\_{a\\in \\mathcal{W}\_s}Q\_{\\mu}^{\*}(s,a)da - \\sigma(\\mathcal{W}\_s) \\inf\_{a \\in \\mathcal{W}\_s}Q\_{\\mu}^{\*}(s,a) = 2\\mu
> $$
>
> Aligned with an illustrating example of the scenario for the one-dimension action space in Figure 1. The lowest admissible action-value corresponds to the red dashed line, and the integral difference is the pink shape area. In general, when $\mu$ increases, the pink area (integral difference) shrinks, and the quasi-optimal sub-regions become narrower. In contrast, a larger $\mu$ allows a large pink area, and the quasi-optimal sub-regions expand with more actions included.
>
> **For the choice of the parameter $\mu$:** There are two ways: First, the $\mu$ can be user-specified according to prior knowledge. Second, the $\mu$ can be selected via a data-driven principle. The $\mu$ can be  determined by a $k$-fold cross-validation procedure. Specifically, we choose a proper $\mu$, which aims to  maximize the average of the lower bound of the estimated discounted sum of rewards, i.e.,
> $$
> \\hat \\mu = argmax\_{\\mu}\\frac{1}{k}\\sum^k\_{j=1}\\mathbb{P}\_{n(j)}\\hat V^{(j)}\_{\\mu}(S^1)-\\frac{\\mu}{1-\\gamma},
> $$
> where $\\hat V^{(j)}\_{\\mu}$ is the value function estimator on the $j$th training set, $\\mathbb{P}\_{n(j)}$ is the empirical measure on the initial state $S^1$ for the $r$th validation set. The second term $\frac{\mu}{1-\gamma}$ is used to mitigate the effect of the additional proximity term $\mathbb{E}_{a \sim \pi(\cdot|s)}[\text{prox}(\pi(a|s))]$.
>
>
> In our empirical studies, we follow the data-driven way to select a proper $\mu$. To validate the cross-validation procedure works in practice, we  conduct sensitivities analyses with respect to the change of the $\mu$ in **Figure 4**. The results confirm that the cross-validation procedure indeed selects a proper $\mu$ which maximizes the discounted return.
>
>
> ## References
> [1] Nesterov, Y. (2005), ‘Smooth minimization of non-smooth functions’, Mathematical program-
> ming 103(1), 127–152

---

> ### Comment · Reviewer_xW9e · 2022-11-18
> **Response to Authors**
>
> Thank you for your very detailed responses. Thank you for your work to improve the paper. I am mostly satisfied by the revisions and appreciate the clarifications provided here and in the paper. I am inclined to raise my initial score but will wait to do so until after further discussions with the other reviewers (who's reviews and the corresponding author responses I have not yet read).
>
> I do however want to make a point about the statements made by the authors talking about safety-optimality trade-offs. In many (if not most) safety-critical decision making scenarios we will rarely have enough data available to develop truly optimal policies, and are constrained from exploratory data collection. This is especially true when the data is complex with unobserved confounding and partial observations. By considering sets of quasi optimal actions or by identifying decisions that should be avoided, we can empower the human expert to act more optimally. I fear that the fixation of "optimality" as a goal when it's intractable sits in conflict with the idea that the algorithms will not be used to replace human experts as stated in the ethics statement.

---

> > ### Author Response · Authors · 2022-11-19
> > **Reply to Reviewer xW9e**
> >
> > Thank you very much for your reply and encouraging message.
> >
> > We believe that our views on "the fixation of optimality as a goal" are in fact similar. Overall, we do not think optimality or any theoretical property is the only factor in improving health care. The goal of a theoretically well-understood model is to provide practitioners/experts with a more interpretable result, hence facilitating the collaboration, while the final decision should still be made by practitioners, at least in the current era. Here the interpretation is not in terms of how the prediction is calculated but rather what properties the prediction may achieve/enjoy. The interactions and collaborations between practitioners and machine learning scientists are the key factors for understanding the biology and creating better strategies to improve health. Our goal in this paper is to contribute to that pool of available tools to facilitate that communication.
> >
> > We hope that you will agree with our view here. And thank you for bringing up this insightful suggestion. We will update the manuscript in the future version to better reflect our aligned view on this issue.

---

> > ### Author Response · Authors · 2022-12-03
> > **Thank you very much!**
> >
> > Dear Reviewer xW9e,
> >
> > We thank you again for your time and efforts in reviewing our paper and providing such insightful and constructive comments, which are super valuable to the improvements of our manuscript. We also really appreciate your recognition and encouragement for the potential initial score raising. Please let us know if you have any additional comments or concerns, and we would be very happy to address them. As the discussion stage is coming an end, we kindly ask if it is possible to reconsider your score if the concerns were appropriately addressed so far.

---

> > ### Author Response · Authors · 2022-12-13
> > **Reply to Reviewer xW9e (updates)**
> >
> > We thank the reviewer again for your positive feedback and the increased score. We greatly appreciate your constructive comments to improve our work!

---

### Official Review · Reviewer_Y35L · 2022-10-24

**Confidence:** 4
**Correctness:** 3
**Technical Novelty And Significance:** 3
**Empirical Novelty And Significance:** 3
**Recommendation:** 8

**Clarity, Quality, Novelty And Reproducibility:**

The paper is clear and in good quality. The proposed algorithm as well as the theoretical results are novel. GitHub links are included to reproduce the numerical results.

**Strength And Weaknesses:**

The strengths and advances of the proposal includes:

1. The development of the introduction of the quasi-optimal Bellman operator for policy learning in continuous action space;
2. The use of q-Gaussian policy distribution to realize bounded support;
3. A practical algorithm for quasi-optimal policy learning;
4. Nice theoretical properties of the proposed algorithm;
5. The use of real datasets to justify the finite-sample performance of the algorithm.

Some suggestions, questions and comments:

1. The paper considers estimating the optimal policy from an offline dataset. It would benefit from a discussion about existing offline RL algorithms and whether/how these algorithms are related to your proposal.
2. Page 2 (minor) The notation paragraph. For the two sequences $(\Psi(m))_m$ and $(\Upsilon(m))_m$, I guess you require their elements to be strictly positive.
3. Page 3 (minor) $\Delta_{\textit{convex}}(\mathcal{A})$ is not defined.
4. I found Section 3.1 a bit difficult to follow. For instance, does Equation (2) holds for any $\rho$? When $\rho=0$, it seems the right-hand-side equals the optimal value function. So why does the penalty term $\rho$ exist on the right-hand-side? In addition, it seems the right-hand-side does not involve the state transition. So why is it referred as the Bellman operator? Equation (3), you may want to discuss the choice of $\mu$. Is the proximal term used to smooth the policy class? Is $\mu$ the degree of smoothness? A related question is, I was wondering if the Fenchel representation on the Bellman operator has been employed in some other papers.
5. It seems Theorem 3.2 implicitly requires the action space to be one-dimensional. Do you have similar results for multi-dimensional action space?
6. Section 3.3. It was mentioned that Equation (10) can be directly solved using the transition samples, so it overcomes the distributional shift issue in offline data. However, the distributional shift issue usually refers to the difference in the marginal state-action distribution instead of the state transitions.
7. In additional to the kernel representation, can we use other functional approximators to solve the minimax optimization? Would you please discuss?
8. Page 6, minimax optimization. It is not clear to me how the method based on average Bellman error solve the double sampling issue. Is it related to the couple estimation method (Antos, Szepesvari and Munos, 2008a; Farahmand et al., 2016)?
9. Page 8, numerical experiments. The action space in the four environments are one-dimensional. Can we compare the proposed with method against existing policy learning algorithms with certain discretization? This could better support the claim in the introduction section. When the action space is $[0,1]$, you may want to equally define the interval into e.g., 5 or 10 subintervals. When the action space is unbounded, we could first apply certain transformation (e.g., based on the normal cdf) and then divide the interval.


**Summary Of The Paper:**

The paper considers policy learning in continuous treatment setting, particularly in optimal dose finding. The contributions of the paper includes (1) the development of the quasi-optimal Bellman operator to address the non-smoothness issue; (2) the use of q-Gaussian policy distribution to avoid off-policy support; (3) the development of a PAC learnable algorithm with appealing theoretical properties; (4) comprehensive numerical analysis in comparison to several baseline algorithms.

**Summary Of The Review:**

The paper considers policy learning in continuous treatment setting, particularly in optimal dose finding. The proposed algorithm is novel and statistically sound. It is also empirically verified based on simulation and real data applications. However, I have some questions and comments regarding the presentation and the proposed methodology. I hope the author(s) can help address my comments.

---

> ### Author Response · Authors · 2022-11-18
> **Response to Reviewer Y35L (3/3)**
>
> **Comment 8: Double Sampling Issue.**
>
> The double-sampling issue is basically caused by using a one-sample counterpart of the unknown transition kernel to approximate its population version. That is, replacing $\\mathbb{E}\_{S^{t+1}|S^{t},A^{t}}[R(S^{t+1},S^{t},A^{t})+\\gamma V\_{\mu}(S^{t+1})]$ by  $R^{t}+\gamma V_{\mu}(S^{t+1})$ in our setting, and thus an unwanted conditional variance over $S^{t+1}$ will be introduced.
>
>
> The line of works [1-4] are usually categorized as the Bellman residual minimization (BRM) in offline reinforcement learning community. They solve the double-sampling issue based on nested optimization, where both the inner and outer optimization is over the same function space, and the unwanted conditional variance term can be canceled out. While this stream of methods also provides a treatment for solving the double-sampling issue, however, our weighted loss is
> derived from a different way.
>
> The linearity of the average Bellman error [5] motivates us to address the double sampling in our way. In the  average Bellman error, the conditional expectation $\\mathbb{E}\_{S^{t+1}|s,a}[\\cdot]$ is linear with respect to $\\mathbb{E}\_{S^{t}, A^{t}}[\\cdot]$ as no square function is imposed on $\mathbb{E}\_{S^{t+1}|s,a}[\\cdot]$ like $L^2$ Bellman error loss. In our proposed weighted loss, this linear structure also holds and helps to aggregate the marginal distribution of the state-action pairs $ d_{\pi_{b}}(S^{t},A^{t})$ with the transitional kernel $\textbf{P}(S^{t+1}|S^{t},A^{t})$ to be the joint distribution  $p_{\pi_b}(S^{t+1}, S^{t}, A^{t})$. Therefore, instead of approximating the unknown conditional expectation $\\mathbb{E}\_{S^{t+1}|S^{t},A^{t}}[\\cdot]$ by its one-sample counterpart, we can utilize a sufficient amount of transition pairs to approximate the expectation $\mathbb{E}_{S^{t},A^{t},S^{t+1}}[\cdot]$. This explains why our proposed weight loss fundamentally resolves the double sampling issue compared to the $L^2$ loss.
>
> **Comment 9: Comparison with Methods based on Discretization.**
>
> In the empirical studies, we would like to clarify that the competing method Greedy-GQ learning is a discretization based approach. We discretize the continuous action space with $20$ intervals and apply the existing Greedy-GQ learning method on the fine-grid discrete action space. For bounded action space, we discretize the action space uniformly; for unbounded action space, we use the logistic function to transform actions in $[0,1]$, and discretize the action space uniformly on the transformed action. More details of the setup of the experiment are provided in Appendix.
>
> Following the reviewer's suggestions, we implement one additional discretization-based approach: V-learning [6], for a more comprehensive comparison. We follow the same discretization strategy by dividing the action space into $20$ grids.
>
> The corresponding model performance is summarized in the updated manuscript. In **Figure 1** of the updated manuscript, we observe that the proposed method still outperforms V-learning.
>
> ## References
>
> [1] Antos, A., Szepesv ́ari, C. & Munos, R. (2008), ‘Learning near-optimal policies with bellman-
> residual minimization based fitted policy iteration and a single sample path’, Machine Learning 71(1), 89–129.
>
> [2] Hoffman, M. W., Lazaric, A., Ghavamzadeh, M. & Munos, R. (2011), Regularized least squares
> temporal difference learning with nested 2 and 1 penalization, in ‘European Workshop on
> Reinforcement Learning’, Springer, pp. 102–114.
>
> [3] Farahmand, A.-m., Ghavamzadeh, M., Szepesv ́ari, C. & Mannor, S. (2016), ‘Regularized policy
> iteration with nonparametric function spaces’, The Journal of Machine Learning Research
> 17(1), 4809–4874.
>
> [4] Chen, J. & Jiang, N. (2019), Information-theoretic considerations in batch reinforcement learn-
> ing, in ‘International Conference on Machine Learning’, PMLR, pp. 1042–105.
>
> [5] Jiang, N., Krishnamurthy, A., Agarwal, A., Langford, J. & Schapire, R. E. (2017), Contextual
> decision processes with low bellman rank are pac-learnable, in ‘International Conference on
> Machine Learning’, PMLR, pp. 1704–1713.
>
> [6] Luckett, D. J., Laber, E. B., Kahkoska, A. R., Maahs, D. M., Mayer-Davis, E. & Kosorok, M. R.
> (2019), ‘Estimating dynamic treatment regimes in mobile health using v-learning’, Journal of
> the American Statistical Association .

---

> ### Author Response · Authors · 2022-11-18
> **Response to Reviewer Y35L (2/3)**
>
> **Comment 5: q-Gaussian extension to  multidimensional action space.**
>
> We present the extension to multidimensional action space of Theorem 3.2 as follows.
>
> Suppose $\mathcal{A} =  \mathbb{R}^p$, to recover the multi-dimensional q-Gaussian distribution for the induced policy, we may model Q-function as a quadratic function $Q_{\mu}^*(s,a) = -a^TXa+a^Ty+z$, where $X$ is a $p\times p$ symmetrical positive definite matrix, $y$ is a $p$-dimensional vector, and $z$ is a constant. As $X$ is positive definite, we have $X=U^TDU$, where $U$ is orthogonal, and $D=diag(\\alpha\_1,\\alpha\_2,....,\\alpha\_p)$ is a diagonal matrix with $\\alpha\_i>0$. We further define $\\widetilde{a}=Ua=[\\widetilde{a}\_1,\\widetilde{a}\_2,...\\widetilde{a}\_p]\\in \\mathbb{R}\^p$, then the Q-function can be rewritten as $Q\_{\\mu}^*(s,\\widetilde{a})=-\\widetilde{a}^TD\\widetilde{a}+\\widetilde{a}^TUy+z=\sum^p_{i=1} -\\alpha\_i \\widetilde{a}\_i^2+\\sum\_{i=1}^p \\beta\_i \\widetilde{a}\_i+z$, where $Uy=[\beta_1,\beta_2,...,\beta_p]^T$. Therefore, we can now treat each $\\widetilde{a}\_i$ as independent variables. Recall from Equation (4), we now have the induced policy as
> $$
> \\pi\_{\\mu}\^\*(\\widetilde{a}|s)=\\bigg\\{\\frac{Q\_{\\mu}^*(s,\\widetilde{a})}{2\\mu}-\frac{\\int\_{\\widetilde{a}\\in \\mathcal{W}\_s} Q\_{\\mu}^*(s,\\widetilde{a}) d \\widetilde{a}}{2\\mu\\sigma(\\mathcal{W}_s)}+\\frac{1}{\\sigma(\\mathcal{W}\_s)}\\bigg\\}^+,
> $$
> where $\mathcal{W}_s$ shares the same definition as the one-dimensional space.
>
> Finally, using the same argument as in Theorem 3.2, we have the density of the induced multi-dimensional q-Gaussian policy as
> $$
> \\pi\_{\\mu}^*(\\widetilde{a}|s)=\\left\\{\\sum^p\_{i=1}\\Big[\\frac{\\alpha_i}{2\\mu}(\\widetilde{a}\_i+\\frac{\\beta\_i}{2\\alpha\_i})^2-\\frac{3}{2}(\\frac{\\alpha\_i}{12\\mu})^{\\frac{1}{3}}\\Big]\\right\\}^{+},
> $$
>  which is the exact form of multivariate q-Gaussian distribution. The original value of actions can be recovered by $a=U^T\widetilde{a}$.
>
> **Comment 6: Distribution shift**
>
> Following the reviewer's suggestion, we have revised our statement for a more precise presentation. In the updated manuscript, we state that
>
> "*The equation (9) connects the quasi-optimal value function $V_{\mu}^\*$ and policy function $\\pi\_{\\mu}^\*$ along with any arbitrary state-action pair. This provides an easy way to incorporate the off-policy data, i.e., the state-action pairs which are sampled from state-action visitation under behavior policy, without adjusting the distribution mismatch.*"
>
> **Comment 7: Other methods to solve minimax optimization.**
>
> Yes, in addition to kernel representation, we might consider using linear basis function representation. Specifically, let the weight function in linear function space, i.e., $u \in H_{\text{linear}}:= \{(s,a) \longmapsto \phi(s, a)^{\top} \beta: \beta \in \mathbb{R}^d \}$, where $\phi(s, a)$ are basis functions and $\beta$ are the parameters. Here the basis $\phi(s, a)$ could be a neural feature, kernel feature, and Fourier feature, etc.
> Then the inner maximization problem in Equation (10) still has a closed-form solution. A similar principle can be found in the existing works [1-2].
>
> ## References
>
> [1] Shi, C., Uehara, M., Huang, J. & Jiang, N. (2022), A minimax learning approach to off-policy
> evaluation in confounded partially observable markov decision processes, in ‘International
> Conference on Machine Learning’, PMLR, pp. 20057-20094.
>
> [2] Uehara, M., Huang, J. & Jiang, N. (2020), Minimax weight and q-function learning for off-policy
> evaluation, in ‘International Conference on Machine Learning’, PMLR, pp. 9659-9668.

---

> ### Author Response · Authors · 2022-11-18
> **Response to Reviewer Y35L (1/3)**
>
> We thank the reviewer for taking the time to review our paper and sharing insightful and thorough
> comments. We address all comments point by point below.
>
> **Comment 1: Discussion about existing offline RL algorithms**
>
> To study the connection between our algorithm and a line of offline RL works, we mainly discuss the popular line of offline RL works including fitted Q-iteration, fitted policy iteration, Bellman residual minimization, Gradient Q-learning and Advantage
> learning. The detailed discussion is provided in Section 5 and Appendix. We refer the reader to page 8 and page 17 in the updated manuscript.
>
> **Comment 2-3: Notations**
>
> Thank you for pointing this out. We have modified the notations accordingly in the updated manuscript.
>
> **Comment 4: Clarification regarding Equation (2) and (3).**
>
> * Equation 2
>
> Equation (2) holds for any $\rho$. We would like to clarify that Equation (2) is the definition of the generalized Fenchel representation for the standard Bellman operator $\mathcal{B}$. The confusion is caused by the misrepresentation of the Bellman operator on the LHS of Equation (2). We should use a newly defined notation, e.g., $\mathcal{B}^{\rho}$
> for discriminating it with $\mathcal{B}$.
> We set $\rho=0$ as a special case in our framework because it satisfies convexity and continuity conditions on $\rho$ without introducing any bias. Thus, $\mathcal{B}^{\rho=0} = \mathcal{B}$. The reason why we make $\rho$ explicitly in Equation (2) is that such explicit form is more straightforward for applying Fenchel transformation [1] or equivalent proximal approximation [2].
>
> However, we found the Fenchel representation, i.e., old Equation (2), causes unnecessary obstacles for the readers to follow. In the updated manuscript, we remove the old Equation (2), and directly claim the use of Fenchel transformation on Equation (1) to obtain new Equation (2). We believe this revision makes a more concise and clear presentation.
>
> The right-hand side of Equation (2) does involve state transition. $Q^* (s,a)$ is defined as $Q^*(s,a) = \mathbb{E}_{S^{t+1}|s,a}[R(S^{t+1},s,a)+\gamma V^*(S^{t+1})]$, which is analogous to the definition under the Equation (3).
>
> * The use of $\mu$
>
> To understand the use of $\mu$,  we can view it from two different perspectives essentially.
>
> **On the smoothing perspective**: The construction of Equation (3) is related to the Nesterov smoothing technique [3], and the scalar parameter $\mu$ controls the degree of the smoothness.
>
> **On the effect towards the quasi-optimal regions**: The parameter $\mu$ plays an important role in affecting the selection of the admissible actions. Based on Equation (5), a large $\mu$ allows more actions to be incorporated in the support set of the derived quasi-optimal policy. Reversely, the screening rule is more restrictive, and few actions are included in the support as admissible actions. For any fixed state $s \in \mathcal{S}$, the parameter $\mu$ is equivalent to the half of the integral difference between on action-value function $Q\_{\\mu}^{\*}(s,\\cdot)$ and on the lowest admissible action-value over the admissible action support set. In mathematical formulation, it is
> $$
> \\mu = \\frac{1}{2}\\left(\\int_{a\\in \\mathcal{W}\_s}Q\_{\\mu}^{\*}(s,a)da - \\sigma(\\mathcal{W}\_s) \\inf\_{a \\in \\mathcal{W}\_s}Q\_{\\mu}^{\*}(s,a)\\right),
> $$
> where $\mathcal{W}_s$ is the support set defined on page 5 underneath Equation (5).
>
> Aligned with an illustrating example of the scenario for the one-dimension action space in Figure 1. As shown in the left panel of Figure 1, the lowest admissible action-value corresponds to the red dash line, and the integral difference is the pink shape area. In the right panel, when $\mu$ decreases, the pink area (integral difference) shrinks, and the quasi-optimal sub-regions become narrower. In contrast, a larger $\mu$ allows a large pink area, the quasi-optimal sub-regions expand, and more action regions are included.
>
> [4] discusses the application of the Fenchel transformation in $\max$ operator in the  differentiable dynamic
> programs, this is close to our motivation for applying transformation on smoothing $\max$ operator in the standard Bellman operator in Equation (1).
>
> ## References
>
> [1] Hiriart-Urruty, J.-B. & Lemar ́echal, C. (2013), Convex analysis and minimization algorithms I:
> Fundamentals, Vol. 305, Springer science & business media
>
> [2] Chen, Y. (2019), ‘Smoothing for nonsmooth optimization, ele522 lecture notes’.
>
> [3] Nesterov, Y. (2005), ‘Smooth minimization of non-smooth functions’, Mathematical program-
> ming 103(1), 127–152.
>
> [4] Mensch, A. & Blondel, M. (2018), Differentiable dynamic programming for structured prediction
> and attention, in ‘International Conference on Machine Learning’, PMLR, pp. 3462–3471

---

> > ### Author Response · Authors · 2022-11-18
> > **Response to Comment 4 (Continued)**
> >
> > * The choice of $\mu$.
> >
> > For the choice of the parameter $\mu$. There are two ways: First, the $\mu$ can be user-specified according to prior knowledge. Second, the $\mu$ can be selected via a data-driven principle. The $\mu$ can be  determined by a $k$-fold cross-validation procedure. Specifically, we choose a proper $\mu$, which aims to  maximize the average of the lower bound of the estimated discounted sum of rewards, i.e.,
> > $$
> > \\hat \\mu = argmax_{\\mu}\frac{1}{k}\sum^k_{j=1}\\mathbb{P}\_{n(j)}\hat V^{(j)}\_{\\mu}(S^1)-\frac{\mu}{1-\gamma},
> > $$
> > where $\\hat V^{(j)}\_{\\mu}$ is the value function estimator on the $j$th training set, $\\mathbb{P}\_{n(j)}$ is the empirical measure on the initial state $S^1$ for the $r$th validation set. The second term $\frac{\mu}{1-\gamma}$ is used to mitigate the effect of the additional proximity term $\mathbb{E}_{a \sim \pi(\cdot|s)}[\text{prox}(\pi(a|s))]$.
> >
> >
> > In our empirical studies, we follow the data-driven way to select a proper $\mu$. To validate the cross-validation procedure works in practice, we  conduct sensitivities analyses with respect to the change of the $\mu$ in Figure 4. The results confirm that the cross-validation procedure indeed selects a proper $\mu$ which maximizes the discounted return.

---

### Official Review · Reviewer_RoFn · 2022-10-24

**Confidence:** 3
**Correctness:** 3
**Technical Novelty And Significance:** 3
**Empirical Novelty And Significance:** 2
**Recommendation:** 6

**Clarity, Quality, Novelty And Reproducibility:**

Clarity: All proofs appear in the appendix. There is not a lot of intuition given for why the authors take the approach that they do.

Quality: It is unclear how significant the work is (see weaknesses).

Novelty: The intuition behind wanting a q-Gaussian distribution is clear and this work is first to do it.

Reproducibility: code and access to data is provided.

**Strength And Weaknesses:**

Strengths:
1. The authors' arguments are comprehensively supported by theory.
2. The empirical performance against other algorithms is strong, even deterministic ones that do not suffer from off-support bias.

Weaknesses:
1. The paper is closely related to offline RL, but there is not a clear discussion on off-policy evaluation or avoiding unwanted behaviors. The experiments appear to use only offline data, but use algorithms that were not developed for offline or safe RL (see #2). The method appears to also be applicable online—it is not clear whether there is also an advantage in online settings.
2. There is no comparison in experiments with other safe RL approaches. Indeed, there is little discussion of safe RL research in the paper—it is unclear to me why this is. The motivation of the authors seems to lie at the intersection of offline and safe RL.
3. The appendix gives the impression that the hyperparameters were tuned for the authors' method but not for the others.

Post-response:
The authors have done substantial updates to the paper during the review period, including running new experiments. I am mainly satisfied with the paper as it stands and have updated my score.

**Summary Of The Paper:**

The authors address the problem of an unreliable policy caused by using a Gaussian distribution to represent it in a continuous control setting. They create a new quasi-optimal learning algorithm that uses the q-Gaussian distribution and show that it has provable convergence in off-policy settings. They also analyze other key theoretical aspects of the algorithm, such as sample complexity and consistency.

In a synthetic environment where there are bounded action regions, they show their method performs better than competitors. They additionally show that it performs better than others in an existing simulator for diabetes.

**Summary Of The Review:**

The authors present a thorough and mathematical analysis of a new algorithm that claims to address issues that arise in continuous action RL, but it is hard to tell how practical or useful the algorithm is because of limited experimental comparisons.

---

> ### Author Response · Authors · 2022-11-18
> **Response to Reviewer RoFn (3/3)**
>
> **Weakness 3: Hyperparameters for competing methods.**
>
> For all competing methods, we actually tuned the hyperparameters for their approaches. In the following, we fully disclose the choices of the hyperparameters and how we tune them.
>
> For the DeepRL-based continuous control methods, i.e., DDPG, SAC, BEAR, CQL and IQN, we implement them based on offline deep reinforcement learning library [1]. For the general optimization and function approximation settings,  we use a multi-layer perceptron (MLP) with 2 hidden layers. We set the batch size to be 64, and use ReLU function as the activation function. In addition to the summary provided below, the initial learning rate is chosen from the set $\\{3\times10^{-4},1\times10^{-4},3\times10^{-5}\\}$. We use Adam [2] as the optimizer for learning the neural network parameters.
>
> We disclose specific hyperparameters of each method in the following:
>
> SAC: The target smoothing coefficient is tuned by automatic tuning  [3]. The target update interval and gradient steps are both set to 1 as in [3].
>
> BEAR: The MMD constraint parameter is tuned over the candidate set $\\{0.1,0.25,0.5,0.75,1\\}$ as in [4]. The samples of MMD $n$ is tuned over the set $5,10,15$. The KL-control baseline uses automatic temperature tuning as in [4].
>
> DDPG: The soft target update is set to be $0.0001$ as in [5]. The $L_2$ weight decay is tuned in the set $\\{10^{-2}, 10^{-3}, 10^{-4}\\}$.
>
> CQL: We choose the value of Lagrangian threshold parameter from the set $\{5,10\}$, and the initial value of the tradeoff factor $\alpha$ is set to be $1$. We tried two Q-function learning rate values $\\{10^{-5}, 10^{-4}\\}$, we did not see significant difference. We tried the policy learning rate from the set $\\{3\times10^{-4},1\times10^{-4},3\times10^{-5}\\}$, we found $3\times10^{-5}$  attains good performance.
>
> IQN: We set the distortion risk measure $\eta=-0.75$ as in [6]. We choose a cosine embedding with $n = 64$ as in [7]. We tried three Lagrangian threshold parameters from the set $\\{4,6,8\\}$, and we found $6$ to attains the best performance.
>
> For the discretization-based methods (Greedy-GQ and V-learning), we discretize the original action space into 20 bins in synthetic experiments and 14 bins for real data analysis. The number of bins is chosen by analyzing the distribution of action, where too few bins could not lead to an accurate approximation, and too many bins may damage the performance of these methods. We use a radial basis to approximate value functions for these two methods based on the recommendation of the original implementation [8-9].
>
> Greedy-GQ: We choose the kernel bandwidth via the Silverman rule of thumb. And we set up the optimizer as L-BFGS with a learning rate from the set $\\{10^{-4}, 3\times10^{-4}, 10^{-5}\\}$. The maximum iterations steps are set up to $200$ as in [8].
>
> V-learning: We set the kernel bandwidth as $\tau=0.25$ as in [9], and select the penalty parameter $\lambda$ for V-function from the set $\\{0.1,0.5,1\\}$ by cross-validation, and set the penalty parameter for policy estimation as $\tau=0.001$. We set up the optimizer as L-BFGS, and select the learning rate from the set  $\\{10^{-4}, 3\times10^{-4}, 10^{-5}\\}$. The maximum iteration step is set to 100 as in [9].
>
> ## References
>
> [1] Seno, T. & Imai, M. (2021), ‘d3rlpy: An offline deep reinforcement learning library’, arXiv
> preprint arXiv:2111.03788 .
>
> [2] Kingma, D. P. & Ba, J. (2014), ‘Adam: A method for stochastic optimization’, arXiv preprint
> arXiv:1412.6980 .
>
> [3] Haarnoja, T., Zhou, A., Abbeel, P. & Levine, S. (2018), Soft actor-critic: Off-policy maximum
> entropy deep reinforcement learning with a stochastic actor, in ‘International conference on
> machine learning’, PMLR, pp. 1861–1870.
>
> [4] Kumar, A., Fu, J., Soh, M., Tucker, G. & Levine, S. (2019), ‘Stabilizing off-policy q-learning via
> bootstrapping error reduction’, Advances in Neural Information Processing Systems 32.
>
> [5] Lillicrap, T. P., Hunt, J. J., Pritzel, A., Heess, N., Erez, T., Tassa, Y., Silver, D. &
> Wierstra, D. (2015), ‘Continuous control with deep reinforcement learning’, arXiv preprint
> arXiv:1509.02971 .
>
> [6] Wang, S. S. (2000), ‘A class of distortion operators for pricing financial and insurance risks’,
> Journal of risk and insurance pp. 15–36.
>
> [7] Dabney, W., Rowland, M., Bellemare, M. & Munos, R. (2018), Distributional reinforcement
> learning with quantile regression, in ‘Proceedings of the AAAI Conference on Artificial Intel-
> ligence’, Vol. 32.
>
> [8] Ertefaie, A. & Strawderman, R. L. (2018), ‘Constructing dynamic treatment regimes over in-
> definite time horizons’, Biometrika 105(4), 963–977.
>
> [9] Luckett, D. J., Laber, E. B., Kahkoska, A. R., Maahs, D. M., Mayer-Davis, E. & Kosorok, M. R.
> (2019), ‘Estimating dynamic treatment regimes in mobile health using v-learning’, Journal of
> the American Statistical Association .

---

> ### Author Response · Authors · 2022-11-18
> **Response to Reviewer RoFn (2/3)**
>
> **2. Experiment results, analyses and comparisons with safe RL**
>
> For a more comprehensive comparison and illustration of the safety for the proposed method, we incorporate two safe RL algorithms for comparison.
>
> First, we update the model performance (discounted return) of the two safe RL algorithms in **Figure 3** and **Table 1** on page 8. Overall, quasi-optimal learning outperforms the two safe RL algos. The performance gains might come from the quasi-optimal regions identification and bypassing off-support bias issues in the continuous action domain. In addition to the model performance, we demonstrate the safety of quasi-optimal learning in the following **three dimensions** with experiments supports.
>
> **Dimension 1: Distribution Evaluation Criterion:** To measure the performance on safety, we aim to evaluate the distribution of Monte-Carlo discounted sum of rewards for each roll-out trajectory [12], instead of just its empirical mean, i.e., discounted return.
>
> In particular, we generate $100$ trajectories under the learned policies and record the discounted sum of rewards of each trajectory. Then we draw the density plots in **Figure 5** on page 44 for all four environments. As shown in Figure 5, the distribution of the quasi-optimal learning is left-skewed, which is aligned to two safe RL algos IQN and CQL. The left-skewed phenomenon indicates that there is less chance to follow a low reward trajectory which is damaged by allocating highly-risk actions. However, the non-safe RL algo SAC is no-skewed distributed;  and it rolls out low reward trajectories in higher probability (fat left tail) compared to the quasi-optimal learning and two safe RL baselines. This validates that quasi-optimal learning avoids risky actions as the other two safe RL algos.
>
> **Dimension 2: Safe Transition:** In this dimension,
> we illustrate the safety by evaluating the proportion of safe transition, i.e., from a fixed current state to a safe transition state. In OhioT1M case study, the goal is to maintain the glucose level in a safe range. The safe state in this study is defined as the state where the glucose level is within the range of 80-140 mg/dL. The reward function, i.e., the index of glycemic control [14],
> \$
> R_i^t=-\frac{\mathbb{1}(S_{i,1}^t>140)(S_{i,1}^t-140)^{1.1}+\mathbb{1}(S_{i,1}^t<80)(S_{i,1}^t-80)^2}{30}
> \$
> where $S_{i,1}^t$ is the glucose level of patient $i$ at $t$ decision stage. This setting of reward favor the safe range and penalize the risky scenario where the glucose level is out of the range of 80-140 mg/dL.
>
> The details of the evaluation procedure are summarized as follows. In OhioT1M data, we pick up the observed states which transited to risky states, i.e., the states out of the safe range of glucose level. On the picked-up states, we calculate the proportion of safe transition, in which the corresponding transition states are sampled from the transition kernel under the learned policy. The transition kernel is estimated by maximum likelihood estimation from the offline dataset.
>
> We summarize the results of the safe proportions on 1000 transition samplings in **Figure 7** on page 46. As shown, the quasi-optimal learning achieves **82.2%** safe proportions, which outperforms **67.3%** in safe RL baseline IQN and **44.6%** in non-safe RL baseline SAC. Thus, we may conclude that quasi-optimal learning enjoys a better safety guarantee when applied to the medical domain.
>
> **Dimension 3: The Learned Policy Distribution:** We illustrate the validity of the quasi-optimal policy distribution on a fixed state. In OhioT1M data, we select a patient state with a glucose level of $217$ mg/dL, which is moderate hyperglycemia. In this state, we draw a density plot in **Figure 8** on page 46 for the policy distribution learned by the quasi-optimal learning, IQN and SAC. **Figure 8** shows that the quasi-optimal learning identified a support region **$[3.15,6.19]$**. As the patient is under moderate hyperglycemia, so the moderate insulin dosage, i.e., $[3.15,6.19]$, works well to decrease the glucose level into a safe range. Meanwhile, it avoids overly dropping the patient's glucose level, which causes hypoglycemia. In comparison, SAC is risky as it has a non-negligible probability to assign too low and too high insulin dosage to the patient. The policy learned by the safe RL algorithm IQN has a tendency to avoid assigning extreme dosage, but it has wider support than the one learned by quasi-optimal learning. Regarding efficiency or safety, the quasi-optimal has certain advantages towards to IQN in this case.

---

> > ### Author Response · Authors · 2022-11-18
> > **Reference for Response to Reviewer RoFn (1/3 & 2/3)**
> >
> > ## References
> > [1] Garcıa, J. & Fern ́andez, F. (2015), ‘A comprehensive survey on safe reinforcement learning’, Journal of Machine Learning Research 16(1), 1437–1480.
> >
> > [2] Pham, T.-H., De Magistris, G. & Tachibana, R. (2018), Optlayer-practical constrained optimization for deep reinforcement learning in the real world, in ‘2018 IEEE International Conference
> > on Robotics and Automation (ICRA)’, IEEE, pp. 6236–6243.
> >
> > [3] Chow, Y., Nachum, O., Duenez-Guzman, E. & Ghavamzadeh, M. (2018), ‘A lyapunov-based approach to safe reinforcement learning’, Advances in neural information processing systems
> > 31.
> >
> > [4] Gu, S., Yang, L., Du, Y., Chen, G., Walter, F., Wang, J., Yang, Y. & Knoll, A. (2022), ‘A review of safe reinforcement learning: Methods, theory and applications’, arXiv preprint
> > arXiv:2205.10330.
> >
> > [5] Morimura, T., Sugiyama, M., Kashima, H., Hachiya, H. & Tanaka, T. (2010), Nonparametric return distribution approximation for reinforcement learning, in ‘ICML’.
> >
> > [6] Mavrin, B., Yao, H., Kong, L., Wu, K. & Yu, Y. (2019), Distributional reinforcement learning for efficient exploration, in ‘International conference on machine learning’, PMLR, pp. 4424–4434.
> >
> > [7] Vincent, R. (2014), Reinforcement learning in models of adaptive medical treatment strategies, McGill University (Canada).
> >
> > [8] Raghu, A., Komorowski, M., Ahmed, I., Celi, L., Szolovits, P. & Ghassemi, M. (2017), ‘Deep reinforcement learning for sepsis treatment’, arXiv preprint arXiv:1711.09602
> >
> > [9] Jia, Y., Burden, J., Lawton, T. & Habli, I. (2020), Safe reinforcement learning for sepsis treat-
> > ment, in ‘2020 IEEE International Conference on Healthcare Informatics (ICHI)’, IEEE, pp. 1–7.
> >
> > [10] Tamar, A., Glassner, Y. & Mannor, S. (2015), Optimizing the cvar via sampling, in ‘Twenty-
> > Ninth AAAI Conference on Artificial Intelligence’.
> >
> > [11] Ma, X., Xia, L., Zhou, Z., Yang, J. & Zhao, Q. (2020), ‘Dsac: distributional soft actor critic for
> > risk-sensitive reinforcement learning’, arXiv preprint arXiv:2004.14547.
> >
> > [12] Dabney, W., Ostrovski, G., Silver, D. & Munos, R. (2018), Implicit quantile networks for dis-
> > tributional reinforcement learning, in ‘International conference on machine learning’, PMLR,
> > pp. 1096–1105.
> >
> > [13] Kumar, A., Zhou, A., Tucker, G. & Levine, S. (2020), ‘Conservative q-learning for offline rein-
> > forcement learning’, Advances in Neural Information Processing Systems 33, 1179–1191.
> >
> > [14] Rodbard, D. (2009), ‘Interpretation of continuous glucose monitoring data: glycemic variability
> > and quality of glycemic control’, Diabetes technology & therapeutics 11(S1), S–55

---

> ### Author Response · Authors · 2022-11-18
> **Response to Reviewer RoFn (1/3)**
>
> We sincerely appreciate the reviewer's insightful comments and constructive suggestions, we address all the comments point by point below.
>
> **Weakness 1: Discussion with offline RL**
>
> We would like to clarify that the proposed algorithm is an off-policy learning (optimization) framework in a fully offline setting. The goal is to learn an optimal policy in a continuous action domain. To study the connection between our algorithm and a line of offline RL works, we provide a detailed discussion in Section 5 and Appendix. We refer the reader to page 8 and page 17 in the updated manuscript.
>
> **Weakness 2: Comparison and Discussion with safe RL algorithms**
>
> Algorithmically, the quasi-optimal learning is closely related to entropy-regularized RL framework. And the goal of the paper target is to tackle the challenges continuous control problem with reliable action recommendations in medical settings. Therefore, we mainly frame our work in the entropy-regularized and continuous action domain.
>
> However, we greatly thank the reviewer for pointing out the interesting line of works in safe RL. Follow the reviewer's suggestion, we investigate the relationship between the proposed algorithm and safe RL. Also, we conduct addition experiments with two safe RL baseline algorithms for comprehensive comparison. The illustration of the safety guarantee of the qausi-optimal learning is also discussed.
>
> **1. Connection between the safe RL and the proposed algorithm**
> Safe Reinforcement Learning (safe-RL) aims at finding an optimal policy while ensuring safety [1]. In the safe-RL framework, the definition of safety and its guarantee varies based on the specific purpose of learning tasks. In our view, there are three mainstream works for safe RL.
>
> * **Safe Exploration**: ensuring safe action allocations in the exploration process by incorporating prior knowledge, which often exists in online RL settings [2].
> * **Safety Constraints**: finding an optimal policy that satisfies external user-specified safe constraints [3-4].
> * **Risk-sensitivity and Conservatism**: finding a policy maximizing the infinite-horizon cumulative discounted reward while incorporating the notion of risk [5-6], e.g., value at risk (quantile), percentile performance, the variance of return.
>
> In medical applications, specifying explicit constraints is typically hard to realize [7]. Alternatively, the notion of safety is usually incorporated in the design of reward functions, where high-risk actions lead to significantly low reward [8-9].
>
> Based on these, our quasi-optimal learning is closely related to the risk-sensitive RL framework, which aims to control value at risk to ensure safety. For example, maintaining the discounted return above a certain threshold [10], reducing the variability of performance by avoiding extremely low performance [11], or target to maximize the robust performance criterion, e.g., quantile of the discounted return [12]. Commonly used algorithms in risk-sensitive RL include conservative Q-learning (CQL; [13]) and implicit quantile network (IQN; [12]). CQL learns a conservative Q-function such that the expected value of a policy under this Q-function lower-bounds its true value and thus avoids selecting high-risk actions with over-estimation action value. IQN models the full quantile function for the state-action return distribution and yields risk-sensitive policies. For a more comprehensive empirical study, **we compare the proposed algorithm with the aforementioned two safe RL baselines, conduct additional numerical experiments and analyze the results from the safety point of view.** See the following response: part 2. Experiment results, analyses and comparisons with safe RL.

---

### Author Response · Authors · 2022-11-18
**General Response**

We want to thank all the reviewers for taking the time to review our paper and sharing insightful comments. Before we reply to each post individually, we summarize the major changes made to our manuscript as follows:

* We provide detailed discussion on the related work of offline RL, safe RL and applying RL to healthcare. And disclose the connection between the quasi-optimal and the closet related works in these three frameworks.

* We strength the arguments on the usage and choice of the proximity threshold parameter $\mu$, and its affects on the proposed quasi-optimal regions.

* We investigate the continuity invariance between the reward amd optimal weight function $u^*$ for better motivation of our kernel representation.

* We conduct additional experiment including thorough comparison to two safe RL algorithm and one more discretization-based sample efficient method.

* We provide a justification and illustration of the safefy for our quasi-optimal learning on three aspects with numerical experiments supports.

---

### Decision · Program_Chairs · 2023-01-20

**Decision:**

Accept: poster

**Justification For Why Not Higher Score:**

This paper can also be a spotlight.

**Justification For Why Not Lower Score:**

The problem is well-motivated. The result is novel, and both theory and empirical parts are well developed.

**Metareview: Summary, Strengths And Weaknesses:**

This paper addresses learning in continuous action-spaces where circumspection of "optimal" behavior is necessary, specifically focused on healthcare. The paper proposes a method by which to constrain the policy search space to only those areas that are "quasi-optimal". Detailed theoretical justification and analysis of the proposed quasi-optimal Bellman operator is provided as well as preliminary application of the approach to a diabetes management dataset.

Reviewers find the problem considered in this paper well-motivated and novel. Reviewers are satisfied with both theoretical and empirical results developed in this paper. During the rebuttal phase, the authors have added new experiments and comparisons with prior literature which successfully address the major concerns of reviewers. We thus no doubt recommend acceptance.

**Note From Pc:**

if the above contains the word "oral" or "spotlight" please see: "oral" presentation means -> notable-top-5% and "spotlight" means -> notable-top-25%. As stated in our emails, we are disassociating presentation type from AC recommendations